# A Theoretical Analysis of Discrete Flow Matching Generative Models

## Abstract

We provide a theoretical analysis for end-to-end training Discrete Flow Matching (DFM) generative models. DFM is a promising discrete generative modeling framework that learns the underlying generative dynamics by training a neural network to approximate the transformative velocity field. Our analysis establishes a clear chain of guarantees by decomposing the final distribution estimation error. We first prove that the total variation distance between the generated and target distributions is controlled by the risk of the learned velocity field. We then bound this risk by analyzing its two primary sources: (i) Approximation Error, where we quantify the capacity of the Transformer architecture to represent the true velocity, and (ii) Estimation Error, where we derive statistical convergence rates that bound the error from training on a finite dataset. By composing these results, we provide the first formal proof that the distribution generated by a trained DFM model provably converges to the true data distribution as the training set size increases.

## 1 Introduction

We provide a comprehensive theoretical analysis of Discrete Flow Matching (DFM), establishing rigorous error bounds and statistical convergence rates for training this emerging class of models. Generative models for discrete data, such as text, proteins, and molecules, are central to modern machine learning. Recently, discrete flow matching (Campbell et al., 2024; Gat et al., 2024) emerges as a powerful and flexible paradigm in these domains. It learns a transformation from a simple prior distribution to a complex data distribution by parameterizing the dynamics of a Continuous-Time Markov Chain (CTMC). A key advantage of this approach is its simulation-free training objective. Instead of solving complex differential equations, discrete flow matching models learn the underlying *velocity field* that governs the probability path, leading to efficient and stable training. This framework has achieved promising results in various applications, including video generation (Fuest et al., 2025), inverse protein folding (Yi et al., 2025), and graph generation (Qin et al., 2024).

Despite its rapid adoption and strong empirical performance, the theoretical foundations of discrete flow matching remain unexplored. This creates a critical gap between practice and theory: How does the error in the learned velocity field translate to error in the final generated distribution? What are the expressive limits of the neural networks, like Transformers, used to parameterize these velocities? And how does the quality of the generated samples depend on the amount of training data? Without answers to these questions, it is difficult to understand the discrete flow matching method's behavior, its fundamental limitations, or how to guide future improvements in a principled manner.

This paper addresses these gaps by providing the comprehensive theoretical analysis of end-to-end training for Discrete Flow Matching. We focus on the popular setting of *factorized velocities* (Lipman et al., 2024) parameterized by the Transformer architecture (Vaswani et al., 2017; Gat et al., 2024), the major workhorse of modern generative AI. Our analysis establishes a clear chain of guarantees connecting model design and data size to the quality of the resulting distribution.

**Contributions.** Our contributions are three-fold:

- **Intrinsic Error Bounds for Discrete Flow Matching.** We establish a fundamental error bound intrinsic to the Discrete Flow Matching (DFM) framework. Our analysis begins with the Kolmogorov equation (2.3), which governs the relationship between a probability

distribution and its underlying velocity field in a Continues Time Markov Chain Section 2. We frame the problem as analyzing the discrepancy between the solutions to two systems of Kolmogorov ODEs (Lemma C.3): one for the true data distribution $p_t$ and its velocity field $u_t$, and the other for the estimated distribution $p_t^\theta$ driven by our learned velocity field $u_t^\theta$. By applying Grönwall's Inequality (Lemma C.2), we derive an explicit upper bound on the total variation distance between the true distribution $p_t$ and the estimated distribution $p_t^\theta$ in (Theorem 3.1). This bound is *intrinsic* because it originates from the core discrete flow matching paradigm of modeling the velocity field rather than the distribution—an inherent source of error that exists irrespective of model architecture or data volume. This result validates the intuition that a more accurate velocity field approximation yields a higher-fidelity generative model. It provides a distribution convergence guarantee for any discrete flow matching implementation, including empirical works that do not use Transformers (Campbell et al., 2024; Gat et al., 2024; Lipman et al., 2024).

- **Approximation Error Analysis.** We analyze the approximation error, proving that Transformer networks possess sufficient expressive power to approximate the ground-truth velocity fields with a controlled error rate. A key challenge is that the existing universal approximation results for Transformers (Appendix B) limits to continuous functions, whereas our velocity field $u(x,t)$ is defined over a discrete space $x \in \mathcal{S}$. We bridge this theoretical gap in Lemma 4.5 by first constructing a continuous extension, $\widetilde{u}(x,t)$, that preserves the temporal smoothness of the discrete velocity function. Building on this, we then derive an upper bound on the approximation error when using a Transformer estimator $u_\theta(x,t)$ to model the ground-truth velocity $u(x,t)$ (Theorem 4.7). This result provides a formal justification for using Transformers to model discrete flows.

- **Estimation Error Analysis.** We derive statistical convergence rates for the estimation error, which arises from learning the velocity field from a finite dataset. Our analysis proceeds in two stages. First, leveraging our approximation error bounds from Theorem 4.7, we analyze the velocity estimation error in Theorem 5.1. This result establishes a rate at which the Transformer-based velocity estimator converges to the true field as the number of training samples increases. Second, we combine this velocity estimation error with the intrinsic error bound for discrete flow matching (Theorem 3.1) to derive a final upper bound on the distribution estimation error in Theorem 5.2. Together, these theorems provide a solid theoretical guarantee for discrete flow matching models implemented with Transformers, grounding existing empirical applications (Fuest et al., 2025; Qin et al., 2024; Yi et al., 2025) in a rigorous framework.

**Organization.**  Section 2 reviews the core concepts of discrete flow matching and the Transformer architecture. Section 3 establishes our intrinsic error bound for the Discrete Flow Matching framework. Section 4 provides an approximation error analysis for discrete flow matching implemented with Transformers. Section 5 derives statistical convergence rates for both velocity and distribution estimation errors. Section 7 summarizes our contributions and discusses their implications. The appendix provides supplementary material and proofs. Appendix B details the theoretical background on the expressive power of Transformers. Appendices C and E to G contain detailed proofs of our main theorems. Appendices H and I study approximation and estimation rates for generic DFM without the factorized velocity technique.

**Notation.**  We denote the index set $\{1, \ldots, I\}$ by $[I]$. Let $x[i]$ denote the $i$-th component of a vector $x$. Let $\mathbb{Z}$ denote integers and $\mathbb{Z}_+$ denote positive integers. Given discrete probability distribution $P$ and $Q$, we denote the total variation distance between $P$ and $Q$ by $\mathrm{TV}(P, Q)$. Given a matrix $Z \in \mathbb{R}^{d \times L}$, $\|Z\|_1$ and $\|Z\|_\mathrm{F}$ denote the induced 1-norm and the Frobenius norm. For vectors $u, v \in \mathbb{R}^d$, the Bregman divergence induced by a strictly convex function $\Phi : \mathbb{R}^d \to \mathbb{R}$ is $D(u, v) := \Phi(u) - \Phi(v) - (u-v)^\top \nabla\Phi(v)$. Let $\|\cdot\|_1$ and $\|\cdot\|_2$ be the $\ell_1$ and $\ell_2$ vector norms. Let $\mathcal{S} = \{s_1, \ldots, s_N\}$ be a finite state space. For each $t \in [0, 1]$, let $f_t : \mathcal{S} \times \mathcal{S} \to \mathbb{R}$ be a scalar-valued function. Then we define a vector-valued function $f_t(\cdot, x) : \mathcal{S} \times [0, 1] \to \mathbb{R}^N$, as $f_t(\cdot, x) := [f_t(s_1, x), \ldots, f_t(s_N, x)]^\top$.

## 2 PRELIMINARIES

In this section, we provide an high level review of discrete flow matching following (Lipman et al., 2024), and the transformer architecture (Vaswani et al., 2017).

**Continues Time Markov Chain.** Consider the discrete data $x$ from state space $\mathcal{S} = \mathcal{V}^d$ where vocabulary $\mathcal{V} = \{1, \ldots, M\}$. In this paper, we utilize a natural embedding $E : \mathcal{S} \hookrightarrow \mathbb{R}^d$ that maps each discrete token $j \in \mathcal{V}$ to its corresponding integer value as a real number.. For convenience, we view $\mathcal{V} = [M]$ as a subspace of $\mathbb{R}$. The Continues Time Markov Chain (CTMC) (Norris, 1998) is a continuous stochastic process $(X_t)_{t \geq 0}$ that models systems evolving over continuous time. A defining characteristic of a Continues Time Markov Chain is the Markov property, meaning the system's future state only depends on its current state, not on its past history. Let $p_t$ denote the probability mass function (PMF) of $X_t$. Then we define an unique Continues Time Markov Chain by specifying an initial distribution $p_0$ and rates function (velocity field) $u_t(y, x) : \mathcal{S} \times \mathcal{S} \to \mathbb{R}$. This function induces the probability transition kernel $p_{t+h|t}$ as

$$p_{t+h|t}(y|x) := P(X_{t+h} = y|X_t = x) = \delta(x, y) + u_t(y, x)h + o(h), \tag{2.1}$$

where $\delta(x, y)$ is the Kronecker delta function, equal to 1 when $x = y$ and 0 otherwise. The values $u_t(y, x)$, called rates or velocities, represent the instantaneous rate of transition from state $x$ to state $y$ at time $t$. We define $u_t$ *generates* $p_t$ if there exists $p_{t+h|t}$ satisfying (2.1) with probability path $(p_t)_{t \geq 0}$. For the total probability to sum to one, i.e., $\sum_y p_{t+h|t}(y|x) = 1$, the rates function $u_t$ must satisfy the following conditions (rates conditions),

$$u_t(y, x) \geq 0 \quad \text{for all} \quad y \neq x, \quad \text{and} \quad \sum_y u_t(y, x) = 0. \tag{2.2}$$

By the definition of transition kernel (2.1), a rates function $u_t$ and an initial distribution $p_0$ define a unique probability path $p_t$ via the Kolmogorov Equation (Lipman et al., 2024, Theorem 12),

$$\frac{\mathrm{d}p_t(y)}{\mathrm{d}t} = \sum_{x \in S} u_t(y, x)p_t(x). \tag{2.3}$$

Finally, we simulate a sample trajectory $(x_t)_{t \geq 1}$ with Euler method

$$P(X_{t+h} = y|X_t = x) = \delta(x, y) + u_t(y, x)h, \quad \text{with} \quad P(X_0) = p_0(x).$$

**Discrete Flow Matching.** Discrete Flow Matching (DFM) is a generative modeling framework that learns a transformation from a source distribution $p_0$ to a target distribution $p_1$ (Campbell et al., 2024; Gat et al., 2024; Lipman et al., 2024). The core principle is to first define a probability path $(p_t)_{t \in [0,1]}$ that interpolates between $p_0$ and $p_1$. This path is induced by a Continuous-Time Markov Chain (CTMC) characterized by a velocity field $u_t$. The learning objective is to train a neural network $u_t^\theta$ to approximate this ground-truth velocity. We train the model by minimizing the discrete flow matching loss, which measures the discrepancy between the ground-truth velocity $u_t$ and predicted velocities $u_t^\theta$ using a Bregman divergence $D(\cdot, \cdot)$ (see Section 1 for definition)

$$\mathcal{L}_{\mathrm{DFM}} = \mathbb{E}_{t, X_t \sim p_t} \left[ D(u_t(\cdot, X_t), u_t^\theta(\cdot, X_t)) \right],$$

where the ground-truth velocity $u(\cdot, X_t)$ (notation follows Section 1) satisfies the rate conditions in (2.2). A tractable method for constructing these paths and velocities is Conditional Discrete Flow Matching (CDFM) (Campbell et al., 2024; Gat et al., 2024), which introduces an auxiliary discrete random variable $Z$ over a space $\mathcal{Z}$ with PMF $p_Z(z)$. The marginal probability path is defined as

$$p_t(x) = \sum_{z \in \mathcal{Z}} p_{t|Z}(x|z)p_Z(z).$$

As shown by (Lipman et al., 2024), if each conditional path $p_t(x|z)$ is generated by a velocity $u_t(y, x|z)$, the corresponding marginal velocity $u_t(x)$ is given by

$$u_t(x) = \sum_{z \in \mathcal{Z}} u_t(y, x|z)p_{Z|t}(z|x), \quad \text{where} \quad p_{Z|t}(z|x) = \frac{p_t(x|z)p_Z(z)}{p_t(x)}.$$

This leads to the CDFM loss, an objective based on the conditional velocity fields

$$\mathcal{L}_{\text{CDFM}} = \mathbb{E}_{t, Z \sim p_Z, X_t \sim p_{t|Z}} \left[ D(u_t(\cdot, X_t|Z), u_t^\theta(\cdot, X_t)) \right].$$

Crucially, the CDFM and DFM objectives yield identical learning gradients (Lipman et al., 2024, Theorem 15), i.e., $\nabla_\theta \mathcal{L}_{\text{CDFM}}(\theta) = \nabla_\theta \mathcal{L}_{\text{DFM}}(\theta)$. This equivalence makes CDFM a powerful and efficient training strategy. In this paper, we instantiate the Bregman divergence as the squared $\ell_2$ distance. Then the conditional flow matching loss takes the form

$$\mathcal{L}_{\text{CDFM}} = \mathbb{E}_{t, Z, X_t \sim p_{t|Z}} \left[ \|u_t(\cdot, X_t|Z) - u_t^\theta(\cdot, X_t)\|_2^2 \right]. \tag{2.4}$$

**Factorized Paths and Velocities.** For sequences of length $d$ over a vocabulary of size $M$, the velocity field $u_t(\cdot, x)$ must specify a transition rate to all $M^d$ possible states. The model's output is therefore a vector in $\mathbb{R}^{M^d}$. This exponential scaling with sequence length makes the direct modeling of the velocity field intractable. To overcome this challenge, we employ factorized velocities (Campbell et al., 2022; 2024; Gat et al., 2024), which decompose the velocity field as

$$u_t(y, x) = \sum_i \delta(y^{\bar{i}}, x^{\bar{i}}) u_t^i(y^i, x), \tag{2.5}$$

where $\delta(\cdot, \cdot)$ is the Kronecker delta and $\bar{i} := (1, ..., i-1, i+1, ..., d)$ denotes all indices except $i$. We then model each component $u_t^i(y^i, x)$ with a neural network $u_t^{\theta, i}(y^i, x)$, which outputs a vector in $\mathbb{R}^M$. This reduces the total output dimension to a tractable $d \cdot M$. Substituting the factorized velocity (2.5) into the CDFM objective (2.4) yields the following loss function

$$\mathcal{L}_{\text{CDFM}} = \mathbb{E}_{t, Z, X_t \sim p_{t|Z}} \left[ \|u_t(\cdot, X_t|Z) - u_t^\theta(\cdot, X_t)\|_2^2 \right], \qquad \text{(By the definition of CDFM (2.4))}$$

$$= \mathbb{E}_{t, Z, X_t \sim p_{t|Z}} \left[ \sum_{y \in V^d} \sum_{i \in [d]} \delta^2(y^{\bar{i}}, X_t^{\bar{i}}) \left( u_t^i(y^i, X_t|Z) - u_t^{\theta, i}(y^i, X_t) \right)^2 \right], \qquad \text{(By (2.5))}$$

$$= \mathbb{E}_{t, Z, X_t \sim p_{t|Z}} \left[ \sum_{i \in [d]} \|u_t^i(\cdot, X_t|Z) - u_t^{\theta, i}(\cdot, X_t)\|_2^2 \right]. \tag{2.6}$$

To generate samples from the trained model (Campbell et al., 2024; Gat et al., 2024), we simulate the CTMC by applying coordinate-wise updates for each $i \in [d]$ using a discrete time step $h$

$$P(X_{t+h}^i = y^i | X_t = x) = \delta(x^i, y^i) + h u_t^{\theta, i}(y^i, x). \tag{2.7}$$

**Mixture Paths.** Following (Gat et al., 2024; Lipman et al., 2024), we adopt mixture paths for our strategy for *conditional* generation. By conditioning on the source-target pair $Z = (X_0, X_1)$, we construct a factorized conditional probability path $p_{t|0,1}(x|x_0, x_1) = \prod_i p_{t|0,1}^i(x^i|x_0, x_1)$, where each per-coordinate path interpolates between the source and target tokens

$$p_{t|0,1}^i(x^i|x_0, x_1) = \kappa_t \delta(x^i, x_1^i) + (1 - \kappa_t)\delta(x^i, x_0^i).$$

Here, $\delta(\cdot, \cdot)$ is the Kronecker delta and $\kappa_t$ is a monotonically increasing smooth function that satisfies the boundary conditions

$$\kappa_0 = 0, \quad \kappa_1 = 1, \quad \text{and} \quad \frac{\mathrm{d}\kappa_t}{\mathrm{d}t} > 0 \quad \text{for} \quad t \in (0, 1).$$

The conditional factorized velocity field that generates this per-coordinate path takes the form

$$u_t^i(y^i, x^i|x_0^i, x_1^i) = \frac{\dot{\kappa_t}}{1 - \kappa_t} [\delta(y^i, x_1^i) - \delta(y^i, x^i)]. \tag{2.8}$$

We parameterize the velocity $u_t^i$ with a model $u_t^{\theta, i}$. The model is trained to match the ground-truth velocity for each coordinate $u_t^i$, resulting in the following CDFM loss objective

$$\mathcal{L}_{\text{CDFM}} = \mathbb{E}_{t, X_0, X_1, X_t \sim p_{t|X_0, X_1}} \left[ \sum_{i \in [d]} \|\frac{\dot{\kappa_t}}{1 - \kappa_t} [\delta(\cdot, X_1^i) - \delta(\cdot, X_t^i)] - u_t^{\theta, i}(\cdot, X_t)\|_2^2 \right], \tag{2.9}$$

Figure 1: **Discrete Flow Matching (with Factorized Velocity) Network Architecture.** Our model processes a discrete input $x \in \mathcal{S}$ and a continuous time $t \in [0, 1]$ as input. Initially, embedding layer $E$ maps discrete tokens to continuous embeddings (Section 2). Sequentially, the model concat the continues embeddings with the time variable. A reshape layer then structures this combined representation into a sequence format with a hidden dimension of $d_0$, making it compatible with the Transformer network. This Transformer block processes the sequence to learn the complex temporal dynamics of the discrete flow. Finally, a reverse reshape layer flattens the output for a linear projection that predicts the underlying velocity field over the vocabulary space.

where $\delta(\cdot, z)$ denotes a one-hot vector in $\mathbb{R}^M$ corresponding to a token $z \in \mathcal{V}$. For notational simplicity, we define $u(x, t) := u_t(\cdot, x)$ as the vector-value function representing the full velocity field for a state $x$ at time $t$. This function maps the state-time space $S \times [0, 1]$ to the velocity space $\mathbb{R}^{M^d}$. Similarly, we apply this convention to the learnable model and their factorized counterparts: $u_\theta(x, t) := u_t^\theta(\cdot, x)$, $u^i(x, t) := u_t^i(\cdot, x)$, and $u_\theta^i(x, t) := u_t^{\theta, i}(\cdot, x)$.

This paper focuses on discrete flow matching (DFM) using factorized velocities and the mixture path construction, as these are most common choices in practice (Campbell et al., 2024; Gat et al., 2024; Lipman et al., 2024). Given $n$ i.i.d training $\{x_i\}_{i=1}^n$ , the factorized empirical loss used to train the velocity model for coordinate $i_0$ is defined as:

$$\widehat{\mathcal{L}}_{\mathrm{CDFM}}^{i_0} := \frac{1}{n} \sum_{i=1}^n \int_{t_0}^T \mathop{\mathbb{E}}_{X_0 \sim p_0, X_t \sim p_{t|x_0 = X_0, x_1 = x_i}} \| \frac{\dot{\kappa}_t}{1 - \kappa_t} [\delta(\cdot, x_i^{i_0}) - \delta(\cdot, X_t^{i_0})] - u_\theta^{i_0}(X_t, t) \|_2^2 \mathrm{d}t.$$

$$(2.10)$$

**Discrete Flow Matching Transformers.** We parameterize the velocity model $u_t^\theta$ using a Transformer architecture (Vaswani et al., 2017). Due to space limits, we defer a detailed definition of the Transformer block and its theoretical properties to Appendix B, including Lipschitzness and universal approximation. We also illustrate the specific architecture used in our paper in Figure 1.

## 3 ERROR BOUNDS FOR DISCRETE FLOW MATCHING

Instead of estimating the distribution paths $p_t$, the Discrete Flow Matching (DFM) framework (Campbell et al., 2024; Gat et al., 2024) learns the underlying dynamics by estimating the velocity field $u_t$. This section provides the first theoretical verification for this approach by establishing rigorous error bounds for the discrete flow matching (with factorized velocity). We prove that the quality of the generated distribution is controlled by the accuracy of the learned velocity field. We formalize this relationship by presenting an upper bound on the total variation distance between the estimated distribution $\widehat{P}$ and target distributions $P$, in terms of the risk of the velocity estimator.

**Theorem 3.1** (Error Bound for Discrete Flow Matching). *Consider the discrete state space $\mathcal{S} = \mathcal{V}^d$ with vocabulary $\mathcal{V} = \{1, \dots, M\}$. Let $P$ be the true data distribution and let $\widehat{P}$ be the distribution generated by a DFM model using factorized velocity estimators $\widehat{u}_\theta^1, \dots, \widehat{u}_\theta^d$. For each coordinate $i_0 \in [d]$, define the factorized risk as the mean squared error of its velocity estimator:*

$$\mathcal{R}^{i_0}(\widehat{\Theta}) := \int_{t_0}^T \mathop{\mathbb{E}}_{X_t \sim p_t(x)} \| u^{i_0}(X_t, t) - \widehat{u}_\theta^{i_0}(X_t, t) \|_2^2 \mathrm{d}t,$$

*where the time interval is clipped to $[t_0, T]$ to ensure numerical stability and $p_t(x)$ is true probability path generated by factorized velocities $u^1, \dots, u^d$. Then, the total variation distance between the*

*true and generated distributions is bounded by the sum of the risks from each factorized component:*

$$\mathrm{TV}(P, \widehat{P}) \lesssim \sqrt{M} \exp(2M_u) \sum_{i_0 \in [d]} \sqrt{\mathcal{R}^{i_0}(\widehat{\Theta})},$$

*where $M_u$ is the upper bound of estimated velocity such that $\left| u_t^{\theta, i_0}(y, x) \right| \leq M_u$ for all $y, x \in \mathcal{S}$.*

*Proof.* Please see Appendix C for a detailed proof. $\qquad\square$

**Remark 3.2** (Comparison with Flow Matching Error Bounds). *Our error bounds Theorem 3.1 provides an analogue to flow matching bounds (in 2-Wasserstein distance) like (Benton et al., 2023), with foundational differences in technique. We bound the solution error of the Kolmogorov forward equation governing the probability distribution. In contrast, their approach uses the Alekseev-Gröbner formula to control the trajectory-wise error of the underlying flow ODE. This technical distinction leads to different complexity sources: total variation distance bound scales exponentially with upper bound of the estimated velocity $M_u$, whereas their 2-Wasserstein bound scales with the velocity field's Lipschitz constant $L$.*

Theorem 3.1 confirms that the central challenge in discrete flow matching is to learn factorized velocity estimators $\widehat{u}_\theta^{i_0}$ with low risk $\mathcal{R}^{i_0}(\widehat{\Theta})$. Therefore, the subsequent sections analyzes the two primary sources of this risk: (i) Section 4: the approximation error (the error arising from learning velocity estimators with neural networks, which is the inherent limitation of model class) and (ii) Section 5: the estimation error (the error arising from training on a finite dataset).

**Roadmap of Our Theoretical Results.** For the convenience of readers, we provide the logical structure of our theoretical results in Figure 2 below. It illustrates the progression from supporting lemmas to intermediate error bounds. Altogether, these bounds culminate in our four main results (error bounds for discrete flow matching): intrinsic (Theorem 3.1), approximation (Theorem 4.7), velocity estimation (Theorem 5.1), and distribution estimation (Theorem 5.2).

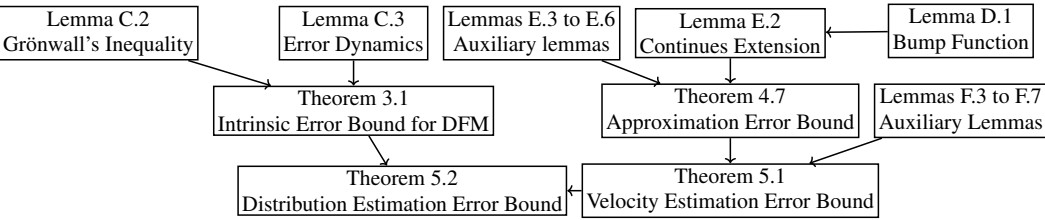

Figure 2: **Roadmap of Our Theoretical Results.**

## 4 APPROXIMATION ERROR FOR DISCRETE FLOW MATCHING

This section addresses the first component of the learning error: the approximation error of discrete flow matching with transformers. We focus on the transformers since the transformers are the foundational architecture in many of today's most powerful generative models based on discrete flows, such as (Fuest et al., 2025), inverse protein folding (Yi et al., 2025), and graph generation (Qin et al., 2024). Section 4.1 embeds the discrete ground-truth velocity field $u(x, t)$ into a continuous space $\mathbb{R}^{M^d}$. Section 4.2 presents the approximation error bounds for discrete flow matching.

### 4.1 EXTENDING THE VELOCITY FIELD

To analyze our model's approximation error, which operates on the discrete domain $S = \mathcal{V}^d$, we require a continuous extension of the ground-truth velocity field. Therefore, we first embed the discrete input space $S$ into the continuous Euclidean space $\mathbb{R}^d$, following Section 2.

**Remark 4.1.** *While we present our proofs using the inclusion embedding, the functional extension technique in Lemma 4.5 applies to any frozen injective embedding of the vocabulary. Consequently, analogues of our main results continue to hold for arbitrary injective embeddings. We formulate our main results under the inclusion embedding for simplicity of analysis.*

Applying this embedding, we then extend the velocity function $u(x, t)$ to a continuous function $\widetilde{u}(z, t)$ defined over $z \in \mathbb{R}^d$. The goal is to construct this extension $\widetilde{u}$ such that it preserves the smoothness of the original function. We quantify this smoothness using the Hölder space.

**Definition 4.2** (Hölder Class). *Let $d, d' \in \mathbb{Z}^+$, $\Omega \subset \mathbb{R}^d$, and let $\beta = k + \gamma$ be the smoothness parameter with $k = \lfloor \beta \rfloor \in \mathbb{Z}_{\geq 0}$ and $\gamma \in [0, 1)$. For a $k$-times differentiable function $f : \Omega \to \mathbb{R}$, the Hölder norm is defined as*

$$\|f\|_{\mathcal{H}^\beta(\Omega)} := \sum_{\|\alpha\|_1 \leq k} \|\partial^\alpha f\|_{L^\infty(\Omega)} + \sum_{\|\alpha\|_1 = k} \sup_{\substack{x,y \in \Omega \\ x \neq y}} \frac{|\partial^\alpha f(x) - \partial^\alpha f(y)|}{\|x - y\|^\gamma}.$$

*The Hölder class with smoothness $\beta$ and radius $K > 0$ is then*

$$\mathcal{H}^\beta_{d,d'}(\Omega, K) := \{f = (f_1, \ldots, f_{d'})^\top : \Omega \to \mathbb{R}^{d'} \mid \sup_{i \in [d']} \|f_i\|_{\mathcal{H}^\beta(\Omega)} \leq K\}.$$

Our analysis requires the ground-truth velocity field $u(x, t)$ to be smooth in time. We assume that for any fixed state $x \in \mathcal{S}$, the velocity function $t \mapsto u(x, t)$ is Hölder continuous, as stated below.

**Assumption 4.3.** *For each state $x \in \mathcal{S}$, the true velocity function $t \mapsto u(x, t)$ lies in the Hölder space $\mathcal{H}^\beta_{1,|S|}([0, 1], K)$ for some smoothness parameter $\beta \geq 1$.*

**Remark 4.4.** *This is a standard assumption in the analysis of differential equations, ensuring the velocity field changes smoothly over time. Assumption 4.3 is not restrictive and holds for many common probability path constructions. A prominent example is the mixture path. The velocity that generates the mixture path (2.8) is smooth with respect to time $t$. This ensures that for any smoothness level $\beta \geq 1$, the condition in Assumption 4.3 is satisfied.*

Then the following lemma demonstrates that a smooth extension $\widetilde{u}(z, t)$ exists that interpolates the original function while preserving its smoothness.

**Lemma 4.5** (Discrete-to-Continuous Functional Extension). *Let $\mathcal{S} \subset \mathbb{R}^d$ be the discrete state space. For each $x \in \mathcal{S}$, let $t \mapsto u(x, t) \in \mathcal{H}^\beta_{1,M^d}([0, 1], K)$ with $\beta = k_1 + \gamma \geq 1$, where $k = \lfloor \beta \rfloor$ and $\gamma \in [0, 1)$. Then there exists an continuous extension $\widetilde{u} \in \mathcal{H}^\beta_{d+1,M^d}(\mathbb{R}^d \times [0, 1], C)$ such that*

$$\widetilde{u}(s, t) = u(s, t) \quad \text{for all } s \in \mathcal{S}, t \in [0, 1],$$

*where the Hölder norm $C = 3e \cdot (k_1 + 2)(2k_1)^{2k_1} K M^d$.*

*Proof.* Please see Appendix D for a detailed proof. □

**Remark 4.6.** *This lemma shows that it is possible to extend a family of smooth functions indexed by discrete points to a smooth function on the whole domain with controlled Hölder norm.*

### 4.2 DISCRETE FLOW MATCHING APPROXIMATION

Building on the continuous extension of the velocity field from Section 4.1, we now derive a specific approximation rate for the discrete flow matching model. Following practical implementation (Campbell et al., 2024; Gat et al., 2024; Lipman et al., 2024), our analysis in this section focuses on the setting that combines factorized velocities with the mixture path construction (Section 2).

**Theorem 4.7** (Informal Version of Theorem E.7: Approximation Theorem for Discrete Flow Matching). *Let $u^i(x, t)$ be the factorized velocity field for coordinate $i \in [d]$ under mixture path setting. Let $M$ be the vocabulary size. Assume Assumption 4.3 holds, then for any $\epsilon \in (0, 1)$, there exists a transformer network $u^i_\theta(x, t)$ satisfying that for any $t \in [t_0, T]$:*

$$\sum_{x \in \mathcal{S}} \|u^i_\theta(x, t) - u^i(x, t)\|_2^2 \cdot p_t(x) \lesssim \epsilon^{\frac{2}{M}} M^{13},$$

*where the size of transformers depends on $\epsilon$ and $d_0$ is the transformer feature dimension.*

*Proof.* Please see Appendix E for a formal version and a detailed proof. □

## 5 VELOCITY AND DISTRIBUTION ESTIMATIONS

While Section 4 confirms that Transformers are powerful enough to approximate true velocity field with any precision, this section addresses the practical challenge of learning from data. We analyze the estimation error—the error that arises from having access only to a finite set of $n$ training samples rather than the true underlying data distribution. Specifically, Section 5.1 derives convergence rates for the velocity estimator, showing how its error decreases as the number of training samples $n$ increases. Then, by applying the error bounds for discrete flow matching Theorem 3.1, Section 5.2 translates this velocity error into a bound on the final distribution error under total variation distance.

### 5.1 VELOCITY ESTIMATION

We begin by establishing the estimation error bounds of training the factorized velocity estimator.

**Theorem 5.1** (Velocity Estimation with Discrete Flow Matching Transformer). *Let $\widehat{u}_\theta^{i_0} \in \mathcal{T}_R^{h,s,r}$ with parameter $\widehat{\Theta}^{i_0}$ be the factorized velocity estimator for coordinate $i_0 \in [d]$ under mixture path setting. Given $n$ i.i.d training samples $\{x_i\}_{i=1}^n$ from state space $\mathcal{S} = [M]^d$, we train the model by minimizing empirical loss $\widehat{\mathcal{L}}_{\mathrm{CDFM}}^{i_0}$ following (2.10). Then for large enough $n$ we have:*

$$\mathbb{E}_{\{x_i\}_{i=1}^n} [\mathcal{R}^{i_0}(\widehat{\Theta}^{i_0})] \lesssim M^{13d_0} n^{-\frac{1}{5Md_0}} (\log n)^{\frac{1}{5Md_0}},$$

*Here we set transformer feature dimension $d_0 > 25$ for simplicity.*

*Proof.* Please see Appendix F for a detailed proof. □

### 5.2 DISTRIBUTION ESTIMATION

The velocity estimation rate is a critical intermediate step. We now leverage this result to derive the main statistical guarantee of our work: an end-to-end bound on the final distribution generated by the discrete flow matching process. By combining the velocity estimation error bound from Theorem 5.1 with the internal error analysis for the discrete flow matching (Theorem 3.1), we establish the convergence rates for discrete flow matching distribution estimation error.

**Theorem 5.2** (Discrete Flow Matching Velocity Estimation with Transformer). *For any coordinate $i_0 \in [d]$, let $\widehat{u}_\theta^{i_0}$ be the $i$-th velocity estimator trained by minimizing empirical loss $\widehat{\mathcal{L}}_{\mathrm{CDFM}}^{i_0}$ following (2.10). Let $P$ denote the true distribution and $\widehat{P}$ the distribution generated by the discrete flow matching framework with factorized velocity estimators $\widehat{u}_\theta^1, \widehat{u}_\theta^2, \ldots, \widehat{u}_\theta^d$. Then for a vocabulary size $M$, the expected total variation distance $TV(P, \widehat{P})$ over training data $\{x_i\}_{i=1}^n$ is bounded by:*

$$\mathbb{E}_{\{x_i\}_{i=1}^n} [TV(P, \widehat{P})] \lesssim M^{7d_0} n^{-\frac{1}{9Md_0}} (\log n)^{\frac{1}{9Md_0}}.$$

*Here we set transformer feature dimension $d_0 > 25$ for simplicity.*

*Proof.* Please see Appendix G for a detailed proof. □

Theorem 5.2 establishes a concrete convergence rate, confirming that the model's generated distribution provably converges to the true data distribution as the size of the training set increases.

## 6 DISCUSSION AND LIMITATIONS

Our analysis also provides a strong theoretical justification for employing factorized velocities, a common practical choice. A comparison between the statistical rates for the factorized setting (Sections 4 and 5) and the general, non-factorized setting (Appendices H and I) reveals a critical insight. The intrinsic error bound for the general case scales with a term of $M^{d/2}$ (Theorem C.4), where $M$ is the vocabulary size and $d$ is the sequence length. In contrast, the intrinsic error bound for the factorized velocity discrete flow matching depends only on $\sqrt{M}$ (Theorem 3.1), mitigating this severe curse of dimensionality. Because this error is model-agnostic and intrinsic to the discrete

flow matching framework itself, the $\sqrt{M}$ term represents a fundamental barrier, not an artifact of a specific network architecture. This intrinsic weakness propagates to the final learning guarantees, resulting in looser estimation error bounds for the non-factorized approach (Theorem I.7 and Theorem I.8). This finding demonstrates that factorization is not only a computational convenience but is also crucial for statistical efficiency.

While our work provide solid statistical foundation for discrete flow matching, we highlight limitations and opens avenues for future research. A key limitation revealed by our analysis is the polynomial dependence of the error bounds on the vocabulary size $M$. As shown in our main theorems Theorem 5.2, the error bounds scale with terms like $M^{7d_0}$, and thus do not provide meaningful guarantees for typical large-vocabulary tasks such as text generation. This provides a critical insight: the discrete flow matching framework may be better suited for applications with small to medium sized vocabularies, such as coding or protein design. As part of our future work, we plan to investigate whether this polynomial dependence constitutes a fundamental hardness result.

## 7 CONCLUSION

In this work, we present the first comprehensive theoretical analysis of Discrete Flow Matching (DFM), providing an end-to-end guarantee that the generated distribution provably converges to the true data distribution. Our key innovation is establishing a model-agnostic, intrinsic error bound for the discrete flow matching (Theorem 3.1). This foundational result demonstrates that the final distribution error is controlled by the accuracy of the learned velocity field, a principle that holds true for discrete flow matching with any arbitrary implementation. Building on this intrinsic bound, we specialized our analysis to the popular case of Transformer-based models, decomposing the velocity risk into its two fundamental components: *approximation error*, which concerns the expressive power of the model architecture, and *estimation error*, which results from learning on a finite sample.

To address the approximation error, we first bridge the theoretical gap between the discrete data space $\mathcal{S}$ and our transformer universal approximation theory. We then construct an embedding that maps the discrete space into Euclidean space (Section 2). This embedding allows us to extend the discrete velocity field to a continuous one (Lemma E.2). By applying transformer universal approximation theory (Proposition B.17 and Theorem B.18) to the continuous extension of the velocity field, we obtain explicit approximation rates for discrete flow matching with Transformers (Theorem 4.7). To bound the estimation error, we analyze the complexity of the function class learned by the Transformer. By applying covering number arguments, we establish a precise rate for the velocity estimation error (Theorem 5.1). Finally, by composing these results, we derive the overall distribution estimation error in Theorem 5.2. This final bound characterizes the convergence rate of the learned distribution, providing a complete statistical analysis of the discrete flow matching Transformers pipeline.

In conclusion, this paper establishes a solid theoretical foundation for the discrete flow matching framework, with our intrinsic error bound serving as the cornerstone. By validating the core principles of discrete flow matching and the practical utility of techniques like velocity factorization, our work moves the understanding of these models from an empirical art to a more rigorous science, paving the way for more principled and robust advancements in discrete generative modeling.

**Related Work: Statistical Rates for Generative Models.** The theoretical study of generative models involves analyzing their statistical properties, including approximation error, estimation error, convergence rates, and sample complexity. Recent works make significant progress in this area for continuous data models. For instance, Fu et al. (2024) establish sharp statistical rates for conditional diffusion models with MLP backbones, while Jiao et al. (2024) derive explicit convergence rates for flow models in a latent space. In a key development for Flow Matching (FM), Fukumizu et al. (2024) show that Flow Matching achieve nearly minimax optimal convergence rates under the $p$-Wasserstein distance, providing the first theoretical evidence of its competitiveness with diffusion models. This line of studies extend to analyze conditional diffusion transformers (Hu et al., 2024b) and higher-order flow matching methods (Su et al., 2025).

However, these theoretical work only focus on models for continuous data, leaving the discrete setting unexplored. A fundamental challenge for discrete generative approaches is how to define a tractable denoising process on discrete spaces. Our key innovation is to overcome this obstacle by

constructing an embedding layer that maps discrete data into a continuous space. This crucial step enables a rigorous statistical analysis of discrete generative models within continuous-time framework. Building on this, we provide an end-to-end statistical guarantee for discrete flow matching.

We defer an extended discussion on related work to Appendix A due to page limits.

## ETHIC STATEMENT

This paper does not involve human subjects, personally identifiable data, or sensitive applications. We do not foresee direct ethical risks. We follow the ICLR Code of Ethics and affirm that all aspects of this research comply with the principles of fairness, transparency, and integrity.

## REPRODUCIBILITY STATEMENT

We ensure reproducibility of our theoretical results by including all formal assumptions, definitions, and complete proofs in the appendix. The main text states each theorem clearly and refers to the detailed proofs. No external data or software is required.

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

## A  RELATED WORK

In this section, we discuss the recent success of DFM and techniques used in our work.

**Discrete Generative Models and Discrete Flow Matching.**    While autoregressive models remain the predominant paradigm for discrete data generation (Achiam et al., 2023; Liu et al., 2024; Ingraham et al., 2019), recent diffusion and flow-matching alternatives show impressive performance across many domains, including sound generation (Yang et al., 2023), graph generation (Vignac et al., 2022), and protein design (Campbell et al., 2024). Progress in adapting these continuous-time models to discrete settings follows two strategies. The first involves designing diffusion processes over discrete state spaces (Sohl-Dickstein et al., 2015; Hoogeboom et al., 2021; Austin et al., 2021; Lou et al., 2023; Yang et al., 2023; Vignac et al., 2022). The second embeds discrete data into a continuous space, where standard diffusion or flow-matching techniques then be applied (Dieleman et al., 2022; Campbell et al., 2022; Davis et al., 2024).

Most recently, Campbell et al. (2024) and Gat et al. (2024) introduce Discrete Flow Matching (DFM), which emerge as a powerful new paradigm for discrete generative modeling. DFM offers significant flexibility in the design of the denoising process and the choice of the source distribution. Consequently, there is a growing interest in exploring the efficiency and application of DFM for various generation tasks. This interest lead to a rapid expansion of DFM-based models. For instance, Hu & Ommer (2024) validate its efficiency in the image domain. In graph generation, Qin et al. (2024) introduce DeFoG, a framework that uses DFM to respect the inherent symmetries of graphs and disentangle sampling from training for more efficient optimization. Fuest et al. (2025) introduce MaskFlow, a unified video generation framework that leverages DFM for efficient, high-quality long video synthesis. Similarly, in structural biology, Yi et al. (2025) present ADFLIP, a DFM-based model for designing protein sequences conditioned on all-atom structural contexts. However, the success of these models are driven by empirical validation. Despite their impressive performance and growing adoption, a rigorous theoretical understanding of DFM is lacking. Our work fills this critical gap by providing the solid theoretical foundations for Discrete Flow Matching.

A closely related line of work investigates discrete diffusion models. Austin et al. (2021) introduce Discrete Diffusion Models, and extend diffusion models to discrete state spaces. Subsequent work develops masked discrete diffusion frameworks and demonstrates their effectiveness for graph generalization (Shi et al., 2025) and language modeling (Sahoo et al., 2024). There is also a growing literature on the theoretical foundations of discrete diffusion models. For instance, Chen & Ying (2024) analyze the theoretical properties of the discrete diffusion model and Zhang et al. (2025) investigate the convergence result of discrete diffusion model under a certain sampling algorithm. However, previous theoretical analyses rely on cross-entropy loss function used in discrete diffusion model. As clarified by Lipman et al. (2024), the standard diffusion training objective can be interpreted as training a particular flow-matching model with $x_0$–prediction under an appropriate time reparameterization. In particular, masked discrete diffusion model introduced in (Austin et al., 2021) can be viewed as a special case of mixture path model after an appropriate time reparameterization, with source distribution $P_0 \sim \delta_{(M,M,...,M)}$ and cross-entropy loss function. Analysis and techniques developed in our work extend to a broader class of loss functions, beyond the MSE considered in this paper and the cross-entropy commonly used in discrete diffusion models. Our results thus provide a general framework for the theoretical analysis of discrete flow matching, accommodating a wider range of loss functions tailored to different tasks.

**Transformer Universal Approximation.**    Transformers are universal approximators, possessing the capacity to model any arbitrary sequence-to-sequence function with a desired level of precision. Yun et al. (2019) establish universality for deep stacks of self-attention and feed-forward layers via a contextual mapping method, under the assumption that hidden representations remain sufficiently separated. Later, (Alberti et al., 2023) broaden the scope of this guarantee to encompass variants using sparse attention mechanisms. Building on this foundation, more recent findings relax the architectural requirements. Research from Hu et al. (2024a); Kajitsuka & Sato (2023) demonstrate that the powerful approximation capability is not dependent on depth, showing that a single Transformer block with one self-attention layer is itself sufficient to achieve universal approximation. In our work, we leverage this powerful result to analyze the approximation error of Transformer-based discrete flow matching models.

# B SUPPLEMENTARY BACKGROUND: TRANSFORMER BLOCK

In this section, we introduce the transformer network structure (Vaswani et al., 2017) and its properties. Following the notations in (Hu et al., 2024b), We start with the definition of transformers.

## B.1 TRANSFORMERS

**Transformer Block.** Let $h$ be the number of heads and $s$ be the hidden dimension of the multi-head attention layer. The multi-head attention layer $F^{\mathrm{SA}} : \mathbb{R}^{d \times L} \to \mathbb{R}^{d \times L}$ is then defined as:

$$F^{\mathrm{SA}}(Z) := Z + \sum_{i=1}^{h} W_O^i(W_V^i Z)\,\mathsf{Softmax}((W_K^i Z)^\top (W_Q^i Z)),$$

where $W_K^i, W_Q^i, W_V^i, (W_O^i)^\top \in \mathbb{R}^{s \times d}$ are weight matrices for all $i \in [h]$ and $\mathsf{Softmax}(\cdot)$ is the column-wise softmax function.

Let $r$ be the dimension of hidden of the feed-forward layer. The feed-forward layer $F^{\mathrm{FF}}(Z) : \mathbb{R}^{d \times L} \to \mathbb{R}^{d \times L}$ is then defined as:

$$F^{\mathrm{FF}}(Z) := Z + W_2\,\mathsf{ReLU}(W_1 Z + b_1 \mathbb{1}_L^\top) + b_2 \mathbb{1}_L^\top,$$

where $W_1, (W_2)^\top \in \mathbb{R}^{r \times d}$ are weight matrices, $b_1 \in \mathbb{R}^r, b_2 \in \mathbb{R}^d$ are bias. Throughout this paper, we treat $\mathsf{ReLU}(\cdot)$ as element-wise operation when applied to vectors or matrices.

We define a transformer block as the composition of a self-attention layer and a feed-forward layer.

**Definition B.1** (Transformer Block). *For $h, s, r \in \mathbb{Z}^+$, we define a transformer block $F^{h,s,r} : \mathbb{R}^{d \times L} \to \mathbb{R}^{d \times L}$ as:*

$$F^{h,s,r} := F^{\mathrm{FF}} \circ F^{\mathrm{SA}},$$

*where $F^{\mathrm{SA}}$ has $h$ heads and hidden dimension $s$, $F^{\mathrm{FF}}$ has hidden dimension $r$.*

Then we define the transformer networks function class as composition of transformer blocks.

**Definition B.2** (Transformer Network Function Class). *Let transformer block $F^{h,s,r}$ be as defined in Definition B.1. Then we define the transformer network function class $\mathcal{T}^{h,s,r}$ as a function class with each component being the composition of transformer blocks:*

$$\mathcal{T}^{h,s,r} = \{\tau : \mathbb{R}^{d \times L} \to \mathbb{R}^{d \times L} \mid \tau = F^{h,s,r} \circ \cdots \circ F^{h,s,r}\}.$$

**Discrete Flow Matching Transformers.** Following the common structure of diffusion transformers (Peebles & Xie, 2023) and flow matching transformers (Su et al., 2025), we introduce the transformer architecture used in this paper. We start with the definition of a reshape layer that converts a vector input $x \in \mathbb{R}^{d_x}$ into a matrix input $Z \in \mathbb{R}^{d \times L}$, where $d_x = d \times L$.

**Definition B.3** (Reshape Layer). *The reshape layer $R(\cdot) : \mathbb{R}^{d_x} \to \mathbb{R}^{d \times L}$ is an operator transforming vector input of dimension $d_x$ to matrix output of size $d \times L$. Te reshape layer is frozen when training. Further, we define the reverse reshape layer as $R^{-1}(\cdot) : \mathbb{R}^{d \times L} \to \mathbb{R}^{d_x}$.*

For instance, the most commonly used reshape layer in diffusion models is the operator turning vector input of dimension $d_x$ to matrix input of size $d \times L$ by rearranging entries, where $d_x = d \cdot L$.

Finally, we define the following transformer function class with reshape layer.

**Definition B.4** (Transformer Function Class With Reshape Layer and Parameter Bound). *Let $F^{\mathrm{E}}(Z) := Z + E$ represent the position encoding layer and $R$ represent the reshape layer. The transformer network class with reshape layer is defined as:*

$$\mathcal{T}_R^{h,s,r} := \{R^{-1} \circ f_{\mathcal{T}} \circ F^{\mathrm{E}} \circ R : \mathbb{R}^{d_0} \to \mathbb{R}^{d_0} \mid f_{\mathcal{T}} \in \mathcal{T}^{h,s,r}\}.$$

*We write $W_{KQ}^i := (W_K^i)^\top W_Q^i$ and $W_{OV}^i := W_O^i W_V^i$ for simplicity of notations. Then, a transformer function class with reshape layer and parameter bound is defined as as $\mathcal{T}_R^{h,s,r}(C_{\mathcal{T}}, C_{KQ}^{2,\infty}, C_{KQ}, C_{OV}^{2,\infty}, C_{OV}, C_E, C_F^{2,\infty}, C_F, L_{\mathcal{T}})$, which satisfies:*

- $h, s, r$ as defined above;

- Transformer output bound: $\sup_Z \|f_{\mathcal{T}}(Z)\| \leq C_{\mathcal{T}}$;

- Parameter bound in $F^{\mathrm{FF}}$: $\max\{\|W_1\|_{2,\infty}, \|W_2\|_{2,\infty}\} \leq C_F^{2,\infty}$, $\max\{\|W_1\|_2, \|W_2\|_2\} \leq C_F^2$;

- Parameter bound in $F^{\mathrm{SA}}$: $\|W_{KQ}^i\|_2 \leq C_{KQ}$, $\|W_{OV}^i\|_2 \leq C_{OV}$, $\|W_{KQ}^i\|_{2,\infty} \leq C_{KQ}^{2,\infty}$, $\|W_{OV}^i\|_2 \leq C_{OV}^{2,\infty}$;

- Parameter bound in $F^{\mathrm{E}}$: $\|E^\top\|_{2,\infty} \leq C_E$;

- Frobenius Lipschitzness of $f_{\mathcal{T}} \in \mathcal{T}^{h,s,r}$: $\|f_{\mathcal{T}}(Z_1) - f_{\mathcal{T}}(Z_2)\|_F \leq L_{\mathcal{T}}\|Z_1 - Z_2\|_F$.

### B.2 LIPSCHITZNESS OF TRANSFORMER NETWORK

To prepare our proofs, we first establish a new result on the Lipschitzness of transformer networks (i.e., Lemma B.11). It shows that a function composed of Lipschitz functions remains Lipschitz.

**Preparations of Lemma B.11.** We first present some helper lemmas for proving Lemma B.11.

**Lemma B.5.** Let $f_1, f_2 : \mathbb{R}^{d \times L} \to \mathbb{R}^{d \times L}$ be $L_1$- and $L_2$-Lipschitz w.r.t. Frobenius norm $\|\cdot\|_F$ respectively. Then $f_1 \circ f_2$ is $(L_1 L_2)$-Lipschitz with respect to $\|\cdot\|_F$.

*Proof.* For all $X_1, X_2 \in \mathbb{R}^{d \times L}$, it holds:

$$\|f_1 \circ f_2(X_1) - f_1 \circ f_2(X_2)\|_F \leq L_1\|f_2(X_1) - f_2(X_2)\|_F$$
$$\leq L_1 L_2\|X_1 - X_2\|_F,$$

where the first line is by $\|f_1(X_1) - f_1(X_2)\|_F \leq L_1\|X_1 - X_2\|_F$ and the second line is by $\|f_2(X_1) - f_2(X_2)\|_F \leq L_2\|X_1 - X_2\|_F$. This completes the proof. □

Next, we analyze the Lipschitzness of RELU function. Throughout this paper, we treat $\mathrm{ReLU}(\cdot)$ as element-wise operator when applied to vectors or matrices.

**Lemma B.6** (Lipschitzness of $\mathrm{ReLU}(\cdot)$). *The ReLU function $\mathrm{ReLU} : \mathbb{R}^{d \times L} \to \mathbb{R}^{d \times L}$ is 1-Lipschitz with respect to the Frobenius norm $\|\cdot\|_F$.*

*Proof.* For all $X_1, X_2 \in \mathbb{R}^{d \times L}$, it holds:

$$\|\mathrm{ReLU}(X_1) - \mathrm{ReLU}(X_2)\|_F = \sqrt{\sum_{i=1}^d \sum_{j=1}^L |\mathrm{ReLU}((X_1)_{i,j}) - \mathrm{ReLU}((X_2)_{i,j})|^2}$$

(By the definition of Frobenius Norm)

$$\leq \sqrt{\sum_{i=1}^d \sum_{j=1}^L |(X_1)_{i,j} - (X_2)_{i,j}|^2}$$
$$= \|X_1 - X_2\|_F,$$

where the second line is by $|\mathrm{ReLU}(x_1) - \mathrm{ReLU}(x_2)| \leq |x_1 - x_2|$ for $x_1, x_2 \in \mathbb{R}$.

This completes the proof. □

With Lemma B.6, we now prove the Lipschitzness of feed-forward layer.

**Lemma B.7** (Lipschitzness of FFN). *Let $Z \in \mathbb{R}^{d \times L}$ and define the feedforward layer as*

$$F^{\mathrm{FF}}(Z) := Z + W_2 \mathrm{ReLU}(W_1 Z + b_1 \mathbb{1}_L^\top) + b_2 \mathbb{1}_L^\top.$$

*Then $F^{\mathrm{FF}}$ is Lipschitz continuous with respect to Frobenius norm, with Lipschitz constant $\|W_1\|_2 \cdot \|W_2\|_2 + 1$.*

*Proof.* For all $X_1, X_2 \in \mathbb{R}^{d \times L}$, it holds:

$$\|F^{\mathrm{FF}}(X_1) - F^{\mathrm{FF}}(X_2)\|_F$$

$$\leq \|X_1 - X_2\|_F + \|(W_2 \operatorname{ReLU}(W_1 X_1 + b_1 \mathbb{1}_L^\top) + b_2 \mathbb{1}_L^\top) - (W_2 \operatorname{ReLU}(W_1 X_2 + b_1 \mathbb{1}_L^\top) + b_2 \mathbb{1}_L^\top)\|_F$$

(By triangle inequality)

$$\leq \|X_1 - X_2\|_F + \|W_2\|_2 \cdot \|\operatorname{ReLU}(W_1 X_1 + b_1 \mathbb{1}_L^\top) - \operatorname{ReLU}(W_1 X_2 + b_1 \mathbb{1}_L^\top)\|_F$$

$$\leq \|X_1 - X_2\|_F + \|W_2\|_2 \cdot \|(W_1 X_1 + b_1 \mathbb{1}_L^\top) - (W_1 X_2 + b_1 \mathbb{1}_L^\top)\|_F$$

$$\leq \|X_1 - X_2\|_F + \|W_1\|_2 \cdot \|W_2\|_2 \cdot \|X_1 - X_2\|_F$$

$$= (\|W_1\|_2 \cdot \|W_2\|_2 + 1) \cdot \|X_1 - X_2\|_F,$$

where the third line is by $\|AX\|_F \leq \|A\|_2 \cdot \|X\|_F$, the fourth line is by Lemma B.6, and the fifth line is by $\|AX\|_F \leq \|A\|_2 \|X\|_F$.

This completes the proof. $\square$

Next, we establish the Lipschitzness of self-attention. We remark that Castin et al. (2024) also establish similar results but with a different method not applicable in our setting. We start with a lemma proving Lipschitzness of any function whose Jacobian norm is uniformly bounded.

**Lemma B.8** (Lipschitzness of Functions with Bounded Jacobian; Modified from Lemma A.6 of (Edelman et al., 2022)). *Let $\Delta^{n-1} = \{x \in \mathbb{R}^n | x \geq 0, \|x\|_1 = 1\}$ denote the $n$-simplex. Suppose $f : \mathbb{R}^d \to \Delta^{n-1}$ belongs to $C^1$ and satisfies $\|Jf(x)\|_2 \leq c_f$ for all $x \in \mathbb{R}^d$. Then for all $x_1, x_2 \in \mathbb{R}^d$, it holds:*

$$\|f(x_1) - f(x_2)\|_2 \leq c_f \|x_1 - x_2\|_2.$$

*Proof.* Our proof follows the proof of (Edelman et al., 2022, Lemma A.6). With Newton-Leibniz formula and change of variables, we have

$$\|f(x_1) - f(x_2)\|_2 \leq \|(\int_0^1 J(tx_1 + (1-t)x_2)\mathrm{d}t)(x_1 - x_2)\|_2$$

$$\leq \int_0^1 \|J(tx_1 + (1-t)x_2)(x_1 - x_2)\|_2 \mathrm{d}t \qquad \text{(By Jensen's inequality)}$$

$$\leq \int_0^1 \|J(tx_1 + (1-t)x_2)\|_2 \cdot \|x_1 - x_2\|_2 \mathrm{d}t \qquad (\|Ax\|_2 \leq \|A\|_2 \|x\|_2)$$

$$\leq c_f \|x_1 - x_2\|_2. \qquad (\|Jf(x)\|_2 \leq c_f)$$

This completes the proof. $\square$

With Lemma B.8, we now prove the Lipschitzness of $\operatorname{Softmax}(\cdot)$.

**Lemma B.9** (Lipschitzness of $\operatorname{Softmax}(\cdot)$; Modified from Corollary A.7 of (Edelman et al., 2022)). *Let $\operatorname{Softmax}(\cdot) : \mathbb{R}^{d \times L} \to \mathbb{R}^{d \times L}$ denote the column-wise softmax function. Then for all $X_1, X_2 \in \mathbb{R}^{d \times L}$, it holds:*

$$\|\operatorname{Softmax}(X_1) - \operatorname{Softmax}(X_2)\|_F \leq \|X_1 - X_2\|_F.$$

*Proof.* For the simplicity of presentation, let $s : \mathbb{R}^d \to \mathbb{R}^d$ denote the (element-wise) softmax function. Its Jacobian is $J = \operatorname{diag}(s) - ss^\top$. Thus, $J$ is symmetric and positive semi-definite, so its singular values equal its eigenvalues. Recall that for softmax function, its Jacobian matrix's eigenvalues are smaller than 1. Hence,

$$\|J\|_2 \leq 1.$$

Combining $\|J\|_2 \leq 1$ with Lemma B.8, for each column $i \in [L]$ we have

$$\|(\operatorname{Softmax}(X_1) - \operatorname{Softmax}(X_2))_{\cdot,i}\|_2 \leq \|(X_1 - X_2)_{\cdot,i}\|_2.$$

Summing over all columns gives

$$\|\operatorname{Softmax}(X_1) - \operatorname{Softmax}(X_2)\|_F \leq \|X_1 - X_2\|_F.$$

This completes the proof. $\square$

Finally, we prove the Lipschitzness of self-attention layer.

**Lemma B.10** (Lipschitzness of Multi-Head Self-Attention). *Let $X \in \mathbb{R}^{d \times L}$ satisfy $\|X\|_2 \leq B_X$ on a compact domain. Define $F^{\mathrm{SA}}(X) := X + \sum_{i=1}^{h} W_O^i (W_V^i X) \, \mathsf{Softmax}((W_K^i X)^\top (W_Q^i X))$. Then, $F^{\mathrm{SA}}$ is Lipschitz continuous w.r.t. the Frobenius norm $\|\cdot\|_F$, with Lipschitz constant*

$$1 + 2(B_X)^2 \sum_{i=1}^{h} \|W_{OV}^i\|_2 \cdot \|W_{KQ}^i\|_2 + L \sum_{i=1}^{h} \|W_{OV}^i\|_2,$$

*where $W_{OV}^i = W_O^i W_V^i$ and $W_{KQ}^i = (W_K^i)^\top W_Q^i$ for any $i \in [h]$.*

*Proof.* For $\|X\|_2 \leq B_X$, the quadratic form $X^\top A X$ is Lipschitz continuous with Lipschitz constant w.r.t. the Frobenius norm $\|\cdot\|_F$, with Lipschitz constant $2\|A\|_2 B_X$. Therefore, by Lemma B.5 and Lemma B.9, function $\mathsf{Softmax}((W_K^i X)^\top (W_Q^i X))$ is Lipschitz continuous with Lipschitz constant w.r.t. the Frobenius norm $\|\cdot\|_F$, with Lipschitz constant $2\|W_{KQ}^i\|_2 B_X$.

Then, for every $X_1, X_2 \in \mathbb{R}^{d \times L}$ such that $\|X_1\|_2, \|X_2\|_2 \leq B_X$, it holds:

$$\|F^{\mathrm{SA}}(X_1) - F^{\mathrm{SA}}(X_2)\|_F$$

$$\leq \|X_1 - X_2\|_F + \|\sum_{i=1}^{h} W_{OV}^i X_1 \, \mathsf{Softmax}(X_1^\top W_{KQ}^i X_1) - W_{OV}^i X_2 \, \mathsf{Softmax}(X_2^\top W_{KQ}^i X_2)\|_F$$

$$\leq \|X_1 - X_2\|_F + \sum_{i=1}^{h} \|W_{OV}^i\|_2 \cdot \|X_1 \, \mathsf{Softmax}(X_1^\top W_{KQ}^i X_1) - X_2 \, \mathsf{Softmax}(X_2^\top W_{KQ}^i X_2)\|_F$$

$$\hspace{8cm} (\|AX\|_F \leq \|A\|_2 \|X\|_F)$$

$$\leq \|X_1 - X_2\|_F + \sum_{i=1}^{h} \|W_{OV}^i\|_2 \cdot \|X_1 \, \mathsf{Softmax}(X_1^\top W_{KQ}^i X_1) - X_1 \, \mathsf{Softmax}(X_2^\top W_{KQ}^i X_2)\|_F$$

$$+ \sum_{i=1}^{h} \|W_{OV}^i\|_2 \cdot \|X_1 \, \mathsf{Softmax}(X_2^\top W_{KQ}^i X_2) - X_2 \, \mathsf{Softmax}(X_2^\top W_{KQ}^i X_2)\|_F$$

$$\leq \|X_1 - X_2\|_F + \sum_{i=1}^{h} \|W_{OV}^i\|_2 \cdot B_X \cdot \| \mathsf{Softmax}(X_1^\top W_{KQ}^i X_1) - \mathsf{Softmax}(X_2^\top W_{KQ}^i X_2)\|_F$$

$$+ \sum_{i=1}^{h} \|W_{OV}^i\|_2 \cdot \|X_1 - X_2\|_F \cdot \| \mathsf{Softmax}(X_2^\top W_{KQ}^i X_2)\|_F$$

$$\hspace{6cm} (\|X_1\|_2 \leq B_X \text{ and } \|AX\|_F \leq \|A\|_F \cdot \|X\|_F)$$

$$\leq \|X_1 - X_2\|_F + \sum_{i=1}^{h} \|W_{OV}^i\|_2 \cdot B_X \cdot 2 W_{KQ}^i B_X \cdot \|X_1 - X_2\|_F + \sum_{i=1}^{h} \|W_{OV}^i\|_2 \cdot \|X_1 - X_2\|_F \cdot L$$

$$\hspace{6cm} (\text{By } \| \mathsf{Softmax}(Z)\|_F \leq L \text{ when } Z \in \mathbb{R}^{L \times L})$$

$$= (1 + 2(B_X)^2 \sum_{i=1}^{h} \|W_{OV}^i\|_2 \cdot \|W_{KQ}^i\|_2 + L \sum_{i=1}^{h} \|W_{OV}^i\|_2) \cdot \|X_1 - X_2\|_F. \quad (\|X_1\|_2, \|X_2\|_2 \leq B_X)$$

This completes the proof. $\qquad\square$

**Lipschitzness of Transformer.** Now we state our result on the Lipschitzness of transformer.

**Lemma B.11** (Lipschitzness of Transformer Block). *Let*

$$f_{\mathcal{T}} := F_1^{\mathrm{FF}} \circ F^{\mathrm{SA}} \circ F_2^{\mathrm{FF}},$$

*be a Transformer block in the class*

$$\mathcal{T}^{h,s,r}(C_{\mathcal{T}}, C_{KQ}^{2,\infty}, C_{KQ}, C_{OV}^{2,\infty}, C_{OV}, C_E, C_F^{2,\infty}, C_F).$$

*If $X \in \mathbb{R}^{d \times L}$ satisfies $\|X\| \leq B_X$, then $f_{\mathcal{T}}$ is Lipschitz continuous w.r.t. the Frobenius norm $\|\cdot\|_F$ with Lipschitz constant*

$$L_{\mathcal{T}} \leq (1 + 2h(B_X)^2 C_{OV} C_{KQ} + hL C_{OV}) \cdot (C_F^2 + 1)^2.$$

*Proof.* This is a direct consequence of combining Lemmas B.5, B.7 and B.10. □

**Remark B.12** (Near-Optimality of the Lipschitz Bound). *We remark that the upper bound on the Lipschitz constant in Lemma B.11 is near-optimal in its dependence on the key parameters. To illustrate this, we construct a worst-case example as follows. Let $A \in \mathbb{R}^{d \times L}$ be the diagonal matrix*

$$A_{ij} = \begin{cases} 1, & i = j, \\ 0, & i \neq j. \end{cases}$$

*Consider the network $f_{\mathcal{T}}$ with parameter*

$$W_{OV}^i = C_{OV} I_d, \quad W_{KQ}^i = C_{KQ} I_d, \quad W_1 = W_2 = C_F A,$$

*where $I_d$ is the $d \times d$ identity.*

*A direct calculation shows that the operator norm of the directional derivative of $f_{\mathcal{T}}$ at an input $Z$ in direction $V$ scales with the same order as the upper bound in Lemma B.11, up to constants and polynomial factors in $d$. Hence the bound in Lemma B.11 provides a tight estimate by construction.*

### B.3 UNIVERSAL APPROXIMATION OF TRANSFORMERS

Previous works (Hu et al., 2024b; Kajitsuka & Sato, 2023; Yun et al., 2019) study the universal approximation property of Transformers for continuous functions. In this section, we restate the proofs for completeness and adapt the parameter estimates to the discrete flow-matching framework. This adaptation highlights the connection between universal approximation and our discrete setting.

**Background: Contextual Mapping.** Concept of contextual mapping is key to the proof of universal approximation of transformer. We restate the definition of contextual mapping and related concepts introduced by Kajitsuka & Sato (2023) for completeness. To start with, we introduce the concept of Vocabulary. We use $Z_{:,k}$ to denote the $k$-th column of vector $Z$.

**Definition B.13** (Vocabulary). *Let $Z \in \mathbb{R}^{d \times L}$ represent input embeddings. Specifically, given $N$ embeddings $Z^{(1)}, \ldots, Z^{(N)} \in \mathbb{R}^{d \times L}$, we call $Z^{(i)}$ the $i$-th sequence for $i \in [N]$. Further, we define the $i$-th vocabulary set as $\mathcal{V}^{(i)} = \cup_{k \in [L]} Z_{:,k}^{(i)} \subset \mathbb{R}^d$. Then the whole vocabulary set $\mathcal{V}$ is defined as $\mathcal{V} = \cup_{i \in [N]} \mathcal{V}^{(i)} \in \mathbb{R}^d$.*

We assume embeddings are separate. Specifically, we assume embeddings are $(\gamma_{\min}, \gamma_{\max}, \delta)$-separated defined below.

**Definition B.14** (Tokenwise Separateness). *Let $Z^{(1)}, \ldots, Z^{(N)} \in \mathbb{R}^{d \times L}$ be embeddings. Then $Z^{(1)}, \ldots, Z^{(N)}$ are called tokenwise $(\gamma_{\min}, \gamma_{\max}, \delta)$-separated if the following conditions hold:*

*(i) For any $i \in [N]$ and $k \in [L]$, $\|Z_{:,k}^{(i)}\| > \gamma_{\min}$ holds.*

*(ii) For any $i \in [N]$ and $k \in [L]$, $\|Z_{:,k}^{(i)}\| < \gamma_{\max}$ holds.*

*(iii) For any $i, j \in [N]$ and $k, m \in [L]$, if $Z_{:,k}^{(i)} \neq Z_{:,m}^{(j)}$, then $\|Z_{:,k}^{(i)} - Z_{:,m}^{(j)}\| > \delta$ holds.*

*Further, $Z^{(1)}, \ldots, Z^{(N)}$ are called tokenwise $(\gamma, \delta)$-separated if only (ii) and (iii) hold. Also, $Z^{(1)}, \ldots, Z^{(N)}$ are called tokenwise $\delta$-separated if only (iii) holds.*

Building on the condition (ii) and (iii) in Definition B.14, we introduce the concept of contextual mapping. Contextual mapping describes attention layer's ability to distinguish difference and relationship between tokens in different input sequences.

**Definition B.15** (Contextual Mapping). *Let $Z^{(1)}, \ldots, Z^{(N)} \in \mathbb{R}^{d \times L}$ be embeddings. Then, we say a map $f : \mathbb{R}^{d \times L} \to \mathbb{R}^{d \times L}$ is a $(\gamma, \delta)$-contextual mapping if the following conditions hold:*

*(i) For For any $i \in [N]$ and $k \in [L]$, $\|f(Z^{(i)})_{:,k}\| < \gamma$ holds.*

*(ii) For any $i, j \in [N]$ and $k, m \in [L]$, if $\mathcal{V}^{(i)} \neq \mathcal{V}^{(j)}$ or $Z_{:,k}^{(i)} \neq Z_{:,m}^{(j)}$, then $\|f(Z^{(i)})_{:,k} - f(Z^{(j)})_{:,m}\| > \delta$ holds.*

**Helper Lemma.** We restate a lemma from (Hu et al., 2024a). This lemma guarantee the existence of 1-layer single head attention that is $(\gamma, \delta)$-contextual mapping.

**Lemma B.16** (Any-Rank Attention is $(\gamma, \delta)$-Contextual Mapping, Lemma 2.2 of (Hu et al., 2024a))**.** *Let $Z^{(1)}, \ldots, Z^{(N)} \in \mathbb{R}^{d \times L}$ be tokenwise $(\gamma_{\min}, \gamma_{\max}, \epsilon)$-separated embeddings with the vocabulary set $\mathcal{V} = \cup_{i \in [N]} \mathcal{V}^{(i)} \subset \mathbb{R}^d$. Assume there are no duplicate token in each sequence; that is, $Z^{(i)}_{:,k} \neq Z^{(i)}_{:,m}$ for $i \in [N]$ and $k, m \in [L]$. Then there exists a 1-layer single head attention layer that is a $(\gamma, \delta)$-contextual mapping for the embeddings $Z^{(1)}, \ldots, Z^{(N)}$ with*

$$\gamma = \gamma_{\max} + \frac{\epsilon}{4}, \ \delta = \exp\left(-\frac{5|\mathcal{V}|^4 d \kappa \gamma_{\max} \log L}{\epsilon}\right),$$

*where $\kappa := \frac{\gamma_{\max}}{\gamma_{\min}}$.*

*Proof.* See the proof of (Hu et al., 2024a, Lemma 2.2). $\square$

**Universal Approximation of Transformer.** We introduce the universal approximation theory of transformer in (Su et al., 2025) and restate the proof for completeness.

**Proposition B.17** (Transformer Universal Approximation, Theorem H.2 of (Su et al., 2025))**.** *Let $\epsilon \in (0, 1)$ and $p \in [1, \infty)$. Let $Z \in [-I, I]^{d \times L}$ be an input sequence on a bounder domain, where $I > 0$. Let $f(Z) : [-I, I]^{d \times L} \to \mathbb{R}^{d \times L}$ be a continuous function on a bounded domain. Then there exists a $g(Z) = F_1^{\mathrm{FF}} \circ F^{\mathrm{SA}} \circ F_2^{\mathrm{FF}} \in \mathcal{T}^{h,s,r}$ such that $d_F(f(Z), g(Z)) < \epsilon$, where $d_F := (\int \|f(Z) - g(Z)\|_F^2 dZ)^{\frac{1}{2}}$.*

*Proof.* Here, we restate the proof of (Su et al., 2025, Theorem H.2) for completeness.

The proof proceeds in four parts:

- Step 1: Approximation using step function

- Step 2: Quantization by the first feed-forward layer

- Step 3: Contextual mapping through the self-attention layer

- Step 4: Memorization via the second feed-forward layer

**Step 1: Approximation using Step Function.** We assume the domain of $f$ is $\Omega = [-I, I]^{d \times L}$.

Then we construct the grid $\mathbb{G}_D$ as :

$$\mathbb{G}_D := \{C \in \Omega | C_{i,k} = -I + \frac{s_{i,k}}{D}, s_{i,k} = 1, \ldots, 2ID, \}$$

where $D > 0$ is the grid granularity. Given $Z \in \Omega$, we approximate $f$ via the step function

$$g_1(Z) = \sum_{C \in \mathbb{G}_D} f(C) \mathbb{1}\{Z \in C + [-1/D, 0)^{d \times L}\}.$$

By uniform continuity of $f$, there exists $D$ such that

$$d_F(f, g_1) < \frac{\epsilon}{3}.$$

Then we use a transformer to approximate the step function $g_1(Z)$.

**Step 2: Quantization by the First Feed-forward Layer.** The quantification function we want to approximate consists of two parts, namely, the quantize function and the penalty function. We approximate two parts separately.

- **Quantize Function** We define the quantization function $\mathrm{quant}_D : \mathbb{R} \to \mathbb{R}$:

$$\mathrm{quant}_D(z) := \begin{cases} -I & z < -I, \\ -I + 1/D & -I \leq z < -I + 1/D, \\ \vdots & \vdots \\ I & I - 1/D \leq z. \end{cases}$$

Further, we define the quantize function $\text{quant}_D^{d \times L}(Z) : \mathbb{R}^{d \times L} \to \mathbb{R}^{d \times L}$ as the entrywise quantize function, such that $(\text{quant}_D^{d \times L}(Z))_{t,k} = \text{quant}_D(Z_{t,k})$. Notice that $\text{quant}_D(z)$ is approximated through the following function by taking sufficiently small $\delta$:

$$f_1(z) := -I + \sum_{t=-ID}^{I(D-1)} \frac{\text{RELU}[z/\delta - t/\delta D] - \text{RELU}[z/\delta - 1 - t/\delta D]}{D}. \tag{B.1}$$

That's to say, there exists RELU feed-forward network approximating $\text{quant}_D^{d \times L}(Z)$.

- **Penalty Function** We define the penalty function $\text{penalty} : \mathbb{R} \to \mathbb{R}$:

$$\text{penalty}(z) := \begin{cases} -1 & z < -I, \\ 0 & z \in [-I, I], \\ 1 & z > I. \end{cases}$$

Further, we define the penalty function $\text{penalty}^{d \times L}(Z) : \mathbb{R}^{d \times L} \to \mathbb{R}^{d \times L}$ as the entrywise penalty function, such that $(\text{penalty}^{d \times L}(Z))_{t,k} = \text{quant}_D(Z_{t,k})$. Notice that $\text{penalty}(z)$ is approximated through the following function by taking sufficiently small $\delta$:

$$\begin{aligned} f_2(z) = \;& \text{ReLU}[(z - I)/\delta] - \text{ReLU}[(z - I)/\delta + 1] \\ & + \text{ReLU}[(-z - I)/\delta] - \text{ReLU}[(-z - I)/\delta + 1] \end{aligned} \tag{B.2}$$

That's to say, there exists RELU feed-forward network approximating $\text{penalty}^{d \times L}(Z)$.

Altogether, we define $g_2(Z) : \mathbb{R}^{d \times L} \to \mathbb{R}^{d \times L}$ as :

$$g_2(Z) := \frac{\text{quant}_D^{d \times L}(Z) + I}{2I} + \text{penalty}^{d \times L}(Z).$$

$g_2(Z)$ map $[-I, I]^{d \times L}$ into normalized grid $\mathbb{G}_D^{\text{norm}} \subset [0, 1]^{d \times L}$ with gride granularity $2ID$. At the same time, $g_2(Z)$ guarantees non-positive outputs on domain $\mathbb{R}^{d \times L} \setminus [-I, I]^{d \times L}$. We use $f_1(z)$ and $f_2(z)$ introduced above to construct the first feed-forward layer $F_1^{\text{FF}}$.

**Step 3: Contextual Mapping through the Self-attention Layer.** Let $\bar{\mathbb{G}}_D$ denote the following sub-grid class on $[0, 1]^{d \times L}$:

$$\bar{\mathbb{G}}_D := \{G \in \mathbb{G}_D^{\text{norm}} \mid \text{for all } k, m \in [L], G_{:,k} \neq G_{:,m}\}.$$

The by definition, $\bar{\mathbb{G}}_D$ is a token class with token-wise $((2ID)^{-1}, \sqrt{d}, (2ID)^{-1})$-separated sequence. Following the construction of $F^{\text{SA}}$ in proof of (Su et al., 2025, Theorem H.2), for sufficiently large $D$ we have:

$$F^{\text{SA}} \circ F_1^{\text{FF}}(Z)_{t,k} < \frac{1}{4D} \quad \text{for all} \quad Z \in \mathbb{R}^{d \times L} \setminus [-I, I]^{d \times L}, \quad t \in [d], k \in [L],$$

$$F^{\text{SA}} \circ F_1^{\text{FF}}(Z)_{t,k} > \frac{3}{4D} \quad \text{for all} \quad Z \in [-I, I]^{d \times L}, \quad t \in [d], k \in [L].$$

**Step 4: Memorization via the Second Feed-forward Layer.** Finally, we construct a bump function of scale $R > 0$ to map every $c \in \bar{\mathbb{G}}_D$ to its label $f(C)$ and sends any sequence that lies componentwise below the threshold $1/(4D)$ to zero. Precisely, for each $C \in \mathbb{G}_D^{\text{norm}}$ we construct a bump function of scale $R$:

$$\begin{aligned} \text{bump}_R(Z) = \;& \frac{f(2C - I)}{dL} \sum_{t=1}^{d} \sum_{k=1}^{L} (\text{RELU}[R(Z_{t,k} - C_{t,k}) - 1] \\ & - \text{ReLU}[R(Z_{t,k} - C_{t,k})] + \text{ReLU}[R(Z_{t,k} - C_{t,k}) + 1]). \end{aligned} \tag{B.3}$$

Summing up over $C \in \mathbb{G}_D^{\text{norm}}$ and we obtain the second feed-forward layer $F_2^{\text{FF}}$.

We sum up the error bound in four steps. As discussed in the step approximation using step function, there exists $D$ such that

$$d_F(f, g_1) < \frac{\epsilon}{3}.$$

By choosing sifficiently quantization step $\delta > 0$, we obtain

$$d_F(F_2^{\text{FF}} \circ F^{\text{SA}} \circ F_1^{\text{FF}}, F_2^{\text{FF}} \circ F^{\text{SA}} \circ g_2) < \frac{\epsilon}{3}.$$

By choosing granularity $D$ large enough, $|\mathbb{G}_D \backslash \mathbb{G}_D^{\text{norm}}|$ is negligible. Then we have for large enough $D$ and $R$,

$$d_F(F_2^{\text{FF}} \circ F^{\text{SA}} \circ g_2, g_1) < \frac{\epsilon}{3}.$$

Altogether, summing up the error we have:

$$d_F(f(Z), g(Z)) < \epsilon.$$

This completes the proof. □

**Parameter Norm Bounds of Transformer Approximator.**   Next, we compute the norm bound of the approximator transformer in Proposition B.17. This theorem is modified from of (Su et al., 2025, Lemma H.4). The main difference is that we also take polynomial factor of $L$ into the final result instead of neglecting it, while other parts of computation is similar.

**Theorem B.18** (Parameter Norm Bound for Approximator Transformer, Modified from Lemma H.4 of (Su et al., 2025))**.** *Let $\epsilon \in (0,1)$. Consider $Z \in [-I,I]^{d \times L}$ be input sequence, where $I > 0$ and $L > 2$. Let Let $f(Z) : [-I,I]^{d \times L} \to \mathbb{R}^{d \times L}$ be a Lipschitz continuous function with respect to Frobenius norm on a bounded domain. Then for the approximator $g(Z) = F_1^{\text{FF}} \circ F^{\text{SA}} \circ F_2^{\text{FF}} \in \mathcal{T}^{h,s,r}$ in Proposition B.17 within $\epsilon$ precision, i.e., $d_F(f,g) < \epsilon$, the parameter bound in the transformer network class follow:*

$$C_{KQ}, C_{KQ}^{2,\infty} = O(I^{4d+2} \epsilon^{-4d-2} L^{2d+1} \log L); C_{OV}, C_{OV}^{2,\infty} = O(\epsilon L^{-1/2})$$
$$C_F, C_F^{2,\infty} = O(I \epsilon^{-1} L \cdot \max \|f(Z)\|_F); C_E = O(L),$$

*where $O(\cdot)$ hides polynomial factors depending on $d$.*

*Proof.* The proof is modified from proof of (Su et al., 2025, Lemma H.4).

Recall that we take sufficiently large $D, R$ and sufficiently small $\delta$ in proof of Proposition B.17 to ensure the precision. We then start with bounding $D, R, \delta$ in terms of $\epsilon$.

- **Bound on $\delta$.** Recall the approximation in (B.1) and (B.2). To guarantee the effect of grid, we hope partition $(i/D, i/D + \delta)$ is a contained in the interval $(i/D, (i+1)/D)$ where $i \in \mathbb{Z}^+$. Then it's sufficient to take $\delta = o(1/D)$.

- **Bound on $D$.** For Lipschitz continuous function $f$ with respect to Frobenius norm with Lipschitz constant $L_f$, we have

$$d_F(f(Z), g_1(Z)) \leq L_f \|Z - Z^*\|_F \leq 2\sqrt{dL} L_f / D,$$

where $Z^* = \operatorname{argmin}_{Z' \in \mathbb{G}_D} \|Z - Z'\|_F$. Then we take $D = O(\epsilon^{-1} \sqrt{L})$.

- **Bound on $R$.** To obtain the correct labeling in (B.3), we need $S_{t,k} := Z_{t,k} - C_{t,k} \in (0, 1/R)$ for all $t \in [d], k \in [L]$. Then since $C_{t,k}$ is defined on $\mathbb{G}_D^{\text{norm}}$ with granularity $2DI$ and is chosen close enough to $Z_{t,k}$, it suffices to set $R = O(DI)$.

Next, we get the norm bound of matrices on the basis of computation above. Since the $F^{\text{SA}}$ is a single head self-attention layer, we directly write $W_K^1, W_Q^1, W_V^1, W_O^1$ as $W_K, W_Q, W_V, W_O$ for simplicity of notations.

- **Bound on Norm of $W_{KQ}$.** Recall that in the proof of Theorem H.1 of (Su et al., 2025), $W_K$ and $W_Q$ follows the construction of

$$W_K = \sum_{i=1}^{\rho} p_i q_i^\top \in \mathbb{R}^{s \times d}, W_Q = \sum_{i=1}^{\rho} p_i' q_i'^\top \in \mathbb{R}^{s \times d},$$

where $p_i^\top p_i' = (|\mathcal{V}| + 1)^4 d \log L / (\epsilon_s \gamma_{\min})$. Then we have

$$\|W_{KQ}\|_2 \le \|W_{KQ}\|_F = \|W_K^\top W_Q\|_F = O(\frac{4 \log L |V|^4}{\epsilon_s \gamma_{\min}}),$$

$$\|W_{KQ}\|_{2,\infty} = \|W_K^\top W_Q\|_{2,\infty} = O(\frac{4 \log L |V|^4}{\epsilon_s \gamma_{\min}}).$$

Recall that input $\bar{\mathbb{G}}_D$ is a token class with token-wise $((2ID)^{-1}, \sqrt{d}, (2ID)^{-1})$-separated sequence. Then $|V| = O((DI)^d)$ and $\gamma_{\min}, \epsilon_s = (2DI)^{-1}$. Finally, by $D = O(\epsilon^{-1}\sqrt{L})$, we get

$$\|W_{KQ}\|_2, \|W_{KQ}\|_{2,\infty} = O(\epsilon^{-4d-2} I^{4d+2} L^{2d+1} \log L).$$

- **Bound on Norm of $W_{OV}$.** Recall that in the proof of Theorem H.1 of (Su et al., 2025), $W_O$ and $W_V$ follows the construction of

$$W_O = \sum_{i=1}^{\rho} p_i'' q_i''^\top \in \mathbb{R}^{s \times d}, W_V = \sum_{i=1}^{\rho} p_i'''' p_i''^\top \in \mathbb{R}^{d \times s},$$

where $\|p_i'''\| \lesssim \epsilon_s / (4\rho \gamma_{\max} \|p_i''\|)$ and $p_i'' \in \mathbb{R}^s$ is any nonzero vector. Then with the $(\gamma_{\min} = 1/D, \gamma_{\max} = \sqrt{d}, \epsilon_s = 1/D)$-separateness and $D = O(\epsilon^{-1}\sqrt{L}), \rho < d$, we get:

$$\|W_V\|_2 = \sup_{\|x\|_2=1} \|W_V x\|_2 \le O(\sqrt{\rho}) \le O(\sqrt{d}),$$

$$\|W_V\|_{2,\infty} = \max_{1 \le i \le d} \|(W_V)_{(i,:)}\|_2 \le O(\rho) \le O(d),$$

$$\|W_O\|_2 = \sup_{\|x\|_2=1} \|W_O x\|_2 \le O(\sqrt{\rho} \cdot \rho^{-1} \cdot \gamma_{\max}^{-1} \cdot \epsilon_s) = O(d^{-1}\epsilon L^{-1/2}),$$

$$\|W_O\|_{2,\infty} = \max_{1 \le i \le d} \|(W_O)_{(i,:)}\|_2 \le O(\rho \cdot \rho^{-1} \cdot \gamma_{\max}^{-1} \cdot \epsilon_s) = O(d^{-1/2}\epsilon L^{-1/2}).$$

Therefore we get:

$$\|W_{OV}\|_2 = \|W_O W_V\|_2 \le O(\epsilon L^{-1/2}), \|W_{OV}\|_{2,\infty} = \|W_O W_V\|_{2,\infty} \le O(\epsilon L^{-1/2}).$$

- **Bound on Norm of $W_1^1$ and $W_2^1$ in $F_1^{\text{FF}}$.** Recall that in the construction of the first feed-forward layer, we have two key approximator:

$$f_1(z) := -I + \sum_{t=-ID}^{I(D-1)} \frac{\text{RELU}[z/\delta - \text{t}/\delta\text{D}] - \text{ReLU}[z/\delta - 1 - \text{t}/\delta\text{D}]}{D},$$

and

$$f_2(z) = \text{ReLU}[(z-I)/\delta] - \text{ReLU}[(z-I)/\delta + 1] + \\ \text{ReLU}[(-z-I)/\delta] - \text{ReLU}[(-z-I)/\delta + 1].$$

Then for $t \in [d], k \in [L]$, we approximate each entry of $g_1(Z)$ with

$$F_1^{\text{FF}}(Z)_{t,k} = \frac{f_1(Z_{t,k}) + I}{2I} + f_2(Z_{t,k}).$$

That's to say, each element in $W_1$ and $W_2$ is bounded by $1/\delta$ and $I$. Recall that $\delta = o(1/D)$ and $D = O(\epsilon^{-1}\sqrt{L})$, we have

$$\max\{\|W_1^1\|_2, \|W_2^1\|_2\} \le O(\epsilon^{-1}L), \max\{\|W_1^1\|_{2,\infty}, \|W_2^1\|_{2,\infty}\} \le O(\epsilon^{-1}L).$$

- **Bound on Norm of $W_1^2$ and $W_2^2$ in $F_2^{\text{FF}}$.** Recall that in the construction of the second feed-forward layer, we construct $F_2^{\text{FF}}$ through bump function:

$$\text{bump}_R(Z) = \frac{f(2C-I)}{dL} \sum_{t=1}^{d} \sum_{k=1}^{L} (\text{RELU}[\text{R}(\text{Z}_{\text{t,k}} - \text{C}_{\text{t,k}}) - 1]$$

$$- \mathsf{ReLU}[R(Z_{t,k} - C_{t,k})] + \mathsf{ReLU}[R(Z_{t,k} - C_{t,k}) + 1]).$$

Therefore, the output of $F_2^{\mathrm{FF}}$ is bounded by $R$ and $\max \|f(Z)\|_F$. Then by $R = O(DI)$ and $D = O(\epsilon^{-1}\sqrt{L})$, we have:

$$\max\{\|W_1^2\|_2, \|W_2^2\|_2\} \leq O(I\epsilon^{-1}L \cdot \max \|f(Z)\|_F),$$
$$\max\{\|W_1^2\|_{2,\infty}, \|W_2^2\|_{2,\infty}\} \leq O(I\epsilon^{-1}L \cdot \max \|f(Z)\|_F).$$

- **Bound on Norm of Encoding Matrix** $E$. By (Kajitsuka & Sato, 2023, Corollary 2), it suffices to take the encoding matrix $E$:

$$E = \begin{bmatrix} 2\gamma_{\max} & 4\gamma_{\max} & \dots & 2L\gamma_{\max} \\ 2\gamma_{\max} & 4\gamma_{\max} & \dots & 2L\gamma_{\max} \\ \vdots & \vdots & & \vdots \\ 2\gamma_{\max} & 4\gamma_{\max} & \dots & 2L\gamma_{\max} \end{bmatrix}.$$

Recall that we have $\gamma_{\max} = \sqrt{d}$, then we obtain:

$$\|E^\top\|_{2,\infty} = \sqrt{4dL^2}\gamma_{\max} = O(L).$$

This complete the proof. $\qquad\square$

## C    PROOF OF THEOREM 3.1

This section provides the proof of Theorem 3.1.

**Organization.** Due to the complexity of the proof, our proof proceeds in two steps: (i) First, in Lemma C.3, we establish a key intermediate result by presenting the distribution error in terms of the velocity error. (ii) Second, building upon this lemma, we present the final proof of Theorem 3.1 in Theorem C.5 by applying Grönwall's Inequality.

### C.1    PRELIMINARIES

At beginning, we recall that for any matrix $Z$, $\|Z\|_1$ denote the operator norm induced by vector $\ell_1$ norm. Total Variation Distance is a distance function defined between probability distributions. We start with the definition of total variation distance as below.

**Definition C.1** (Total Variation Distance). *Let $P$ and $Q$ be two probability distributions defined on a discrete state space $S$, with corresponding probability mass functions $p(x)$ and $q(x)$. Then the total variation (TV) distance between them is defined as:*

$$TV(P, Q) := \frac{1}{2} \sum_{x \in S} |p(x) - q(x)|.$$

Our analysis relies on Grönwall's Inequality. It is a fundamental tool for establishing bounds on the solutions of ordinary differential equations (ODEs).

**Lemma C.2** (Grönwall's Inequality, (Gronwall, 1919)). *Let $a, b \in \mathbb{R}$ satisfy $a < b$. Let $y(t)$ and $f(t)$ be two real value function defined on $[a, b]$. Suppose that $y(t)$ is differentiable on $[a, b]$ and satisfies:*

$$\frac{\mathrm{d}}{\mathrm{d}t} y(t) \leq y(t) f(t), t \in [a, b].$$

*Then we have:*

$$y(b) \leq y(a) \exp\left( \int_a^b f(s) \mathrm{d}s \right).$$

To analyze the distribution error, we first express the DFM framework in the language of linear algebra. Let the discrete state space be indexed as $S = \{w_1, \ldots, w_{|S|}\}$. We represent the ground-truth probability mass function $p_t(x)$ and its corresponding estimator $p_{t,\theta}(x)$ as vectors $p_t, p_{t,\theta}$ in $\mathbb{R}^{|S|}$, where $p_t[i] = p_t(w_i)$ and $p_{t,\theta}[i] = p_{t,\theta}(w_i)$. Similarly, the velocity fields are represented as rates matrices $U_t, U_{t,\theta} \in \mathbb{R}^{|S| \times |S|}$, where the entry $[U_t]_{ij} = u_t(w_i, w_j), [U_{t,\theta}]_{ij} = u_{t,\theta}(w_i, w_j)$ is the corresponding rate of transition from state $w_j$ to state $w_i$. With these definitions, we rewrite the Kolmogorov Forward Equation (2.3) for both the true and estimated paths into the compact matrix form (C.1). Then we derive the distribution error bounds in terms of the risk function.

**Lemma C.3** (Variation of Constants Formula for Error). *Let the true probability vector $p_t$ and the estimated vector $p_{t,\theta}$ be solutions to the linear differential systems*

$$\begin{cases} \frac{\mathrm{d}}{\mathrm{d}t} p_t = U_t p_t, \\ \frac{\mathrm{d}}{\mathrm{d}t} p_{t,\theta} = U_{t,\theta} p_{t,\theta}, \end{cases} \tag{C.1}$$

*with a shared initial condition $p_0 = p_{0,\theta}$. Let the evolution operator for the estimated system be $\mathcal{P}_{s,t,\theta} \in \mathbb{R}^{|S| \times |S|}$, which is the solution to $\frac{\mathrm{d}}{\mathrm{d}t} \mathcal{P}_{s,t,\theta} = U_{t,\theta} \mathcal{P}_{s,t,\theta}$ with $\mathcal{P}_{s,s,\theta} = I$. The difference between the distributions at time $t > 0$ is given by*

$$p_{t,\theta} - p_t = \int_0^t \mathcal{P}_{s,t,\theta}(U_{s,\theta} - U_s) p_s \mathrm{d}s.$$

*Proof.* We construct the helping function $Z(s)$ as:

$$Z(s) = \mathcal{P}_{s,t,\theta} p_s. \tag{C.2}$$

Then since $\mathcal{P}_{s,t,\theta}$ and $p_s$ is differentiable, we have:

$$\frac{\mathrm{d}}{\mathrm{d}s}Z(s) = (\frac{\mathrm{d}}{\mathrm{d}s}\mathcal{P}_{s,t,\theta})p_s + \mathcal{P}_{s,t,\theta}(\frac{\mathrm{d}}{\mathrm{d}s}p_s) \qquad \text{(By (C.2))}$$

$$= (-\mathcal{P}_{s,t,\theta}U_{s,\theta})p_s + \mathcal{P}_{s,t,\theta}(U_s p_s) \qquad \text{(By backward Kolmogorov Equation (2.3) and (C.1))}$$

$$= -\mathcal{P}_{s,t,\theta}(U_{s,\theta} - U_s)p_s. \qquad (C.3)$$

Integrating (C.3), we obtain:

$$-\int_0^t \mathcal{P}_{s,t,\theta}(U_{s,\theta} - U_s)p_s \mathrm{d}s = \int_0^t \frac{\mathrm{d}}{\mathrm{d}s}Z(s)\mathrm{d}s = Z(t) - Z(0) = p_t - p_{t,\theta}.$$

This completes the proof. $\qquad \square$

## C.2 MAIN PROOF OF THEOREM 3.1

Before studying the tractable factorized velocity (Section 2) case, we first establish a foundational error bound for the general discrete flow matching framework. Presenting the universal principle first makes our subsequent, more detailed analysis clearer and more readable.

**Theorem C.4** (Error Bounds for Discrete Flow Matching). *Consider the discrete state space $\mathcal{S} = \mathcal{T}^d$, where the vocabulary is $\mathcal{T} = \{1, \ldots, M\}$. Let $P$ be the true data distribution over $\mathcal{S}$. Let $\widehat{u}_\theta$ be the velocity estimator, with parameters $\widehat{\Theta}$ and let $\widehat{P}$ be the distribution generated using this estimator. Define the risk of the velocity estimator as*

$$\mathcal{R}(\widehat{\Theta}) := \int_0^1 \mathop{\mathbb{E}}_{X_t \sim p_t(x)} \|u(X_t, t) - u_{\widehat{\theta}}(X_t, t)\|_2^2 \mathrm{d}t,$$

*where $p_t(x)$ is the true probability path. Then, the total variation distance between the true and generated distributions is bounded by the risk of this estimator:*

$$TV(P, \widehat{P}) \lesssim \exp(2M_u)M^{\frac{d}{2}}\sqrt{\mathcal{R}(\widehat{\Theta})},$$

*where $M_u$ is the upper bound of the true velocity field, satisfying $\max\limits_{y,x\in S, t\in[0,1]}|u_t(y,x)| \leq M_u$.*

*Proof.* Following the definitions in Lemma C.3, let $p_t \in \mathbb{R}^{M^d}$ and $U_t \in \mathbb{R}^{M^d \times M^d}$ be the true probability vector and the true rates matrix. Let $\widehat{p}_{t,\theta} \in \mathbb{R}^{M^d}$ and $\widehat{U}_{t,\theta} \in \mathbb{R}^{M^d \times M^d}$ be the estimated probability vectors and rates matrix defined by estimated velocity $\widehat{u}_{t,\theta}$. Then we have:

$$\begin{cases} \frac{\mathrm{d}}{\mathrm{d}t}p_t = U_t p_t, \\ \frac{\mathrm{d}}{\mathrm{d}t}\widehat{p}_{t,\theta} = \widehat{U}_{t,\theta}p_{t,\theta}, \\ p_0 = \widehat{p}_{0,\theta} \sim P_0. \end{cases}$$

Let evolution operator $\mathcal{P}_{s,t,\theta}$ be defined as in Lemma C.3. Then by definition of total variation distance, we have:

$$\mathrm{TV}(P, \widehat{P}) = \frac{1}{2}\sum_{x\in S}|p_1(x) - p_{1,\theta}(x)|$$

$$= \frac{1}{2}\|\widehat{p}_{1,\theta} - p_1\|_1 \qquad \text{(By the definition of vectors } \widehat{p}_{1,\theta} \text{ and } p_1)$$

$$= \frac{1}{2}\|\int_0^1 \mathcal{P}_{s,1,\theta}(U_{s,\theta} - U_s)p_s \mathrm{d}s\|_1 \qquad \text{(By Lemma C.3)}$$

$$\leq \frac{1}{2}\int_0^1 \|\mathcal{P}_{s,1,\theta}(U_{s,\theta} - U_s)p_s\|_1 \mathrm{d}s,$$

$$\leq \frac{1}{2}\int_0^1 \|\mathcal{P}_{s,1,\theta}\|_1 \|(U_{s,\theta} - U_s)p_s\|_1 \mathrm{d}s \qquad (C.4)$$

where the last line follows by $\|p_s\|_1 = 1$ the sub-multiplicative property of norm $\|\cdot\|_1$. To bound $\|\mathcal{P}_{s,1,\theta}\|_F$, we first bound the derivative of $\|\mathcal{P}_{s,t,\theta}\|_F$. Let $M$ be the vocabulary size and set the transformer estimator $u_{\widehat{\theta}}$ bound by $M_u$, then we have

$$
\begin{aligned}
\frac{\partial}{\partial t}\|\mathcal{P}_{s,t,\theta}\|_1 &\leq \|\frac{\partial}{\partial t}\mathcal{P}_{s,t,\theta}\|_1 && \text{(By the Lipschitz continuous of } \ell_1 \text{ norm)}\\
&= \|U_{t,\theta}\mathcal{P}_{s,t,\theta}\|_1 && \text{(By Kolmogorov Equation)}\\
&\leq \|U_{t,\theta}\|_1 \cdot \|\mathcal{P}_{s,t,\theta}\|_1 && \text{(By sub-multiplicativity of operator norm)}\\
&\leq 2M_u\|\mathcal{P}_{s,t,\theta}\|_1, && \text{(C.5)}
\end{aligned}
$$

where the last line follows by the maximum of the sum of absolute value if entry of $U_{t,\theta}$ in each column is less than $2M_u$, since $\sum_{i=1}^{M^d}|[U_{t,\theta}]_{ij}| = -[U_{t,\theta}]_{jj} + \sum_{i\neq j}|[U_{t,\theta}]_{ij}| \leq 2M_u$ (please see the rates condition (2.2).) Then by Grönwall's Inequality Lemma C.2, we have:

$$
\|\mathcal{P}_{s,1,\theta}\|_1 \leq \|\mathcal{P}_{s,s,\theta}\|_1 \exp\left(\int_0^t 2M_u \mathrm{d}s\right) \leq \exp(2M_u). \tag{C.6}
$$

Substituting (C.6) into (C.4), we get:

$$
\begin{aligned}
\mathrm{TV}(P,\widehat{P}) &\leq \frac{1}{2}\int_0^1 \|\mathcal{P}_{s,1,\theta}\|_1 \|(U_{s,\theta} - U_s)p_s\|_1 \mathrm{d}s && \text{(By (C.4))}\\
&\leq \frac{1}{2}\int_0^1 \exp(2M_u)\|(U_{s,\theta} - U_s)p_s\|_1 \mathrm{d}s\\
&\lesssim \exp(2M_u)M^{\frac{d}{2}}\sqrt{\mathcal{R}(\widehat{\Theta})},
\end{aligned}
$$

where the last line is by the definition of risk function $\mathcal{R}$ and Cauchy-Schwarz inequality.

This completes the proof. $\qquad\qquad\square$

Now we consider the error bounds for discrete flow matching with factorized velocity (Section 2).

**Theorem C.5** (Error Bounds for Discrete Flow Matching with Factorized Velocity, Theorem 3.1 Restate). *Consider the discrete state space $\mathcal{S} = \mathcal{T}^d$ with vocabulary $\mathcal{T} = \{1,\ldots,M\}$. Let $P$ be the true data distribution and let $\widehat{P}$ be the distribution generated by a DFM model using factorized velocity estimators $\widehat{u}_\theta^1,\ldots,\widehat{u}_\theta^d$. For each coordinate $i_0 \in [d]$, define the factorized risk as the mean squared error of its velocity estimator:*

$$
\mathcal{R}^{i_0}(\widehat{\Theta}) := \int_{t_0}^T \mathop{\mathbb{E}}_{X_t\sim p_t(x)}\|u^{i_0}(X_t,t) - \widehat{u}_\theta^{i_0}(X_t,t)\|_2^2 \mathrm{d}t,
$$

*where the time interval is clipped to $[t_0,T]$ to ensure numerical stability and $p_t(x)$ is true probability path generated by $u^1,\ldots,u^d$. Then, the total variation distance between the true and generated distributions is bounded by the sum of the risks from each factorized component:*

$$
TV(P,\widehat{P}) \lesssim \sqrt{M}\exp(2M_u)\sum_{i_0\in[d]}\sqrt{\mathcal{R}^{i_0}(\widehat{\Theta})},
$$

*where $M_u$ is the upper bound of estimated velocity such that $\left|u_t^{\theta,i}(y,x)\right| \leq M_u$ for all $y,x \in \mathcal{S}$.*

*Proof.* Following the definitions in Lemma C.3 and Theorem C.4. Let evolution operator $\mathcal{P}_{s,t,\theta}^{i_0}$ be the transformation operator for coordinate $i_0$. Let $p_t^{i_0} \in \mathbb{R}^M$ and $U_t^{i_0} \in \mathbb{R}^{M\times M}$ be the true probability vector and the true rates matrix for coordinate $i_0$. Let $\widehat{p}_{t,\theta}^{i_0} \in \mathbb{R}^M$ and $\widehat{U}_{t,\theta}^{i_0} \in \mathbb{R}^{M\times M}$ be the estimated probability vectors and rates matrix defined by estimated velocity $\widehat{u}_{t,\theta}$ for coordinate $i_0$. Then by definition of total variation distance, we have:

$$
\mathrm{TV}(P,\widehat{P}) = \frac{1}{2}\sum_{x\in S}|p_1(x) - p_{1,\theta}(x)|
$$

$$\leq \frac{1}{2} \sum_{i_0 \in [d]} \sum_{x \in S} |p_1^{i_0}(x) - p_{1,\theta}^{i_0}(x)| \qquad \text{(By } p_1 = \prod_{i_0=1}^{d} p_1^{i_0} \text{ and } p_{1,\theta} = \prod_{i_0=1}^{d} p_{1,\theta}^{i_0})$$

$$\lesssim \int_0^1 \sum_{i_0 \in [d]} \|\mathcal{P}_{s,1,\theta}^{i_0}\|_1 \cdot \|(U_{s,\theta}^{i_0} - U_s^{i_0})p_s^{i_0}\|_1 \mathrm{d}s, \tag{C.7}$$

where the last equation follows from the proof of Theorem C.4. As in Theorem C.4, we then bound the derivative of $\|\mathcal{P}_{s,t,\theta}^{i_0}\|_F$. Let $M$ be the vocabulary size and set the transformer estimator $u_{\widehat{\theta}}^{i_0}$ bound by $M_u$ for any $i_0 \in [d]$, then we have

$$\frac{\partial}{\partial t}\|\mathcal{P}_{s,t,\theta}^{i_0}\|_1 \leq \|\frac{\partial}{\partial t}\mathcal{P}_{s,t,\theta}^{i_0}\|_1 \qquad \text{(By the Lipschitz continuous of } \ell_1 \text{ norm)}$$

$$= \|U_{t,\theta}^{i_0}\mathcal{P}_{s,t,\theta}^{i_0}\|_1 \qquad \text{(By Kolmogorov Equation)}$$

$$\leq \|U_{t,\theta}^{i_0}\|_1 \cdot \|\mathcal{P}_{s,t,\theta}^{i_0}\|_1 \qquad \text{(By sub-multiplicativity of operator norm)}$$

$$\leq 2M_u\|\mathcal{P}_{s,t,\theta}^{i_0}\|_1, \tag{C.8}$$

where the last line follows by the maximum of the sum of absolute value of entry of $U_{t,\theta}^{i_0}$ in each column is less than $2M_u$, since $\sum_{i=1}^{M} \left|[U_{t,\theta}^{i_0}]_{ij}\right| = -[U_{t,\theta}^{i_0}]_{jj} + \sum_{i \neq j} \left|[U_{t,\theta}^{i_0}]_{ij}\right| \leq 2M_u$ (please see the rates condition (2.2).) Then by Grönwall's Inequality Lemma C.2, we have:

$$\|\mathcal{P}_{s,1,\theta}^{i_0}\|_1 \leq \|\mathcal{P}_{s,s,\theta}^{i_0}\|_1 \exp\left(\int_0^t 2M_u \mathrm{d}s\right) \leq \exp(2M_u). \tag{C.9}$$

Substituting (C.9) into (C.7), we get:

$$\mathrm{TV}(P, \widehat{P}) \leq \int_0^1 \sum_{i_0 \in [d]} \|\mathcal{P}_{s,1,\theta}^{i_0}\|_1 \cdot \|(U_{s,\theta}^{i_0} - U_s^{i_0})p_s^{i_0}\|_1 \mathrm{d}s \qquad \text{(By (C.7))}$$

$$\leq \sum_{i_0 \in [d]} \int_0^1 \exp(2M_u)\|(U_{s,\theta}^{i_0} - U_s^{i_0})p_s^{i_0}\|_1 \mathrm{d}s$$

$$\lesssim \sqrt{M} \exp(2M_u) \sum_{i_0 \in [d]} \sqrt{\mathcal{R}^{i_0}(\widehat{\Theta})},$$

where the last line is by the definition of risk function $\mathcal{R}^{i_0}$ and Cauchy-Schwarz inequality.

This completes the proof. $\qquad \square$

# D  PROOF OF LEMMA 4.5

This section provides the proof of Lemma 4.5. Note that we view $\mathcal{T} = [M]$ as a subspace of $\mathbb{R}$ in this section. In other words, we embed $\mathcal{S} = \mathcal{T}^d$ into $\mathbb{R}^d$ through the inclusion map $E : \mathcal{S} \hookrightarrow \mathbb{R}^d$.

**Organization.** In Appendix D.1, we define a $C^\infty$ function $\eta(x)$ and derive bounds for all its derivatives in Lemma D.1. Then we present the main proof of Lemma 4.5 in Appendix D.2 on the basis of function constructed in Lemma D.1.

## D.1  PRELIMINARIES

We start with defining a smooth function $\eta(x)$ and bounding its derivatives.

**Lemma D.1.** *Define $\eta(x) : [0, \infty) \to [0, 1]$ by*

$$
\eta(x) = \begin{cases} e \cdot \exp\left(-\frac{1}{1-x}\right), & x \in [0, 1), \\ 0, & x \in [1, \infty). \end{cases}
$$

*Then $\eta(t) \in C^\infty$ and $\left|\frac{\mathrm{d}^n \eta}{\mathrm{d}x^n}(x)\right| \leq 3e \cdot \left(\frac{2n}{e}\right)^{2n}$ for all $x \in [0, 1]$.*

*Proof.* Our proof consists of three steps.

**Step 1: Smoothness.** First, we show that $\eta(x) \in C^\infty$.

For $x > 1$, $\eta(x) = 0$ so all derivatives vanish (i.e., $\frac{\mathrm{d}^n \eta}{\mathrm{d}x^n} = 0$ for any $n \in \mathbb{Z}^+$).

For $x \in [0, 1)$, we have $\frac{\mathrm{d}\eta}{\mathrm{d}x}(x) = e \cdot \left(-\frac{1}{(x-1)^2}\right) \cdot \exp\left(\frac{1}{x-1}\right)$.

We denote, for $x < 1$,

$$
\frac{\mathrm{d}^n \eta}{\mathrm{d}x^n}(x) := e \cdot p_n\left(\frac{1}{x-1}\right) \cdot \exp\left(\frac{1}{x-1}\right), \quad \text{for} \quad n \in \mathbb{Z}^+, \tag{D.1}
$$

where $p_n(x)$ is a function to be determined. Then $p_0(x) = 1$ and $p_1(x) = -x^2$.

For $n \in \mathbb{Z}^+$, it holds

$$
\begin{aligned}
&\frac{\mathrm{d}^n \eta}{\mathrm{d}x^n}(x) \\
&= \frac{\mathrm{d}}{\mathrm{d}x}\left(\frac{\mathrm{d}^{n-1}\eta}{\mathrm{d}x^{n-1}}\right)(x) \\
&= \frac{\mathrm{d}}{\mathrm{d}x}\left(e \cdot p_{n-1}\left(\frac{1}{x-1}\right) \cdot \exp\left(\frac{1}{x-1}\right)\right) \\
&= e \cdot \left(-\frac{1}{(x-1)^2}p'_{n-1}\left(\frac{1}{x-1}\right) + -\frac{1}{(x-1)^2}p_{n-1}\left(\frac{1}{x-1}\right)\right) \cdot \exp\left(\frac{1}{x-1}\right). \quad \text{(By chain rule)}
\end{aligned}
$$

Then we have the recurrence relation $p_n(x) = -x^2(p_{n-1}(x) + p'_{n-1}(x))$ for $n \in \mathbb{Z}^+$ and $p_0(x) = 1$.

By induction, $p_n$ is a polynomial of degree $2n$. Then for $n \in \mathbb{Z}^+$, we have

$$
\begin{aligned}
\lim_{x \to 1^-} \frac{\mathrm{d}^n \eta}{\mathrm{d}x^n}(x) &= \lim_{x \to 1^-} e \cdot p_n\left(\frac{1}{x-1}\right) \exp\left(\frac{1}{x-1}\right) && \text{(By (D.1))} \\
&= \lim_{x \to -\infty} e \cdot p_n(x) \cdot e^x && \text{(By setting } z = \tfrac{1}{x-1}) \\
&= 0 && ((\text{polynomial}) \cdot e^z \text{ vanishes as } z \to -\infty) \\
&= \lim_{x \to 1^+} \frac{\mathrm{d}^n \eta}{\mathrm{d}x^n}(x), && \text{(By } \eta(x) = 0 \text{ for } x > 1)
\end{aligned}
$$

showing that all derivatives match at $x = 1$.

Then, since $\eta(x)$ is smooth on $[0, 1)$ and $(1, \infty)$, we have $\eta(x) \in C^\infty$.

**Step 2: Growth Bound.** Next, we bound $|\frac{\mathrm{d}^n \eta}{\mathrm{d}x^n}(x)|$.

Define $q_n(x) := |e^x \cdot p_n(x)|$. Then $q_{n+1}(x) = x^2 \cdot q_n'(x)$, $n \in \mathbb{Z}^+$.

Introduce the generating function

$$G(t, x) := \sum_{n \geq 0} \frac{q_n(x)}{n!} t^n.$$

Then $G(t, x)$ satisfies the partial differential equation

$$\partial_t G(t, x) = x^2 \partial_x G(t, x), \quad G(0, x) = e^x. \tag{D.2}$$

One solution to (D.2) is $G(t, x) = \exp\left(\frac{x}{1-xt}\right)$. Hence, $\frac{q_n(x)}{n!}$ is the $t^n$-th coefficient in the Taylor expansion of $G(t, x) = \exp\left(\frac{x}{1-xt}\right)$ at point $t = 0$.

By Cauchy integral formula, for all $x \leq -1$ and $0 < r < 1/|x|$, we have

$$\frac{q_n(x)}{n!} = \frac{1}{2\pi i} \int_{|t|=r, t \in \mathbb{C}} \frac{G(t, x)}{t^{n+1}} \mathrm{d}t$$

$$\leq \frac{1}{2\pi} \int_{|t|=r, t \in \mathbb{C}} |\frac{G(t, x)}{t^{n+1}}| \mathrm{d}t \qquad \text{\small (By $q_n(x) \in \mathbb{R}$ and $|\int f \mathrm{d}x| \leq \int |f| \mathrm{d}x$)}$$

$$\leq \frac{1}{2\pi} \cdot \frac{\max_{|t|=r, t \in \mathbb{C}} |G(t, x)|}{r^{n+1}} \int_{|t|=r, t \in \mathbb{C}} 1 \mathrm{d}x \qquad \text{\small (By $\int |fg| \mathrm{d}x \leq (\sup |f|) \int |g| \mathrm{d}x$)}$$

$$= \frac{\max_{|t|=r, t \in \mathbb{C}} |G(t, x)|}{r^n}. \tag{D.3}$$

Further, for $x \leq -1$ and $0 < r < 1/|x|$, we set $t = re^{i\theta} \in \mathbb{C}$ and get

$$\max_{|t|=r, t \in C} |G(t, x)| = \max_{\theta \in [0, 2\pi]} |\exp\left(\frac{x}{1 - xre^{i\theta}}\right)|$$

$$= \max_{\theta \in [0, 2\pi]} \exp\left(x \cdot \mathrm{Re}(\frac{1}{1 - xre^{i\theta}})\right)$$

$$= \exp\left(\frac{x}{1 - xr}\right), \tag{D.4}$$

where the second line is by $|\exp(z)| = \exp(\mathrm{Re}(z))$, and the last line is by $x \leq -1$ and $|xr| < 1$.

Substituting (D.4) into (D.3), we get

$$\frac{q_n(x)}{n!} \leq \inf_{0 < r < \frac{1}{|x|}} \frac{1}{r^n} \exp\left(\frac{x}{1 - xr}\right)$$

$$\leq (2|x|)^n \cdot \exp\left(\frac{2}{3}x\right)$$

$$\leq (\frac{3n}{e})^n, \tag{D.5}$$

where the second line is by setting $r = \frac{1}{2|x|}$, and the last line follow from $x \leq -1$ and optimizing over $x$.

**Step 3: Final Bound.** Finally we bound $|\frac{\mathrm{d}^n \eta}{\mathrm{d}x^n}(x)|$ as

$$|\frac{\mathrm{d}^n \eta}{\mathrm{d}x^n}(x)| = e \cdot q_n(\frac{1}{x-1})$$

$$\leq e \cdot n! \cdot (\frac{3n}{e})^n \qquad \text{\small (By (D.5) and $\frac{1}{x-1} \leq -1$ when $x \in [0, 1)$)}$$

$$\leq e \cdot (\frac{n}{e})^n \cdot \sqrt{2\pi n} \cdot e^{\frac{1}{12n}} \cdot (\frac{3n}{e})^n \qquad \text{\small (By Stirling's formula)}$$

$$\leq 3e \cdot (\frac{2n}{e})^{2n}.$$

This completes the proof. □

## D.2 MAIN PROOF OF LEMMA 4.5

We now establish Lemma 4.5 building on Lemma D.1. This lemma guarantees the existence of a smooth function that interpolates a given discrete function by matching its values at prescribed points. In this way, it provides a bridge between discrete functions and their continuous counterparts.

**Lemma D.2** (Lemma 4.5 Restate). *Let $\mathcal{S} \subset \mathbb{R}^d$ be the state space of discrete flow matching. Recall that $\mathcal{S} \subset \mathbb{R}^d$ is a 1-separated finite set with $|\mathcal{S}| = M^d$, i.e., $\|s - s'\| \geq 1$ for all distinct $s, s' \in \mathcal{S}$. For each $x \in \mathcal{S}$, let $u(x, \cdot) \in \mathcal{H}_{1,M^d}^{\beta}([0,1], K)$ with $\beta = k_1 + \gamma \geq 1$ and $k_1 = \lfloor \beta \rfloor$. Then there exists an extension $\widetilde{u} : \mathbb{R}^d \times [0,1] \to \mathbb{R}$ such that*

$$\widetilde{u}(x, t) \in \mathcal{H}_{d+1,M^d}^{\beta}(\mathbb{R}^d \times [0,1], 3e \cdot (k_1 + 2)(2k_1)^{2k_1} K M^d),$$

*and $\widetilde{u}(s, t) = u(s, t)$ for all $s \in \mathcal{S}$ and $t \in [0,1]$.*

*Proof.* Our proof consists of five steps. We give the construction of $\widetilde{u}(x, t)$ first in **Step 1 & 2**, and then prove that it has desired properties in **Step 3 - 5**.

**Step 1: Partition of Unity around $\mathcal{S}$.** Let $\eta(t)$ be the bump function from Lemma D.1.

Let $r = 1/e < 1/2$. For $s \in \mathcal{S} \subset \mathbb{R}^d$, we define $\phi_s^r(x) : \mathbb{R}^d \to \mathbb{R}$ as

$$\phi_s^r(x) := \eta\left(\frac{\|x - s\|^2}{r^2}\right).$$

Then $\phi_s^r(x) \in C^\infty$, $\text{supp } \phi_s^r(x) \subset B(s, r) = \{x \in \mathbb{R}^d \mid \|x - s\| < r\}$, and $\phi_s^r(s) = 1$. Since $\mathcal{S}$ is 1-separated and $r < 1/2$, the supports $\{\text{supp}(\phi_s)\}_{s \in \mathcal{S}}$ are pairwise disjoint.

**Step 2: Extension.** Next, we extend the discrete function $u$ to a continuous function $\widetilde{u}$.

We construct $\widetilde{u}(x, t)$ as:

$$\widetilde{u}(x, t) = \sum_{s \in \mathcal{S}} \phi_s(x) \cdot u(s, t). \tag{D.6}$$

Disjointness implies that for each fixed $x$ at most one term in (D.6) is nonzero. Hence, $\widetilde{u}(s, t) = u(s, t)$ for all $s \in \mathcal{S}$.

**Step 3: Derivative Bounds up to Order $k_1$.** We now prove $\widetilde{u}(x, t) \in \mathcal{H}_{d,M^d}^{\beta'}(\mathbb{R}^d \times [0,1], K')$.

Let $(\lambda, m)$ be multi-indices with $\lambda \in \mathbb{N}_0^d$, $m \in \mathbb{N}_0$, and $|\lambda| + m \leq k_1$.

Since $\phi_s$ is independent of $t$ and $u(s, \cdot)$ is independent of $x$

$$\partial_x^\lambda \partial_t^m \widetilde{u}(x, t) = \sum_{s \in \mathcal{S}} \partial_x^\lambda \phi_s(x) \partial_t^m u(s, t),$$

or in component-wise form, for every $k \in \mathbb{R}^d$, it holds:

$$\partial_x^\lambda \partial_t^m \widetilde{u}_k(x, t) = \sum_{s \in \mathcal{S}} \partial_x^\lambda \phi_s(x) \partial_t^m u_k(s, t).$$

Further, for all $s \in \mathcal{S}$, $x \in \mathbb{R}^d$ and $j \in [d]$, it holds

$$\left|\frac{\partial}{\partial x[j]} \phi_s(x)\right| = \left|\frac{2(x[j] - s[j])}{r^2} \eta'\left(\frac{\|x - s\|^2}{r^2}\right)\right|.$$

Then with $|s[j] - x[j]| \leq \frac{1}{e}$ when $x \in \text{supp}\{\phi_s(x)\}$, using mathematical induction we get that for all $s \in \mathcal{S}$, $x \in \mathbb{R}^d$ and $m \in \mathbb{Z}^+$, it holds

$$|\partial_x^\lambda \partial_t^m \phi_s(x)| \leq \frac{1}{r^{2m}} \cdot \left|\frac{d^m \eta}{dx^m}\left(\frac{\|x - s\|^2}{r^2}\right)\right|.$$

By Lemma D.1 and $r = 1/e$, we have $\left|\frac{d^m \eta}{dx^m}(x)\right| \leq 3e \cdot \left(\frac{2m}{e}\right)^{2m}$ and hence

$$\|\partial_x^\alpha \phi_s\|_{L^\infty(\mathbb{R}^d)} \leq 3e(2|\alpha|)^{2|\alpha|} \quad \text{for all} \quad s \in \mathcal{S}. \tag{D.7}$$

**Step 4: $\gamma$-Hölder Seminorm for Order $k_1$.** Return to the discussion of $\widetilde{u}_k$.

Let $|\lambda| + m = k_1$. For $(x, t_1), (y, t_2) \in \mathbb{R}^d \times [0,1]$ and for every $k \in [M^d]$, we have

$$\sum_{\|\lambda\|_1 + m \leq k_1} \|\partial_x^\lambda \widetilde{u}_k(x,t)\|_{L^\infty} \leq \sum_{\|\lambda\|_1 + m \leq k_1} \sum_{s \in \mathcal{S}} \|\partial_x^\lambda \phi_s(x) \partial_t^m u_k(s,t)\|_{L^\infty} \qquad \text{(By (D.6))}$$

$$\leq 3e \cdot (2k_1)^{2k_1} \sum_{m \leq k_1} \sum_{s \in \mathcal{S}} \|\partial_t^m u_k(s,t)\|_{L^\infty} \qquad \text{(By (D.7))}$$

$$= 3e \cdot (2k_1)^{2k_1} \sum_{s \in \mathcal{S}} \sum_{m \leq k_1} \|\partial_t^m u_k(s,t)\|_{L^\infty}$$

$$\text{(Interchange the order of summations)}$$

$$\leq 3e \cdot (2k_1)^{2k_1} K M^d, \qquad (D.8)$$

where the last line follows that $u(s,t) \in \mathcal{H}_{1,M^d}^\beta([0,1], K)$ with respect to $t$. Also, we have

$$\sum_{\|\lambda\|_1 + m = k_1} \sup_{\substack{(x,t_1),(y,t_2) \in \mathbb{R}^d \times [0,1] \\ (x,t_1) \neq (y,t_2)}} \frac{|\partial_x^\lambda \partial_t^m \widetilde{u}_k(x,t_1) - \partial_x^\lambda \partial_t^m \widetilde{u}_k(y,t_2)|}{\|(x,t_1) - (y,t_2)\|^\gamma}$$

$$= \sum_{\|\lambda\|_1 + m = k_1} \sup_{\substack{(x,t_1),(y,t_2) \in \mathbb{R}^d \times [0,1] \\ (x,t_1) \neq (y,t_2)}} \frac{|\sum_{s \in \mathcal{S}} \partial_x^\lambda \phi_s(x) \partial_t^m u_k(s,t_1) - \partial_x^\lambda \phi_s(y) \partial_t^m u_k(s,t_2)|}{\|(x,t_1) - (y,t_2)\|^\gamma} \qquad \text{(By (D.6))}$$

$$\leq 3e \cdot (2k_1)^{2k_1} M^d \sum_{n' \leq k_1} \sup_{t_1, t_2 \in [0,1], t_1 \neq t_2, s \in \mathcal{S}} \frac{|\partial_t^m u_k(s,t_1) - \partial_t^m u_k(s,t_2)|}{|t_1 - t_2|^\gamma} \qquad \text{(By (D.7))}$$

$$= 3e \cdot (2k_1)^{2k_1} M^d \cdot \Big( \sum_{n' = k_1} \sup_{t_1, t_2 \in [0,1], t_1 \neq t_2, s \in \mathcal{S}} \frac{|\partial_t^m u_k(s,t_1) - \partial_t^m u_k(s,t_2)|}{|t_1 - t_2|^\gamma}$$

$$+ \sum_{n' < k_1} \sup_{t_1, t_2 \in [0,1], t_1 \neq t_2, s \in \mathcal{S}} \frac{|\partial_t^m u_k(s,t_1) - \partial_t^m u_k(s,t_2)|}{|t_1 - t_2|^\gamma} \Big)$$

$$\leq 3e \cdot (2k_1)^{2k_1} M^d \cdot \Big( \sum_{n' = k_1} \sup_{t, t_1, t_2 \in [0,1], t_1 \neq t_2, s \in \mathcal{S}} \frac{|\partial_t^m u_k(s,t_1) - \partial_t^m u_k(s,t_2)|}{|t_1 - t_2|^\gamma}$$

$$+ \sum_{n' < k_1} \sup_{t_1, t_2 \in [0,1], t_1 \neq t_2, s \in \mathcal{S}} \frac{\|\partial^{n'+1} u_k(s,t)\|_{L^\infty} \cdot |t_1 - t_2|}{|t_1 - t_2|^\gamma} \Big)$$

$$\text{(By Newton-Leibniz formula)}$$

$$\leq 3e \cdot (2k_1)^{2k_1} M^d \cdot \Big( K + \sum_{n' < k_1} K \Big) \qquad (u(s,t) \in \mathcal{H}_{1,M^d}^\beta([0,1], K) \text{ and } |t_1 - t_2|^{1-\gamma} \leq 1)$$

$$= 3e \cdot (k_1 + 1)(2k_1)^{2k_1} K M^d. \qquad (D.9)$$

**Step 5: Altogether.** Combing (D.8) and (D.9), we get

$$\widetilde{u}(x,t) \in \mathcal{H}_{d+1, M^d}^\beta(\mathbb{R}^d \times [0,1], 3e \cdot (k_1 + 2)(2k_1)^{2k_1} K M^d).$$

This completes the proof. $\qquad \square$

# E    PROOF OF THEOREM 4.7

This section provides the proof of Theorem 4.7. We first develop auxiliary lemmas characterizing Lipschitz continuity properties and bound point-wise values of Lipschitz continuous functions using integral upper bounds. Building on these technical tools, we then present the proof of approximation theorem Theorem 4.7, which guarantees the existence of transformer networks that approximate the target function with controlled error and parameter bounds.

**Organization.** We recall basic concepts of factorized discrete flow matching and mixture path setting in Appendix E.1. Then we introduce and prove auxiliary lemmas in Appendix E.2. Finally, we present the main proof of Theorem 4.7 in Appendix E.3.

## E.1    PRELIMINARIES

To start with, recall from Section 2 that when constructing a factorized path, the probability path has a factorized generating velocity of the form

$$u_t(y, x) = \sum_i \delta(y^{\bar{i}}, x^{\bar{i}}), u_t^i(y^i, x), \tag{E.1}$$

where $\bar{i} = (1, \ldots, i-1, i+1, \ldots, d)$ denotes all indices except $i$. Following notations in Section 1, we write $u_t(\cdot, x)$ and $u_t^i(\cdot, x)$ as $u(x, t)$ and $u^i(x, t)$ respectively.

Next, recall that in Section 2 we construct mixture path $p_{t|0,1}(x|x_0, x_1) = \prod_i p_{t|0,1}^i(x^i|x_0, x_1)$, where each per-coordinate path interpolates between the source and target tokens:

$$p_{t|0,1}^i(x^i|x_0, x_1) = \kappa_t \delta(x^i, x_1^i) + (1 - \kappa_t)\delta(x^i, x_0^i).$$

Here, $\delta(\cdot, \cdot)$ is the Kronecker delta and $\kappa_t$ is a monotonically increasing smooth function that satisfies the boundary conditions:

$$\kappa_0 = 0, \quad \kappa_1 = 1, \quad \text{and} \quad \frac{d\kappa_t}{dt} > 0 \quad \text{for} \quad t \in (0, 1).$$

Then the corresponding conditional factorized velocity field that generates this per-coordinate path takes the form:

$$u_t^i(y^i, x^i|x_0^i, x_1^i) = \frac{\dot{\kappa}_t}{1 - \kappa_t}[\delta(y^i, x_1^i) - \delta(y^i, x^i)].$$

Taking expectation on $x_1$ and we obtain:

$$u_t^i(y^i, x) = \frac{\dot{\kappa}_t}{1 - \kappa_t} \mathbb{E}_{x_1 \sim P_1}[\delta(y^i, x_1^i) - \delta(y^i, x^i)]. \tag{E.2}$$

Further, we clip the time interval for training stability. Specifically, we focus on the time period $[t_0, T]$, where $0 < t_0 < T < 1$. This clipping is to prevent $\frac{\dot{\kappa}(t)}{1-\kappa(t)}$ from blowing up at $t = 0, 1$. We assume $\frac{\dot{\kappa}(t)}{1-\kappa(t)} = O(1)$ and $(\frac{\dot{\kappa}(t)}{1-\kappa(t)})' = O(1)$ in $t \in [t_0, T]$.

**Remark E.1.** *We demonstrate that clipping the interval of $t$ is necessary in discrete flow matching, for there doesn't exist a construction of $\kappa(t)$ that keeps stability of $\frac{\dot{\kappa}(t)}{1-\kappa(t)}$ at both $t = 0$ and $t = 1$. To show this, we set $r(t) = \frac{\dot{\kappa}(t)}{1-\kappa(t)}$ and $g(t) = 1 - \kappa(t)$. Then we have:*

$$(-\log(g(t))' = \frac{-g'(t)}{g(t)} = r(t).$$

*Since $\kappa(0) = 0$ and $\kappa(1) = 1$, we have $-\log g(0) = 0$ and $-\log g(1) = \infty$. This means $r(t) = (-\log(g(t))'$ is not bounded on [0,1]. Then for $\kappa(t)$ finite on (0,1), it must be instable at $t = 0$ or $t = 1$. Therefore, clipping the interval of $t$ is necessary.*

## E.2 Auxiliary Lemmas

In this section, we introduce auxiliary lemmas for the proof of Theorem 4.7. We adapt Lemma 4.5 to the mixture path setting, stated as Lemma E.2. In Lemma E.3 and Lemma E.4, we compute the Lipschitz constants of the functions constructed in these bridging lemmas. Finally, we establish connections between local Lipschitz behavior and integral bounds in Lemma E.5 and Lemma E.6.

To begin with, we introduce a lemma parallel to Lemma 4.5, guaranteeing the existence of a smooth function taking same value as a given discrete function at certain points.

**Lemma E.2** (Discrete-to-Continuous Functional Extension of Velocity under Mixture Path Setting, Modified from Lemma 4.5). *Consider velocity function $u(x,t)$ with the form* (E.2) *generating mixture path. For each $x \in \mathcal{S}$ and coordinate $i \in [d]$, let $t \mapsto u^i(x,t) \in \mathcal{H}_{1,M}^\beta([t_0,T],K)$ with $\beta = k_1 + \gamma \geq 1$, where $k = \lfloor \beta \rfloor$ and $\gamma \in [0,1)$. Then there exists an continuous extension $\widetilde{u}^i \in \mathcal{H}_{d+1,M}^\beta(\mathbb{R}^d \times [0,1], C)$ such that*

$$\widetilde{u}(s,t) = u(s,t) \quad \text{for all } s \in \mathcal{S}, t \in [t_0,T],$$

*where the Hölder norm $C = 3e \cdot (k_1+2)(2k_1)^{2k_1}KM$.*

*Proof.* The construction of $\widetilde{u}(x,t)$ is same as the construction in the proof of Lemma 4.5. We restate it for completeness. Let $\eta(t)$ be as defined in Lemma D.1. For $s \in \mathcal{S}$, we define $\phi_s(x)$ as:

$$\phi_s(x) = \eta(e^2 \|x - E(s)\|^2).$$

Then we construct $\widetilde{u}(x,t)$ as:

$$\widetilde{u}^i(x,t) = \sum_{s \in \mathcal{S}} \phi_s(x) \cdot u^i(s,t). \tag{E.3}$$

This construction takes the same form as in (D.6), while $u$ and $\widetilde{u}$ has output dimension of $M$ in mixture path case instead of $M^d$. Then $\widetilde{u}^i(s,t) = u^i(s,t)$ for every $s \in \mathcal{S}$ given that $\phi_s(s) = 1$.

The computation of Hölder constant of $\widetilde{u}^i(x,t)$ is exactly in the same form to the computation in proof of Lemma 4.5, while the only difference is to replace $M^d$ with $M$. $\square$

Next, we prove that when $u(x,t)$ is Lipschitz continuous with respect to $t$ for fixed $x$, $\widetilde{u}(x,t)$ we construct is Lipschitz continuous. Further, we give the Lipschitz constant of $\widetilde{u}(x,t)$ under $\ell_2$-norm. We first present the result under general case to increase readability.

**Lemma E.3** (Lipschitzness of Extension). *Suppose that for every given $s \in \mathcal{S}$, it holds $\|u(s,t)\|_2 \leq M_u$ and $u(s,t)$ is Lipschitz continuous under $\ell_2$-norm with respect to $t$, with Lipschitz constant $L_u$. Then $\widetilde{u}(x,t)$ defined in the proof of Lemma 4.5 is Lipschitz continuous under $\ell_2$-norm with respect to $(x,t)$, with Lipschitz constant $\max\{L_u, 4e\sqrt{d}M_u\}$.*

*Proof.* By letting $n = 1$ in (D.7), we get $|\partial \phi_s| \leq 4e$. This indicates $\|\nabla \phi_s\|_2 \leq 4e\sqrt{d}$, meaning that $\phi_s$ is is Lipschitz continuous under $\ell_2$-norm, with Lipschitz constant $4e\sqrt{d}$.

By definition, for $s_1 \neq s_2 \in \mathcal{S}$, it holds $\|s_1 - s_2\| \geq 1$. Then $B(s_1, \frac{1}{e})$ and $B(s_2, \frac{1}{e})$ does not intersect for distinct $s_1, s_2$. Therefore for $x \in [0,M]^d$, there is at most one $s \in \mathcal{S}$ such that $x \in B(s, \frac{1}{e})$. For $(x_1,t_1), (x_2,t_2) \in \mathbb{R}^d \times \mathbb{R}$, we discuss two possible cases as below.

(1). There exists $s_0 \in \mathcal{S}$, such that $x_1, x_2 \in B(s_0, \frac{1}{e})$.

Then it holds:

$$\begin{aligned}
\|\widetilde{u}(x_1,t_1) - \widetilde{u}(x_2,t_2)\|_2 &= \|\sum_{s \in \mathcal{S}} \phi_s(x_1) \cdot u(s,t_1) - \sum_{s \in \mathcal{S}} \phi_s(x_2) \cdot u(s,t_2)\|_2 \\
&= \|\phi_{s_0}(x_1) \cdot u(s_0,t_1) - \phi_{s_0}(x_2) \cdot u(s_0,t_2)\|_2 \quad {\scriptstyle(\phi_s(x) = 0 \text{ for } x \notin B(s,\frac{1}{e}))} \\
&\leq \|\phi_{s_0}(x_1) \cdot u(s_0,t_1) - \phi_{s_0}(x_1) \cdot u(s_0,t_2)\|_2 \\
&\quad + \|\phi_{s_0}(x_1) \cdot u(s_0,t_2) - \phi_{s_0}(x_2) \cdot u(s_0,t_2)\|_2 \\
&\leq \|u(s_0,t_1) - u(s_0,t_2)\|_2 + M_u\|\phi_{s_0}(x_1) - \phi_{s_0}(x_2)\|_2 \\
&\qquad\qquad {\scriptstyle(\phi_s(x) \leq 1, \|u(s,t)\|_2 \leq M_u)}
\end{aligned}$$

$$\leq L_u \|t_1 - t_2\|_2 + 4e\sqrt{d}M_u \|x_1 - x_2\|_2$$
$$\leq \max\{L_u, 4e\sqrt{d}M_u\} \|(x_1, t_1) - (x_2, t_2)\|_2.$$

(2). For all $s \in \mathcal{S}$, $x_1$ and $x_2$ do not both belong to $B(s, \frac{1}{e})$.

Then $\|x_1 - x_2\| \geq 1 - \frac{2}{e}$. Therefore we have:

$$\|u(x_1, t_1) - u(x_2, t_2)\|_2 \leq 2M_u \leq \frac{2e}{e-2}M_u\|x_1 - x_2\|_2 \leq \frac{2e}{e-2}M_u\|(x_1, t_1) - (x_2, t_2)\|_2.$$

Since $4e\sqrt{d}M_u \geq \frac{2e}{e-2}M_u$ this completes the proof. $\qquad\square$

Next, we introduce a lemma modified from Lemma E.3. This lemma computes the upper bound of $\widetilde{u}^i(x, t)$ for $u^i(x, t)$ with the form (E.2) generating mixture path.

**Lemma E.4** (Lipschitzness of Extension under Mixture Path Setting, Modified from Lemma E.3)**.** *Consider $u^i(x, t)$ under mixture path setting with the form* (E.2) *. Then $\widetilde{u}^i(x, t)$ comstructed in the proof of Lemma E.2 is Lipschitz continuous under $\ell_2$-norm with respect to $(x, t)$ when $t \in [t_0, T]$, with Lipschitz constant $L_{\widetilde{u}} \lesssim 1$.*

*Proof.* Recall that under mixture path setting $M_u, L_u = O(1)$. Substituting $M_u$ and $L_u$ with $O(1)$ in conclusion of Lemma E.3 and we get the result for mixture path case. $\qquad\square$

Observing that for Lipschitz continuous function, its Lipschitz constant gives an upper bound on how fast a function increases or decreases. This leads to the following lemma, connecting a function's local value to its integral's value lower bound.

**Lemma E.5** (Integral Lower Bound via Point-wise Magnitude)**.** *Suppose that $f : \mathbb{R}^{d \times L} \to \mathbb{R}^{d \times L}$ is Lipschitz continuous under Frobenius norm, with Lipschitz constant $L_f$. Let $n = dL$. If there exists $X \in \mathbb{R}^{d \times L}$ such that $\|f(X)\|_F \geq a > 0$, then it holds:*

$$(\int \|f(Z)\|_F^2 dZ)^{1/2} \geq (\frac{2S_n}{n(n+1)(n+2)})^{1/2} \frac{a^{\frac{n+2}{2}}}{L_f^{\frac{n}{2}}},$$

*where $S_n$ denote the surface area of the unit sphere in $n$-dimensional Euclidean space.*

*Proof.* For $Z$ such that $\|Z - X\|_F \leq \frac{a}{L_f}$, it holds:

$$\|f(Z)\|_F \geq a - L_f\|X - Z\|_F.$$

Let $S_n$ denote the surface area of the unit sphere in $n$-dimensional Euclidean space. We have:

$$\begin{aligned}
(\int \|f(Z)\|_F^2 dZ)^{1/2} &\geq (\int_{\|Z-X\|_F \leq \frac{a}{L}} (a - L_f\|X - Z\|_F)^2 dZ)^{1/2} \\
&= (\int_{\|Z\|_F \leq \frac{a}{L}} (a - L_f\|Z\|_F)^2 dZ)^{1/2} && \text{(By change of variable)} \\
&= (\int_0^{a/L} (a - L_f r)^2 S_n r^{n-1} dr)^{1/2} \\
&= (\frac{2S_n a^{n+2}}{n(n+1)(n+2)L_f^n})^{1/2} && \text{(By integration)} \\
&= (\frac{2S_n}{n(n+1)(n+2)})^{1/2} \frac{a^{\frac{n+2}{2}}}{L_f^{\frac{n}{2}}}.
\end{aligned}$$

This completes the proof. $\qquad\square$

With Lemma E.5 we have the following lemma bounding local value with integral value.

**Lemma E.6** (Point-wise Upper Bound via Integral Constraint). *Suppose that $f : [-I, I]^{d \times L} \to \mathbb{R}^{d \times L}$ is Lipschitz continuous on a bounded domain under Frobenius norm, with Lipschitz constant $L_f$. Let $n = dL$. Let $(\int \|f(Z)\|_F^2 dZ)^{1/2} \le b$, then for all $Z \in [-I, I]^{d \times L}$ it holds:*

$$\|f(Z)\|_F \lesssim b^{\frac{2}{n+2}} L_f^{\frac{n}{n+2}}.$$

*Proof.* We obtain the conclusion by rearranging the inequality in conclusion of Lemma E.5 and ignoring constants. $\square$

### E.3 MAIN PROOF OF THEOREM 4.7

In this section, we prove the approximation theorem for discrete flow matching under the mixture path and factorized velocity settings. Notice that in the proof of this theorem, we treat the upper bound of the Lipschitz constant of the approximator Transformer class as a constant independent of $\epsilon$. Also, in the main text we present a simplified version Theorem 4.7 in order to keep the exposition concise, while Theorem E.7 stated below is the explicit form.

**Theorem E.7** ( Approximation Theorem for Mixture Path Discrete Flow Matching, Theorem 4.7 Restate). *Let $u^i(x, t)$ be the factorized velocity field for coordinate $i \in [d]$ under mixture path setting. Assume Assumption 4.3 holds, then for any $\epsilon \in (0, 1)$, there exists a transformer network $u_\theta^i(x, t) \in \mathcal{T}_R^{h,s,r}(C_{\mathcal{T}}, C_{KQ}^{2,\infty}, C_{KQ}, C_{OV}^{2,\infty}, C_{OV}, C_E, C_F^{2,\infty}, C_F)$ satisfying that for any $t \in [t_0, T]$:*

$$\sum_{x \in \mathcal{S}} \|u_\theta^i(x, t) - u^i(x, t)\|_2^2 \cdot p_t(x) \lesssim \epsilon^{\frac{4}{M+2}} M^{\frac{12Md_0 + 25M}{M+2}},$$

*where $d_0$ is the transformer feature dimension. The parameter bound of the transformer network class follows:*

$$C_{KQ}, C_{KQ}^{2,\infty} = \widetilde{O}(M^{6d_0+3} \epsilon^{-4d_0-2}); C_{OV}, C_{OV}^{2,\infty} = O(M^{-\frac{1}{2}} \epsilon)$$

$$C_F, C_F^{2,\infty} = O(M^2 \epsilon^{-1}); C_E = O(M),$$

*where $O(\cdot)$ hides polynomial factors depending on $d, d_0$, $\widetilde{O}(\cdot)$ hides polynomial factors depending on $d, d_0$ and logarithmic factors depending on $M$.*

*Proof.* To begin with, we introduce reshape layer we use in the proof.

While ordinary transformer network approximates function with same input and output dimension, under factorized path setting we need to approximate function $\widetilde{u}(x, t)$, with input dimension $d + 1$ and output dimension $M$. To accommodate this difference, we introduce two reshape layers: $R_1$ and $R_2$ to facilitate the transformation of dimensions. We assume $d + 1 | M$ for simplicity in discussions below.

- $R_1 : [0, M]^d \times [0, 1] \to \mathbb{R}^{d_0 \times \frac{M}{d_0}}$ is a reshape function rearranging a vector of dimension $d + 1$ into a matrix of size $\mathbb{R}^{d_0 \times \frac{M}{d_0}}$, where transformer feature dimension $d_0$ satisfies $d_0 | d + 1$. We realize $R_1$ by first reshaping input vector $(x, t)$ with dimension $d + 1$ into a matrix $A \in \mathbb{R}^{d_0 \times \frac{d+1}{d_0}}$, following the standard procedure of rearranging entries. Then we replicate the matrix $\frac{M}{d+1}$ times along its columns, yielding a matrix of size $d_0 \times \frac{M}{d_0}$. Altogether, the output of $R_1$ is a matrix of size $d_0 \times \frac{M}{d_0}$. As the reverse of $R_1$, $R_1^{-1} : \mathbb{R}^{d_0 \times \frac{M}{d_0}} \to [0, M]^d \times [0, 1]$ is defined by taking first $\frac{d+1}{d_0}$ columns of the matrix, and then rearranging it into a vector of dimension $d + 1$.

- $R_2 : \mathbb{R}^{d_0 \times \frac{M}{d_0}} \to \mathbb{R}^M$ is a reshape function rearranging a matrix of size $\mathbb{R}^{d_0 \times \frac{M}{d_0}}$ into a vector of dimension $M$, where $d_0 | d + 1$. We realize $R_2$ by rearranging the entries of the matrix into a vector preserving the total number of elements, following standard reshape layer construction. We define $R_2^{-1}$ as the reverse map of $R_2$. This is well-defined since $R_2$ is bijection between $\mathbb{R}^{d_0 \times \frac{M}{d_0}}$ and $\mathbb{R}^M$.

It is important to note that these reshape layers do not participate in the learning process of the Discrete Flow Matching, and therefore, are not the main focus of our discussion. $R_1$ and $R_2$ represent a feasible design for the reshape layers, but they are not unique construction to make up for the dimension difference. We present these particular forms of $R_1$ and $R_2$ for clarity and completeness, but the core of our discussion does not depend on them, and readers should avoid overemphasizing these details.

We now return to the main proof. Let $\widetilde{u}(x,t)$ be as defined in Lemma E.2. For any coordinate $i \in [d]$, let reshaped factorized velocity field $u^{i,\mathrm{reshape}} : \mathbb{R}^{d_0 \times \frac{M}{d_0}} \to \mathbb{R}^{d_0 \times \frac{M}{d_0}}$ be:

$$u^{i,\mathrm{reshape}} = R_2^{-1} \circ \widetilde{u}(x,t) \circ R_1^{-1}.$$

Then $u^{i,\mathrm{reshape}}$ is Lipschitz continuous under Frobenius norm, with Lipschitz constant no larger than Lipschitz constant of $\widetilde{u}$. By Proposition B.17, for any $\epsilon$ there exists a

$$u_\theta^{i,\mathrm{reshape}}(Z) = F_1^{\mathrm{FF}} \circ F^{\mathrm{SA}} \circ F_2^{\mathrm{FF}} \circ F^{\mathrm{E}} \in \mathcal{T}^{h,s,r}(C_{\mathcal{T}}, C_{KQ}^{2,\infty}, C_{KQ}, C_{OV}^{2,\infty}, C_{OV}, C_E, C_F^{2,\infty}, C_F),$$

such that $d_F(u^{i,\mathrm{reshape}}(Z), u_\theta^{i,\mathrm{reshape}}(Z)) < \epsilon$, where $d_F(f(Z), g(Z)) := (\int \|f(Z) - g(Z)\|_F^2 \mathrm{d}Z)^{1/2}$. By Theorem B.18, the parameter bound of the transformer network satisfies:

$$C_{KQ}, C_{KQ}^{2,\infty} = \widetilde{O}(M^{6d_0+3}\epsilon^{-4d_0-2}); C_{OV}, C_{OV}^{2,\infty} = O(M^{-\frac{1}{2}}\epsilon)$$
$$C_F, C_F^{2,\infty} = O(M^2\epsilon^{-1}); C_E = O(M), \tag{E.4}$$

where $O(\cdot)$ hides polynomial factors depending on $d, d_0$, $\widetilde{O}(\cdot)$ hides polynomial factors depending on $d, d_0$ and logarithmic factors depending on $M$.

Let $h = u_\theta^{i,\mathrm{reshape}} - u^{i,\mathrm{reshape}}$. Then $(\int \|h\|_F^2 \mathrm{d}Z)^{1/2} < \epsilon$. By Lemma B.11 and Lemma E.4, $u_\theta^{i,\mathrm{reshape}}, u^{i,\mathrm{reshape}}$ are Lipschitz continuous under Frobenius norm, indicating that $h$ is Lipschitz continuous under Frobenius norm. We denote their Lipschitz constant as $L_\theta^{i,\mathrm{reshape}}, L^{i,\mathrm{reshape}}$ and $L_h$ respectively.

We first compute $L_\theta^{i,\mathrm{reshape}}$ according to Lemma B.11:

$$L_\theta^{i,\mathrm{reshape}} \leq (1 + 2M^2 C_{OV} C_{KQ} + h\frac{M}{d_0}C_{OV}) \cdot (C_F^2 + 1)^2$$
$$\lesssim (C_F)^2 \cdot M^2 \cdot hC_{KQ}C_{OV} \cdot (C_F)^2$$
$$\lesssim M^{6d_0+\frac{25}{2}}. \qquad \text{(By (E.4). Note that we drop terms od } \epsilon.)$$

Then we compute the expression of $L_h$:

$$L_h \leq L_\theta^{i,\mathrm{reshape}} + L^{i,\mathrm{reshape}}$$
$$\lesssim M^{6d_0+\frac{25}{2}}. \qquad \text{(By Lemma E.4)}$$

Let $u_\theta := R_2 \circ u_\theta^{i,\mathrm{reshape}} \circ R_1$. For all $(x,t) \in [0,M]^d \times [t_0, T]$, it holds:

$$\|u_\theta^i(x,t) - \widetilde{u}^i(x,t)\|_2^2 \leq \|u_\theta^{i,\mathrm{reshape}} \circ R_1(x,t) - u^{i,\mathrm{reshape}} \circ R_1(x,t)\|_F^2$$
$$= \|h(R_1(x,t))\|_F^2 \qquad \text{(By definition of } h)$$
$$\lesssim \epsilon^{\frac{4}{M+2}} M^{\frac{12Md_0+25M}{M+2}}. \qquad \text{(By Lemma E.6)}$$

Then for every $t \in [t_0, T]$, we have:

$$\sum_{x \in \mathcal{S}} \|u_\theta^i(x,t) - u^i(x,t)\|_2^2 \cdot p_t(x) \lesssim \epsilon^{\frac{4}{M+2}} M^{\frac{12Md_0+25M}{M+2}} \sum_{x \in \mathcal{S}} p_t(x) = \epsilon^{\frac{4}{M+2}} M^{\frac{12Md_0+25M}{M+2}}.$$

This completes the proof. $\qquad \square$

**Remark E.8.** *Our choice of the Transformer architecture is motivated by its widespread practical adoption in modern generative modeling. Transformers is the dominant backbone for parameterizing velocity (or score) networks in state-of-the-art generative models, including Diffusion Transformers (Peebles & Xie, 2023), MaskFlow (Fuest et al., 2025), and DeFoG (Yi et al., 2025). However, the theorem is not restricted to Transformer architectures. It extends to any network architecture that owns a universal approximation theorem similar to Proposition B.17. For example, multi-layer perceptrons (MLP) also satisfy a universal approximation property (Lemma B.5 of (Fu et al., 2024)). By replacing the Transformer universal approximation lemma Proposition B.17 with the corresponding MLP result, deriving velocity approximation theorem with MLP is straightforward.*

# F    PROOF OF THEOREM 5.1

This section provides the proof of Theorem 5.1, deriving velocity estimation rate for factorized discrete flow matching implemented with transformers under mixture path setting. The analysis adapts and modifies the risk-decomposition plus covering-number technique of (Fu et al., 2024) to our setting and parameter bounds from Theorem 4.7.

**Organization.** We derive the estimation rates of discrete flow matching transformers in four steps.

- **Preliminaries.** In Appendix F.1, we introduce several essential concepts, including factorized empirical loss $\widehat{\mathcal{L}}_{\mathrm{CDFM}}^{i_0}$, factorized discrete flow matching risk $\mathcal{R}^{i_0}(\Theta^{i_0})$ and factorized empirical risk $\widehat{\mathcal{R}}^{i_0}(\Theta^{i_0})$.

- **Covering Number Upper bound.** We obtain covering-number bounds for the transformer class using the parameter constraints from Theorem 4.7 and for the induced loss class in Appendix F.2, Lemma F.3-Lemma F.5.

- **Generalization and Approximation Error Bound.** We apply the covering-number machinery and conclusions from Theorem 4.7 to bound generalization and approximation error in Appendix F.2, Lemmas F.6 and F.7.

- **Velocity Estimation rates.** We utilize conclusions from prior three steps to prove Theorem 5.1, the velocity estimation rate.

## F.1    PRELIMINARIES

In this section, we introduce and discuss basic concepts risk function of factorized flow matching.

In factorized velocity discrete flow matching case, for $i_0 \in [M]$, $u_\theta^{i_0}$ is trained to approximate $u^{i_0}(x, t)$ solely, independent of the behaviour of velocity field $u(x, t)$ on other $d-1$ dimensions. In other words, given $i_0 \in [d]$ and $n$ i.i.d training samples $\{x_i\}_{i=1}^n$, the transformer network $u_\theta^{i_0}$ is trained through minimizing the factorized empirical loss:

$$\widehat{\mathcal{L}}_{\mathrm{CDFM}}^{i_0} := \frac{1}{n} \sum_{i=1}^n \int_{t_0}^T \mathop{\mathbb{E}}_{X_0 \sim p_0, X_t \sim p_{t|x_0=X_0, x_1=x_i}} \|u^{i_0}(X_t, t) - u_\theta^{i_0}(X_t, t)\|_2^2 \mathrm{d}t.$$

For simplicity of expression, we define the loss of certain function $f$ with respect to some certain end point $x$ as:

$$\ell^{i_0}(x; f) := \int_{t_0}^T \mathop{\mathbb{E}}_{X_0 \sim p_0, X_t \sim p_{t|x_0=X_0, x_1=x}} \|u^{i_0}(X_t, t) - f(X_t, t)\|_2^2 \mathrm{d}t.$$

Then we have:

$$\widehat{\mathcal{L}}_{\mathrm{CDFM}}^{i_0} = \frac{1}{n} \sum_{i=1}^n \ell^{i_0}(x_i; u_\theta^{i_0}).$$

We use $\widehat{\Theta}^{i_0}$ to denote the parameter of network trained by minimizing the factorized empirical loss $\widehat{\mathcal{L}}_{\mathrm{CDFM}}^{i_0}$ with $n$ i.i.d training samples $\{x_i\}_{i=1}^n$. That's to say, discrete flow matching network $\widehat{u}_\theta^{i_0}$ with parameter $\widehat{\Theta}^{i_0}$ is the factorized empirical risk minimizer, satisfying $\widehat{\Theta}^{i_0} \in \mathop{\mathrm{argmin}}_{\Theta^{i_0}} \widehat{\mathcal{L}}_{\mathrm{CDFM}}(u_\theta^{i_0})$.

For a factorized discrete flow matching network $u_\theta^{i_0}$ with parameter $\Theta^{i_0}$, its performance in velocity estimation is measured by the factorized discrete flow matching risk, which is defined as:

$$\mathcal{R}^{i_0}(\Theta) := \int_{t_0}^T \mathop{\mathbb{E}}_{X_t \sim p_t} \|u^{i_0}(X_t, t) - u_\theta^{i_0}(X_t, t)\|_2^2 \mathrm{d}t. \tag{F.1}$$

In practice, we evaluate the performance of the factorized network $u_\theta^{i_0}$ using factorized empirical discrete flow matching risk, which is defined as:

$$\widehat{\mathcal{R}}^{i_0}(\Theta^{i_0}) := \frac{1}{n} \sum_{i=1}^n \ell^{i_0}(x_i; u_\theta^{i_0}) - \frac{1}{n} \sum_{i=1}^n \ell^{i_0}(x_i; u^{i_0}), \tag{F.2}$$

where $\{x_i\}_{i=1}^n$ are $n$ i.i.d samples and $u^{i_0}$ is the true velocity. We have the following lemma showing that factorized discrete flow matching risk is equal to the expectation of factorized empirical discrete flow matching risk.

**Lemma F.1** (Modified from Remark E.2 of (Su et al., 2025)). *For a factorized discrete flow matching network $u_\theta^{i_0}(x,t)$ with parameters noted as $\Theta^{i_0}$ and i.i.d samples $\{x_i\}_{i=1}^n$, it holds:*

$$\mathbb{E}_{\{x_i\}_{i=1}^n}[\widehat{\mathcal{R}}^{i_0}(\Theta^{i_0})] = \mathcal{R}^{i_0}(\Theta^{i_0}).$$

*Proof.* The proof is modified from Remark E.2 of (Su et al., 2025).

We use $\Theta^{i_0,\text{true}}$ to denote the true parameter of velocity function. That's to say, with parameter $\Theta^{i_0,\text{true}}$ it holds $u_\theta^{i_0} = u^{i_0}$. Then we have:

$$\begin{aligned}
\mathbb{E}_{\{x_i\}_{i=1}^n}[\widehat{\mathcal{R}}^{i_0}(\Theta^{i_0})] &= \mathbb{E}_{\{x_i\}_{i=1}^n}[\frac{1}{n}\sum_{i=1}^n \ell^{i_0}(x_i; u_\theta^{i_0})] - \mathbb{E}_{\{x_i\}_{i=1}^n}[\frac{1}{n}\sum_{i=1}^n \ell^{i_0}(x_i; u^{i_0})] &&\text{(By (F.2))}\\
&= \mathcal{L}_{\text{CDFM}}^{i_0}(\Theta^{i_0}) - \mathcal{L}_{\text{CDFM}}^{i_0}(\Theta^{i_0,\text{true}}) \\
&= \mathcal{R}^{i_0}(\Theta^{i_0}) - \mathcal{R}^{i_0}(\Theta^{i_0,\text{true}}) &&\text{(By gradient equivalence of } \mathcal{L}_{\text{DFM}}^{i_0} \text{ and } \mathcal{L}_{\text{CDFM}}^{i_0})\\
&= \mathcal{R}^{i_0}(\Theta^{i_0}). &&(\mathcal{R}^{i_0}(\Theta^{i_0,\text{true}}) = 0)
\end{aligned}$$

This completes the proof. $\qquad\square$

## F.2  Auxiliary Lemmas

To bound $\mathbb{E}_{\{x_i\}_{i=1}^n}[\mathcal{R}^{i_0}(\widehat{\Theta}^{i_0})]$, we modify the risk decomposition approach formulated in (Fu et al., 2024) to discrete flow matching case. Specifically, we have:

$$\mathbb{E}_{\{x_i\}_{i=1}^n}[\mathcal{R}^{i_0}(\widehat{\Theta}^{i_0})] = \underbrace{\mathbb{E}_{\{x_i\}_{i=1}^n}[\mathcal{R}^{i_0}(\widehat{\Theta}^{i_0}) - \widehat{\mathcal{R}^{i_0}}(\widehat{\Theta}^{i_0})]}_{\text{(I)}} + \underbrace{\mathbb{E}_{\{x_i\}_{i=1}^n}[\widehat{\mathcal{R}}^{i_0}(\widehat{\Theta}^{i_0})]}_{\text{(II)}}, \qquad \text{(F.3)}$$

In this section, we introduce auxiliary lemmas helping us prove Theorem 5.1. Specifically, we compute the covering number bound of transformers in Lemma F.3 and Lemma F.4. Further, we obtain the covering number of loss function class in Lemma F.5. Finally, Lemma F.6 and Lemma F.7 bounds (I) and (II) in (F.3) respectively.

To begin with, we introduce the definition of covering number, a concept that plays a fundamental role in establishing bounds on (I), the generalization error.

**Definition F.2** (Covering Number). *Consider a vector-valued function class $\mathcal{F}$. For $\epsilon > 0$, a point set $\{z_i\}_{i=1}^n$ and a norm $\|\cdot\|$, the quantity $\mathcal{N}_\infty(\mathcal{F}; \epsilon; \{z_i\}_{i=1}^n; \|\cdot\|)$ denotes the minimal cardinality of a subset (a cover) $\mathcal{C} \subset \mathcal{F}$ such that, for any $f \in \mathcal{F}$, there exists $\widehat{f} \in \mathcal{C}$ such that:*

$$\max_{1\le i\le n} \|f(z_i) - \widehat{f}(z_i)\| \le \epsilon.$$

*We call $\mathcal{N}_\infty(\mathcal{F}; \epsilon; \{z_i\}_{i=1}^n; \|\cdot\|)$ the $\epsilon$-covering number of $\mathcal{F}$ with respect to point set $\{z_i\}_{i=1}^n$. We further set:*

$$\mathcal{N}_\infty(\mathcal{F}; \epsilon; m; \|\cdot\|) = \max_{\{z_i\}_{i=1}^m} \mathcal{N}_\infty(\mathcal{F}; \epsilon; \{z_i\}_{i=1}^m; \|\cdot\|).$$

Next, we introduce the following lemma that gives an upper bound on the covering number of multiple-layer transformer network.

**Lemma F.3** (Lemma J.2 of (Hu et al., 2024b), Modified from Theorem A.17 of (Edelman et al., 2022)). *Let $\mathcal{T}_R^{h,s,r}(C_{\mathcal{T}}, C_{KQ}^{2,\infty}, C_{KQ}, C_{OV}^{2,\infty}, C_{OV}, C_E, C_F^{2,\infty}, C_F, L_{\mathcal{T}})$ represent the class of transformer network with parameter bound. Then for data points $x$ such that $\|x\|_2 \le B_X$, we have:*

$$\log \mathcal{N}(\mathcal{T}_R^{h,s,r}, \epsilon, n, \|\cdot\|_2)$$

$$\le \frac{\log(nL_{\mathcal{T}})}{\epsilon^2}\alpha^2(d_0^{\frac{2}{3}}(C_F^{2,\infty})^{\frac{2}{3}} + d_0^{\frac{2}{3}}(2(C_F)^2 C_{OV} C_{KQ}^{2,\infty})^{\frac{2}{3}} + 2((C_F)^2 C_{OV}^{2,\infty})^{\frac{2}{3}})^3,$$

*where $\alpha = (C_F)^2 C_{OV}(1 + 4C_{KQ})(B_X + C_E)$.*

*Proof.* See Remark J.7 of (Hu et al., 2024b) and proof of Lemma A.17 of (Edelman et al., 2022). □

Equipped with Lemma F.3, we compute the covering number of transformer network class with parameter bound given in Theorem 4.7.

**Lemma F.4** (Covering Number Bound for Transformer Class, Modified from Lemma J.3 of (Hu et al., 2024b)). *Let $\epsilon_c > 0$. Consider the transformer class $\mathcal{T}_R^{h,s,r}(C_{\mathcal{T}}, C_{KQ}^{2,\infty}, C_{KQ}, C_{OV}^{2,\infty}, C_{OV}, C_E, C_F^{2,\infty}, C_F)$ with parameter bound given in Theorem E.7 and $x_i$ satisfying $x_i \in \mathcal{S}$. Then the $\epsilon_c$-covering number of $\mathcal{T}_R^{h,s,r}$ has the following upper bound:*

$$\log \mathcal{N}(\mathcal{T}_R^{h,s,r}, \epsilon_c, n, \|\cdot\|_2) \lesssim \frac{\log(nM)}{\epsilon_c^2} M^{24d_0+28} \epsilon^{-16d_0-12}.$$

*Proof.* The proof is modified from proof of Lemma J.2 of (Hu et al., 2024b).

From Theorem E.7, we have the bounds on transformer parameters:

$$C_{KQ}, C_{KQ}^{2,\infty} = \widetilde{O}(M^{6d_0+3}\epsilon^{-4d_0-2}); C_{OV}, C_{OV}^{2,\infty} = O(M^{-\frac{1}{2}}\epsilon)$$

$$C_F, C_F^{2,\infty} = O(M^2\epsilon^{-1}); C_E = O(M); L_{\mathcal{T}} = O(M^{6d_0+\frac{25}{2}}). \tag{F.4}$$

We first use parameter in (F.4) to compute $\alpha$ and get:

$$\alpha \lesssim (M^2\epsilon^{-1})^2 \cdot M^{-\frac{1}{2}}\epsilon \cdot M^{6d_0+3}\epsilon^{-4d_0-2} \cdot M = M^{6d_0+\frac{15}{2}}\epsilon^{-4d_0-3}.$$

Further, by Lemma F.3 we have:

$$\log \mathcal{N}(\mathcal{T}_R^{h,s,r}, \epsilon_c, n, \|\cdot\|_2)$$
$$\lesssim \frac{\log(nL_{\mathcal{T}})}{\epsilon_c^2} \alpha^2 (d_0^{\frac{2}{3}}(2(C_F)^2 C_{OV} C_{KQ}^{2,\infty})^{\frac{2}{3}})^3$$
$$\lesssim \frac{\log(nM)}{\epsilon_c^2} M^{24d_0+28} \epsilon^{-16d_0-12}.$$

This completes the proof. □

Then we compute the covering number bound of loss function class by bounding error of loss function with error of transformer.

**Lemma F.5** (Covering Number Bound for Loss Function Class, Modified from Lemma L.3 of (Su et al., 2025)). *Let $\epsilon_c > 0$ and $i_0 \in [M]$. Suppose that for every given $x \in \mathcal{S}$, $u^{i_0}(x, t)$ represent the velocity field of $x$ at time $t$ that follows mixture path setting (E.2). We define the factorized loss function class by*

$$F_{\text{loss}}^{i_0} := \{\ell^{i_0}(x; u_\theta^{i_0}) | u_\theta^{i_0} \in \mathcal{T}_R^{h,s,r}\},$$

*where $\mathcal{T}_R^{h,s,r}$ is the transformer class with parameter bound given in Theorem 4.7.*

*Then we have:*

$$\log \mathcal{N}(F_{\text{loss}}^{i_0}, \epsilon_c, \{x_i\}_{x_i \in \mathcal{S}}, |\cdot|) \lesssim \frac{\log(M) - \log(\epsilon_c)}{\epsilon_c^2} M^{24d_0+28} \epsilon^{-16d_0-12}.$$

*Proof.* The proof is modified from the proof of Lemma L.3 of (Su et al., 2025).

Consider $i_0 \in [d]$ and $\{x_i\}_{i=1}^n \in \mathcal{S}$. Let $u_1^{i_0}(x, t), u_2^{i_0}(x, t)$ be mixture path velocity function satisfying $\|u_1^{i_0}(x, t) - u_2^{i_0}(x, t)\| \le \delta$ for all $x \in \mathcal{S}$ and $t = 0, \frac{1}{\lceil \frac{L_{\mathcal{T}}}{\delta} \rceil}, \frac{2}{\lceil \frac{L_{\mathcal{T}}}{\delta} \rceil}, \dots, 1$. Then since $u_1^{i_0}$ and $u_2^{i_0}$ is Lipschitz continuous with Lipschitz constant $L_{\mathcal{T}}$ under $\ell_2$-norm, for $x \in \mathcal{S}$ and $t \in [0, 1]$ we have $\|u_1^{i_0}(x, t) - u_2^{i_0}(x, t)\| \le 2\delta$.

Further, for $x = x_i, 1 \le i \le n$ we have:

$$|\ell(^{i_0}x; u_1^{i_0}) - \ell^{i_0}(x; u_2^{i_0})|$$

$$= |\int_{t_0}^{T} \mathbb{E}_{X_0 \sim p_0, X_t \sim p_{t|x_0=X_0,x_1=x}} (\|u_1^{i_0}(X_t,t) - u^{i_0}(X_t,t)\|_2^2 - \|u_2^{i_0}(X_t,t) - u^{i_0}(X_t,t)\|_2^2)dt|$$

$$= |\int_{t_0}^{T} \mathbb{E}_{X_0 \sim p_0, X_t \sim p_{t|x_0=X_0,x_1=x}} (u_1^{i_0}(X_t,t) - u_2^{i_0}(X_t,t))^\top (u_1^{i_0}(X_t,t) + u_2^{i_0}(X_t,t) - 2u^{i_0}(X_t,t)))dt|$$

$$\le 2\delta \int_0^1 \mathbb{E}_{X_0 \sim p_0, X_t \sim p_{t|x_0=X_0,x_1=x}} \|u_1^{i_0}(X_t,t) + u_2^{i_0}(X_t,t) - 2u^{i_0}(X_t,t)\|_2 dt \qquad {\scriptstyle (\|u_1^{i_0} - u_2^{i_0}\|_2 \le 2\delta)}$$

$$\le 8\delta C_{\mathcal{T}},$$

where the last line is by assuming $C_{\mathcal{T}} \ge \max_{t \in [t_0,T]} \frac{2\dot{\kappa}(t)}{1-\kappa(t)}$ without losing generality.

From computation above, for every function class $\mathcal{U}$ being a $\epsilon_c$-covering of $\mathcal{T}_R^{h,s,r}$ with respect to point set $S$, function class $\mathcal{L} = \{\ell(x;u)|u \in \mathcal{U}\}$ is a $4\epsilon_c C_{\mathcal{T}}$-covering of $F_{\text{loss}}^{i_0}$. Recall that we assume $\frac{\dot{\kappa}(t)}{1-\kappa(t)} = O(1)$ when $t \in [t_0,T]$ in Appendix E.2. Then, for small enough $\epsilon$ it holds that $C_{\mathcal{T}} = O(1)$. We further obtain:

$$\log \mathcal{N}(F_{\text{loss}}^{i_0}, \epsilon_c, \{x_i\}_{x_i \in S}, |\cdot|) \le \log \mathcal{N}(\mathcal{T}_R^{h,s,r}, \frac{\epsilon_c}{8C_{\mathcal{T}}}, M^d(\lceil \frac{L_{\mathcal{T}}}{\epsilon_c} \rceil + 1), |\cdot|)$$

$$\lesssim \frac{\log(M) - \log(\epsilon_c)}{\epsilon_c^2} M^{24d_0+28} \epsilon^{-16d_0-12}. \qquad {\scriptstyle \text{(By Lemma F.4)}}$$

This completes the proof. $\qquad \square$

We now bound (I) with covering number of loss function class and (II), the empirical risk.

**Lemma F.6** (Generalization Bound, Modified from Lemma L.5 of (Su et al., 2025)). *Let $\widehat{u}_\theta^{i_0}$ with parameter $\widehat{\Theta}^{i_0}$ be the velocity estimator trained by minimizing $\widehat{\mathcal{L}}_{\text{CDFM}}^{i_0}$ with i.i.d training samples $\{x_i\}_{i=1}^n$, where $x_i \in S$. For simplicity, we use $\mathcal{N}$ to denote $\mathcal{N}(F_{\text{loss}}^{i_0}, \epsilon_c, \{x_i\}_{x_i \in S}, |\cdot|)$. Then we bound (I), the generalization bound as:*

$$\mathbb{E}_{\{x_i\}_{i=1}^n} [\mathcal{R}^{i_0}(\widehat{\Theta}^{i_0}) - \widehat{\mathcal{R}}^{i_0}(\widehat{\Theta}^{i_0})] \lesssim \mathbb{E}_{\{x_i\}_{i=1}^n} [\widehat{\mathcal{R}}^{i_0}(\widehat{\Theta}^{i_0})] + O(\frac{\kappa}{n} \log \mathcal{N} + \epsilon_c),$$

*where $\kappa$ denote the upper bound of $\ell^{i_0}(x; u_\theta^{i_0})$.*

*Proof.* The proof is modified from the proof of Lemma L.5 of (Su et al., 2025).

We use $\widehat{\mathcal{L}}_{\text{CDFM}}^{*,i_0}$ and $\widehat{\mathcal{R}}^{*,i_0}$ to denote the factorize conditional discrete flow matching loss and empirical risk with i.i.d training samples $\{x_i^*\}$. Then we have $\mathbb{E}_{\{x_i^*\}_{i=1}^n} [\widehat{\mathcal{L}}_{\text{CDFM}}^{*,i_0}(\Theta^{i_0})] = \mathcal{L}_{\text{CDFM}}^{i_0}(\Theta^{i_0})$ and $\mathbb{E}_{\{x_i^*\}_{i=1}^n} [\widehat{\mathcal{R}}^{*,i_0}(\Theta^{i_0})] = \mathcal{R}^{i_0}(\Theta^{i_0})$ for all parameter set $\Theta^{i_0}$. We now rewrite (I) as:

$$\mathbb{E}_{\{x_i\}_{i=1}^n} [\mathcal{R}^{i_0}(\widehat{\Theta}^{i_0}) - \widehat{\mathcal{R}}^{i_0}(\widehat{\Theta}^{i_0})]$$

$$= \mathbb{E}_{\{x_i\}_{i=1}^n} [\mathbb{E}_{\{x_i^*\}_{i=1}^n} [\widehat{\mathcal{R}}^{*,i_0}(\widehat{\Theta}^{i_0})] - \widehat{\mathcal{R}}^{i_0}(\widehat{\Theta}^{i_0})]$$

$$= \mathbb{E}_{\{x_i,x_i^*\}_{i=1}^n} [\widehat{\mathcal{R}}^{*,i_0}(\widehat{\Theta}^{i_0}) - \widehat{\mathcal{R}}^{i_0}(\widehat{\Theta}^{i_0})]$$

$$= \frac{1}{n} \mathbb{E}_{\{x_i,x_i^*\}_{i=1}^n} [(\sum_{i=1}^n \ell^{i_0}(x_i^*; \widehat{u}_\theta^{i_0}) - \sum_{i=1}^n \ell^{i_0}(x_i^*; u^{i_0})) - (\sum_{i=1}^n \ell^{i_0}(x_i; \widehat{u}^{i_0}) - \sum_{i=1}^n \ell^{i_0}(x_i; u^{i_0}))].$$

For $\epsilon_c$ to be chosen later, let $\mathcal{L} = \{\ell_1^{i_0}, \ell_2^{i_0}...\ell_{\mathcal{N}}^{i_0}\}$ be a $\epsilon_c$-covering of $F_{\text{loss}}^{i_0}$ with respect to point set $S$. That's to say, for every $\widehat{u}_\theta^{i_0}$, there exists $\ell_j^{i_0} \in \mathcal{L}$ such that $|\ell_j^{i_0} - \ell^{i_0}(x, \widehat{u}_\theta^{i_0})| \le \epsilon_c$ for every $x \in S$. For simplicity of notations, we have the following definitions:

$$\omega(x) := \ell^{i_0}(x; \widehat{u}_\theta^{i_0}) - \ell^{i_0}(x; u^{i_0}),$$

$$\omega_j(x) := \ell_j^{i_0}(x) - \ell^{i_0}(x; u^{i_0}),$$

$$h_j := \max\{A, \sqrt{\mathbb{E}_{z \in \mathcal{S}}[\ell_j^{i_0}(z) - \ell^{i_0}(z, u^{i_0})]}\}, \qquad \text{\small($A$ is a constant to be chosen later)}$$

$$\Omega := \max_{k \in [\mathcal{N}]} |\sum_{i=1}^{n} \frac{\omega_k(x_i) - \omega_k(x_i^*)}{h_k}|.$$

Then we further obtain:

$$|\frac{1}{n}\mathbb{E}_{\{x_i, x_i^*\}_{i=1}^n}[(\sum_{i=1}^{n} \ell^{i_0}(x_i^*; \widehat{u}_\theta^{i_0}) - \sum_{i=1}^{n} \ell^{i_0}(x_i^*; u^{i_0})) - (\sum_{i=1}^{n} \ell^{i_0}(x_i; \widehat{u}_\theta^{i_0}) - \sum_{i=1}^{n} \ell^{i_0}(x_i; u^{i_0}))]| \tag{F.5}$$

$$\leq \frac{1}{n}\mathbb{E}_{\{x_i, x_i^*\}_{i=1}^n}|\omega(x_i^*) - \omega(x_i)| \qquad \text{\small(By definition of $\omega(x)$)}$$

$$\leq \frac{1}{n}\mathbb{E}_{\{x_i, x_i^*\}_{i=1}^n}|\omega_j(x_i^*) - \omega_j(x_i)| + 2\epsilon_c \qquad \text{\small(By $|w_j(x) - \omega(x)| \leq \epsilon_c$ when $x \in \mathcal{S}$)}$$

$$\leq \frac{1}{n}\mathbb{E}_{\{x_i, x_i^*\}_{i=1}^n}[h_j\Omega] + 2\epsilon_c \qquad \text{\small(By definition of $h_j$ and $\Omega$)}$$

$$\leq \frac{1}{2}\mathbb{E}_{\{x_i, x_i^*\}_{i=1}^n}[h_j^2] + \frac{1}{2n^2}\mathbb{E}_{\{x_i, x_i^*\}_{i=1}^n}[\Omega^2] + 2\epsilon_c, \tag{F.6}$$

where the last line follows AM-GM Inequality. In the following paragraphs, we bound $\mathbb{E}_{\{x_i, x_i^*\}_{i=1}^n}[h_j^2]$ and $\mathbb{E}_{\{x_i, x_i^*\}_{i=1}^n}[\Omega^2]$ separately. We start with bounding $\mathbb{E}_{\{x_i, x_i^*\}_{i=1}^n}[h_j^2]$:

$$\mathbb{E}_{\{x_i, x_i^*\}_{i=1}^n}[h_j^2] = \mathbb{E}_{\{x_i\}_{i=1}^n}[h_j^2]$$

$$\leq A^2 + \mathbb{E}_{\{x_i\}_{i=1}^n, z \in \mathcal{S}}[\ell_j^{i_0}(z) - \ell^{i_0}(z, u)]$$

$$\leq A^2 + \mathbb{E}_{\{x_i\}_{i=1}^n, z \in \mathcal{S}}[\ell^{i_0}(z; \widehat{u}_\theta^{i_0}) - \ell^{i_0}(z; u^{i_0})] + \epsilon_c$$

$$\text{\small($|\ell_j^{i_0}(z) - \ell^{i_0}(z, \widehat{u}_\theta^{i_0})| \leq \epsilon_c$ when $z \in \mathcal{S}$)}$$

$$= A^2 + \mathbb{E}_{\{x_i\}_{i=1}^n}[\mathcal{R}^{i_0}(\widehat{\Theta}^{i_0})] + \epsilon_c. \tag{F.7}$$

where the last line follows Lemma F.1. Next, we bound $\mathbb{E}_{\{x_i, x_i^*\}_{i=1}^n}[\Omega^2]$ by using Bernstein's Inequality to bound $\Pr(\Omega \geq b)$ for given $b > 0$. We define $a_{k,i}$ as $a_{k,i} := \frac{\omega_k(x_i) - \omega_k(x_i^*)}{h_k}$. Then $\Omega = \max_{k \in [\mathcal{N}]} |\sum_{i=1}^{n} a_{k,i}|$. Also, since $\kappa$ denote the upper bound of $\ell^{i_0}(x; u_\theta^{i_0})$, we have $\ell^{i_0}(x; u_\theta^{i_0}) \leq \kappa$. For $k \in [\mathcal{N}]$ we have:

$$|a_{k,i}| \leq |\frac{\ell_k^{i_0}(x)}{h_k}| \leq \frac{\kappa}{A}. \tag{F.8}$$

By definition of $\omega_j(x)$ and $h_j$, for all $k \in [\mathcal{N}]$ we have:

$$\mathbb{E}_x[\omega_k(x)] = \mathbb{E}_x[\ell_k^{i_0}(x) - \ell^{i_0}(x; u^{i_0})] \leq h_k^2. \tag{F.9}$$

By symmetrization, we get:

$$\mathbb{E}[a_{k,i}] = \mathbb{E}_{\{x_i, x_i^*\}}[\frac{\omega_k(x_i) - \omega_k(x_i^*)}{h_k}] = 0. \tag{F.10}$$

Then we bound the variation of $a_{k,i}$ as:

$$\text{Var}_{x_i, x_i^*}[a_{k,i}] = \mathbb{E}_{x_i, x_i^*}[(\frac{\omega_k(x_i) - \omega_k(x_i^*)}{h_k})^2]$$

$$= \mathbb{E}_{x_i, x_i^*}[\frac{\omega_k^2(x_i)}{h_k^2} + \frac{\omega_k^2(x_i^*)}{h_k^2} - \frac{2\omega_k(x_i)\omega_k(x_i^*)}{h_k^2}]$$

$$= 2\mathbb{E}_x[\frac{\omega_k^2(x)}{h_k^2}] - 2(\mathbb{E}_x[\frac{\omega_k(x)}{h_k}])^2 \qquad \text{(By symmetrization)}$$

$$\leq 2\mathbb{E}_x[\frac{\omega_k^2(x)}{h_k^2}]$$

$$\leq 2\kappa\mathbb{E}_x[\frac{\omega_k(x)}{h_k^2}] \qquad \text{(By (F.8))}$$

$$\leq 2\kappa, \qquad \text{(F.11)}$$

where the last line follows (F.9).

Then for $b > 0$ we have:

$$\Pr[\Omega^2 \geq b^2] = \Pr[\Omega \geq b] \qquad \text{(F.12)}$$

$$\leq \mathcal{N}\Pr\left[|\sum_{i=1}^k a_{k,i}| \geq b\right] \qquad \text{(By union bound)}$$

$$\leq 2\mathcal{N}\exp\left(-\frac{\frac{b^2}{2}}{\sum_{i=1}^n 2\kappa + \frac{b\kappa}{3A}}\right) \qquad \text{(By (F.8),(F.11) and Bernstein's Inequality)}$$

$$= 2\mathcal{N}\exp\left(-\frac{\frac{b^2}{2}}{2n\kappa + \frac{b\kappa}{3A}}\right) \qquad \text{(F.13)}$$

We now bound $\mathbb{E}_{\{x_i,x_i^*\}_{i=1}^n}[\Omega^2]$ as:

$$\mathbb{E}_{\{x_i,x_i^*\}_{i=1}^n}[\Omega^2] = \int_0^{b_0^2}\Pr[\Omega^2 < b^2]\mathrm{d}b + \int_{b_0^2}^\infty \Pr[\Omega^2 \geq b^2]\mathrm{d}b \qquad (b_0 \text{ is a constant to be determined})$$

$$\leq b_0^2 + \int_{b_0^2}^\infty 2\mathcal{N}\exp\left(-\frac{\frac{b^2}{2}}{2n\kappa + \frac{b\kappa}{3A}}\right)\mathrm{d}b \qquad \text{(By (F.13))}$$

$$\leq b_0^2 + \int_{b_0^2}^\infty 2\mathcal{N}\exp\left(-\frac{Ab}{\kappa}\right)\mathrm{d}b \qquad \text{(Assume } b_0 \geq 12nA)$$

$$= b_0^2 + \frac{2\mathcal{N}\kappa}{A}\exp\left(-\frac{Ab_0^2}{\kappa}\right).$$

Let $b_0 = \sqrt[3]{n\kappa\log\mathcal{N}}$ and $A = \frac{b_0}{12n}$, we have:

$$\mathbb{E}_{\{x_i,x_i^*\}_{i=1}^n}[\Omega^2] \lesssim n\kappa\log\mathcal{N}. \qquad \text{(F.14)}$$

Substituting the result of (F.7) and (F.14) into (F.6), we finish the proof. $\qquad\square$

With Lemma F.6, we reduce bounding (I) to bounding (II). Finally, we bound (II) with Theorem E.7.

**Lemma F.7** (Empirical Risk Bound, Modified from Theorem L.1 of (Su et al., 2025)). *Consider the transformer class $\mathcal{T}_R^{h,s,r}$ with parameter bound given in Theorem E.7. Let $\widehat{u}_\theta^{i_0} \in \mathcal{T}_R^{h,s,r}$ with parameter $\widehat{\Theta}^{i_0}$ be the factorized velocity estimator under mixture path setting trained by minimizing $\widehat{\mathcal{L}}_{\mathrm{CDFM}}^{i_0}$ with i.i.d training samples $\{x_i\}_{i=1}^n$, where $x_i \in \mathcal{S}$. Let factorized empirical risk $\widehat{\mathcal{R}}^{i_0}(\widehat{\Theta}^{i_0})$ be as defined in (F.2). Then we have:*

$$\mathbb{E}_{\{x_i\}_{i=1}^n}[\widehat{\mathcal{R}}^{i_0}(\widehat{\Theta}^{i_0})] \lesssim \epsilon^{\frac{4}{M+2}}M^{\frac{12Md_0+25M}{M+2}}.$$

*Proof.* The proof is modified from proof of Theorem L.1 of (Su et al., 2025).

Given that $\widehat{u}_\theta$ with parameter $\widehat{\Theta}$ is the minimizer of $\widehat{\mathcal{L}}_{\mathrm{CDFM}}$, for all $u_\theta$ with parameter $\Theta$ we have:

$$\mathbb{E}_{\{x_i\}_{i=1}^n}[\widehat{\mathcal{R}}^{i_0}(\widehat{\Theta}^{i_0})] = \mathbb{E}_{\{x_i\}_{i=1}^n}[\widehat{\mathcal{L}}_{\mathrm{CDFM}}^{i_0}(\widehat{u}_\theta^{i_0}) - \widehat{\mathcal{L}}_{\mathrm{CDFM}}^{i_0}(u^{i_0})]$$

$$\leq \underset{\{x_i\}_{i=1}^n}{\mathbb{E}} [\widehat{\mathcal{L}}_{\mathrm{CDFM}}^{i_0}(u_\theta^{i_0}) - \widehat{\mathcal{L}}_{\mathrm{CDFM}}^{i_0}(u^{i_0})]$$

$$= \underset{\{x_i\}_{i=1}^n}{\mathbb{E}} [\widehat{\mathcal{R}}^{i_0}(\Theta^{i_0})]$$

$$= \mathcal{R}^{i_0}(\Theta^{i_0}). \qquad\qquad \text{(By Lemma F.1)}$$

Let $u_\theta^{i_0,\mathrm{approx}}$ with parameter $\Theta^{i_0,\mathrm{approx}}$ be the approximator network in Theorem E.7, we obtain:

$$\underset{\{x_i\}_{i=1}^n}{\mathbb{E}} \widehat{\mathcal{R}}^{i_0}[(\widehat{\Theta}^{i_0})] \leq \mathcal{R}^{i_0}(\Theta^{i_0,\mathrm{approx}})$$

$$= \sum_{x \in \mathcal{S}} \|u_\theta^{i_0,\mathrm{approx}}(x,t) - u^{i_0}(x,t)\|_2^2 \cdot p_t(x)$$

$$\lesssim \epsilon^{\frac{4}{M+2}} M^{\frac{12Md_0+25M}{M+2}}.$$

This completes the proof. $\qquad\qquad\qquad\qquad\qquad\qquad\qquad\qquad\qquad\qquad\square$

### F.3 MAIN PROOF OF THEOREM 5.1

This section presents the main proof of Theorem 5.1. In the main text, we give a simplified version of Theorem 5.1 to keep the exposition concise, while Theorem F.8 provides the explicit form.

**Theorem F.8** (Mixture Path Discrete Flow Matching Velocity Estimation with Transformer, Theorem 5.1 Restate)**.** *Let* $\widehat{u}_\theta^{i_0} \in \mathcal{T}_R^{h,s,r}$ *with parameter* $\widehat{\Theta}^{i_0}$ *be the factorized velocity estimator under mixture path setting trained by minimizing* $\widehat{\mathcal{L}}_{\mathrm{CDFM}}^{i_0}$ *with i.i.d training samples* $\{x_i\}_{i=1}^n$*, where* $x_i \in \mathcal{S}$*. Then for large enough* $n$ *we have:*

$$\underset{\{x_i\}_{i=1}^n}{\mathbb{E}} [\mathcal{R}^{i_0}(\widehat{\Theta}^{i_0})] \lesssim M^{12d_0+25} n^{-\frac{1}{4Md_0+3M+8d_0+9}} (\log n)^{\frac{1}{4Md_0+3M+8d_0+9}}$$

*Proof.* Recall the decomposition given in (F.3):

$$\underset{\{x_i\}_{i=1}^n}{\mathbb{E}} [\mathcal{R}^{i_0}(\widehat{\Theta}^{i_0})] = \underset{\{x_i\}_{i=1}^n}{\mathbb{E}} [\mathcal{R}^{i_0}(\widehat{\Theta}^{i_0}) - \widehat{\mathcal{R}^{i_0}}(\widehat{\Theta}^{i_0})] + \underset{\{x_i\}_{i=1}^n}{\mathbb{E}} [\widehat{\mathcal{R}}^{i_0}(\widehat{\Theta}^{i_0})].$$

Substituting what we get in Lemma F.6 and Lemma F.7 into the decomposition, we get:

$$\underset{\{x_i\}_{i=1}^n}{\mathbb{E}} [\mathcal{R}^{i_0}(\widehat{\Theta}^{i_0})] \lesssim O(\frac{\kappa}{n} \log \mathcal{N} + \epsilon_c) + 2 \underset{\{x_i\}_{i=1}^n}{\mathbb{E}} [\widehat{\mathcal{R}}^{i_0}(\widehat{\Theta}^{i_0})] \qquad \text{(By Lemma F.6)}$$

$$\lesssim O(\frac{\kappa}{n} \log \mathcal{N} + \epsilon_c) + \epsilon^{\frac{4}{M+2}} M^{\frac{12Md_0+25M}{M+2}} \qquad\qquad \text{(By Lemma F.7)}$$

$$\lesssim \frac{\log(M) - \log(\epsilon_c)}{n\epsilon_c^2} M^{24d_0+28} \epsilon^{-16d_0-12} + \epsilon_c + \epsilon^{\frac{4}{M+2}} M^{\frac{12Md_0+25M}{M+2}}.$$

$$\text{(By Lemma F.5 and } \kappa \lesssim 1 \text{ under mixture path setting)}$$

Next, we choose proper $\epsilon, \epsilon_c$ to get a optimal bound for the estimation rates.

First, we let $\epsilon_c = (\frac{\log(n\epsilon) M^{24d_0+28} \epsilon^{-16d_0-12}}{n})^{1/3}$ and get

$$\underset{\{x_i\}_{i=1}^n}{\mathbb{E}} [\mathcal{R}^{i_0}(\widehat{\Theta}^{i_0})] \lesssim (\log(n\epsilon))^{1/3} M^{\frac{24d_0+28}{3}} \epsilon^{-\frac{16d_0+12}{3}} n^{-\frac{1}{3}} + \epsilon^{\frac{4}{M+2}} M^{\frac{12Md_0+25M}{M+2}}.$$

Next, let $\epsilon = M^{-\frac{3}{4}} n^{-\frac{M+2}{16Md_0+12M+32d_0+36}} (\log n)^{\frac{M+2}{16Md_0+12M+32d_0+36}}$, then for large enough $n$ we get:

$$\underset{\{x_i\}_{i=1}^n}{\mathbb{E}} [\mathcal{R}^{i_0}(\widehat{\Theta}^{i_0})] \lesssim M^{12d_0+25} n^{-\frac{1}{4Md_0+3M+8d_0+9}} (\log n)^{\frac{1}{4Md_0+3M+8d_0+9}}.$$

This completes the proof. $\qquad\qquad\qquad\qquad\qquad\qquad\qquad\qquad\qquad\qquad\square$

# G    PROOF OF THEOREM 5.2

This section provides the main proof of Theorem 5.2. In the main text, we give a simplified version of Theorem 5.2 to keep the exposition concise, while Theorem G.1 below presents the explicit form.

**Theorem G.1** (Discrete Flow Matching Distribution Estimation with Transformer, Theorem 5.2 Restate). *For any coordinate $i_0 \in [d]$, let $\widehat{u}_\theta^{i_0}$ be the $i$-th velocity estimator trained by minimizing empirical conditional discrete flow matching loss $\widehat{\mathcal{L}}_{\mathrm{CDFM}}^{i_0}$ following (2.10). Let $P$ denote the true distribution and $\widehat{P}$ the distribution generated by the discrete flow matching framework with factorized velocity estimators $\{\widehat{u}_\theta^{i_0}\}_{i^0=1}^d$. Then for a vocabulary size $M$, the expected total variation distance $TV(P, \widehat{P})$ over training data $\{x_i\}_{i=1}^n$ is bounded by:*

$$
\underset{\{x_i\}_{i=1}^n}{\mathbb{E}} [TV(P, \widehat{P})] \lesssim M^{6d_0+13} \exp(M) n^{-\frac{1}{8Md_0+6M+16d_0+18}} (\log n)^{\frac{1}{8Md_0+6M+16d_0+18}}.
$$

*Proof.* From Theorem 3.1, we have:

$$
\mathrm{TV}(P, \widehat{P}) \lesssim \sqrt{M} \exp(2M_u) \sum_{i_0 \in [d]} \sqrt{\mathcal{R}^{i_0}(\widehat{\Theta})}.
$$

Taking expectation on both sides and recalling that $M_u = O(1)$ under mixture setting, we obtain:

$$
\underset{\{x_i\}_{i=1}^n}{\mathbb{E}} [\mathrm{TV}(P, \widehat{P})] \lesssim \sqrt{M} \underset{\{x_i\}_{i=1}^n}{\mathbb{E}} \Big[ \sum_{i_0 \in [d]} \sqrt{\mathcal{R}^{i_0}(\widehat{\Theta})} \Big]
$$

$$
\lesssim M^{6d_0+13} n^{-\frac{1}{8Md_0+6M+16d_0+18}} (\log n)^{\frac{1}{8Md_0+6M+16d_0+18}}. \qquad \text{(By Theorem F.8)}
$$

This completes the proof. $\qquad\qquad\square$

# H  APPROXIMATION THEORY FOR DISCRETE FLOW MATCHING: GENERAL CASE

This section establishes the approximation theory for discrete flow matching in the general case.

**Organization.** We recall and restate important lemmas in Appendix H.1. Then we present the main proof of Theorem H.3, with same proof strategy in Theorem E.7.

## H.1  AUXILIARY LEMMAS

In this section, we restate auxiliary lemmas for proving the approximation theory Theorem H.3. We start with restating Lemma E.3 as Lemma H.1 , computing the Lipschitz constant of $\widetilde{u}(x,t)$ constructed in Lemma 4.5. Then we restate Lemma E.6 as Lemma H.2, a key lemma in proof of Theorem H.3 bounding function's local value with its integral's value lower bound.

To begin with, we restate Lemma E.3 proved in Appendix E.2 as Lemma H.1. This lemma computes the Lipschitz constant of $\widetilde{u}(x,t)$ in Lemma 4.5.

**Lemma H.1** (Lipschitzness of Extension, Lemma E.3 Restate). *Suppose that for every given $s \in \mathcal{S}$, it holds $\|u(s,t)\|_2 \le M_u$ and $u(s,t)$ is Lipschitz continuous under $\ell_2$-norm with respect to $t$, with Lipschitz constant $L_u$. Then $\widetilde{u}(x,t)$ defined in the proof of Lemma 4.5 is Lipschitz continuous under $\ell_2$-norm with respect to $(x,t)$, with Lipschitz constant $\max\{L_u, 4e\sqrt{d}M_u\}$.*

*Proof.* See the proof of Lemma E.3. □

Then we restate Lemma E.6 proved in Appendix E.2 as Lemma H.2, bounding local value of Lipschitz continuous function with its integral value.

**Lemma H.2** (Point-wise Upper Bound via Integral Constraint, Lemma E.6 Restate). *Suppose that $f : [-I, I]^{d \times L} \to \mathbb{R}^{d \times L}$ is Lipschitz continuous on a bounded domain under Frobenius norm, with Lipschitz constant $L_f$. Let $n = dL$. Let $(\int \|f(Z)\|_F^2 \mathrm{d}Z)^{1/2} \le b$, then for all $Z \in [-I, I]^{d \times L}$ it holds:*

$$\|f(Z)\|_F \lesssim b^{\frac{2}{n+2}} L_f^{\frac{n}{n+2}}.$$

*Proof.* See the proof of Lemma E.6. □

## H.2  APPROXIMATION THEORY FOR DISCRETE FLOW MATCHING

In this section, we prove the approximation theorem for discrete flow matching in the general case. Similar to proof of Theorem E.7, we treat the upper bound of the Lipschitz constant of the approximator Transformer class as a constant independent of $\epsilon$.

**Theorem H.3** (Approximation Theorem for Discrete Flow Matching). *Suppose that for every given $x \in \mathcal{S}$, $u(x,t)$ is bounded and Lipschitz continuous with respect to $t$, such that $\|u(x,t)\|_2 \le M_u$ and Lipschitz constant is $L_u$. Then for every $\epsilon > 0$, there exists a transformer network $u_\theta(x,t) \in \mathcal{T}_R^{h,s,r}(C_{\mathcal{T}}, C_{KQ}^{2,\infty}, C_{KQ}, C_{OV}^{2,\infty}, C_{OV}, C_E, C_F^{2,\infty}, C_F)$ satisfying that for every $t \in [0,1]$:*

$$\sum_{x \in \mathcal{S}} \|u_\theta(x,t) - u(x,t)\|_2^2 \cdot p_t(x) \lesssim \epsilon^{\frac{4}{M^d+2}} M^{\frac{4M^d d d_0 + 13M^d d + 8M^d d_0 + 12M^d}{M^d+2}} M_u^{\frac{8M^d}{M^d+2}},$$

*where $d_0$ is the transformer feature dimension. The parameter bound of the transformer network class follows:*

$$C_{KQ}, C_{KQ}^{2,\infty} = \widetilde{O}(M^{2dd_0+d+4d_0+2} \epsilon^{-4d_0-2}); \quad C_{OV}, C_{OV}^{2,\infty} = O(M^{-\frac{1}{2}d} \epsilon)$$

$$C_F, C_F^{2,\infty} = O(M^{d+1} M_u \epsilon^{-1}); \quad C_E = O(M^d),$$

*where $O(\cdot)$ hides polynomial factors depending on $d, d_0$, $\widetilde{O}(\cdot)$ hides polynomial factors depending on $d, d_0$ and logarithmic factors depending on $M$.*

*Proof.* Similar to proof of Theorem E.7, we start with introducing the reshape layer of $u_\theta(x, t)$. In the general case, we need to approximate function $\widetilde{u}^i(x, t)$, with input dimension $d + 1$ and output dimension $M^d$. To make up for this difference, we introduce two reshape layers: $R_1$ and $R_2$. Note that we assume $d + 1 | M^d$ for simplicity in discussions below.

- $R_1 : [0, M]^d \times [0, 1] \to \mathbb{R}^{d_0 \times \frac{M^d}{d_0}}$ is a reshape function rearranging a vector of dimension $d + 1$ into a matrix of size $\mathbb{R}^{d_0 \times \frac{M^d}{d_0}}$, where transformer feature dimension $d_0$ satisfies $d_0 | d + 1$. We realize $R_1$ by first reshaping input vector $(x, t)$ with dimension $d + 1$ into a matrix $A \in \mathbb{R}^{d_0 \times \frac{d+1}{d_0}}$, following the standard procedure of rearranging entries. Then we replicate the matrix $\frac{M^d}{d+1}$ times along its columns, yielding a matrix of size $d_0 \times \frac{M^d}{d_0}$. Altogether, the output of $R_1$ is a matrix of size $d_0 \times \frac{M^d}{d_0}$. As the reverse of $R_1$, $R_1^{-1} : \mathbb{R}^{d_0 \times \frac{M^d}{d_0}} \to [0, M]^d \times [0, 1]$ is defined by taking first $\frac{d+1}{d_0}$ columns of the matrix, and then rearranging it into a vector of dimension $d + 1$.

- $R_2 : \mathbb{R}^{d_0 \times \frac{M^d}{d_0}} \to \mathbb{R}^{M^d}$ is a reshape function rearranging a matrix of size $\mathbb{R}^{d_0 \times \frac{M^d}{d_0}}$ into a vector of dimension $M^d$. We realize $R_2$ by rearranging the entries of the matrix into a vector preserving the total number of elements, following standard reshape layer construction. We define $R_2^{-1}$ as the reverse map of $R_2$. This is well-defined since $R_2$ is bijection between $\mathbb{R}^{d_0 \times \frac{M^d}{d_0}}$ and $\mathbb{R}^{M^d}$.

Again, as discussed in proof of Theorem E.7, we state the construction above for clarity, while the construction of $R_1$ and $R_2$ is not focus of our discussion.

Now we return to the main proof. Let $\widetilde{u}(x, t)$ be as defined in Lemma 4.5. Let $u^{\mathrm{reshape}} : \mathbb{R}^{d_0 \times \frac{M^d}{d_0}} \to \mathbb{R}^{d_0 \times \frac{M^d}{d_0}}$ be as:

$$u^{\mathrm{reshape}} = R_2^{-1} \circ \widetilde{u}(x, t) \circ R_1^{-1}.$$

Then $u^{\mathrm{reshape}}$ is Lipschitz continuous under Frobenius norm, with Lipschitz constant no larger than Lipschitz constant of $\widetilde{u}$. By Proposition B.17, for any $\epsilon$ there exists a

$$u_\theta^{\mathrm{reshape}}(Z) = F_1^{\mathrm{FF}} \circ F^{\mathrm{SA}} \circ F_2^{\mathrm{FF}} \circ F^{\mathrm{E}} \in \mathcal{T}^{h,s,r}(C_{\mathcal{T}}, C_{KQ}^{2,\infty}, C_{KQ}, C_{OV}^{2,\infty}, C_{OV}, C_E, C_F^{2,\infty}, C_F),$$

such that $d_F(u^{\mathrm{reshape}}(Z), u_\theta^{\mathrm{reshape}}(Z)) < \epsilon$, where $d_F(f(Z), g(Z)) := (\int \|f(Z) - g(Z)\|_F^2 \mathrm{d}Z)^{1/2}$. By Theorem B.18, the parameter bound of the transformer network satisfies:

$$C_{KQ}, C_{KQ}^{2,\infty} = \widetilde{O}(M^{2dd_0 + d + 4d_0 + 2} \epsilon^{-4d_0 - 2}); C_{OV}, C_{OV}^{2,\infty} = O(M^{-\frac{1}{2}d} \epsilon)$$
$$C_F, C_F^{2,\infty} = O(M^{d+1} M_u \epsilon^{-1}); C_E = O(M^d), \tag{H.1}$$

where $O(\cdot)$ hides polynomial factors depending on $d, d_0$, $\widetilde{O}(\cdot)$ hides polynomial factors depending on $d, d_0$ and logarithmic factors depending on $M$.

Let $h = u_\theta^{\mathrm{reshape}} - u^{\mathrm{reshape}}$. Then $(\int \|h\|_F^2 \mathrm{d}Z)^{1/2} < \epsilon$. Further, by Lemma B.11 and Lemma H.1, $u_\theta^{\mathrm{reshape}}, u^{\mathrm{reshape}}$ are Lipschitz continuous under Frobenius norm. Then $h$ is Lipschitz continuous under Frobenius norm. We denote their Lipschitz constant as $L_\theta^{\mathrm{reshape}}, L^{\mathrm{reshape}}$ and $L_h$ respectively.

We first compute $L_\theta^{\mathrm{reshape}}$ according to Lemma B.11. It holds:

$$L_\theta^{\mathrm{reshape}} \leq (1 + 2M^{2d} C_{OV} C_{KQ} + h \frac{M^d}{d_0} C_{OV}) \cdot (C_F^2 + 1)^2$$
$$\lesssim (C_F)^2 \cdot M^{2d} \cdot h C_{KQ} C_{OV} \cdot (C_F)^2$$
$$\lesssim M^{2dd_0 + \frac{13}{2}d + 4d_0 + 6} M_u^4. \qquad \text{(By (H.1) and dropping terms of } \epsilon)$$

Then we compute $L_h$. We have:

$$L_h \leq L_\theta^{\mathrm{reshape}} + L^{\mathrm{reshape}}$$

$$\lesssim M^{2dd_0 + \frac{13}{2}d + 4d_0 + 6} M_u^4. \qquad \text{(By Lemma H.1)}$$

Let $u_\theta := R_2 \circ u_\theta^{\text{reshape}} \circ R_1$. For all $(x, t) \in [0, M]^d \times [0, 1]$, it holds:

$$\|u_\theta(x, t) - \widetilde{u}(x, t)\|_2^2 \leq \|u_\theta^{\text{reshape}} \circ R_1(x, t) - u^{\text{reshape}} \circ R_1(x, t)\|_F^2$$

$$= \|h(R_1(x, t))\|_F^2 \qquad \text{(By definition of } h\text{)}$$

$$\lesssim \epsilon^{\frac{4}{M^d + 2}} M^{\frac{4M^d dd_0 + 13M^d d + 8M^d d_0 + 12M^d}{M^d + 2}} M_u^{\frac{8M^d}{M^d + 2}}, \qquad \text{(By Lemma H.2)}$$

where $L_h = M^{2dd_0 + \frac{13}{2}d + 4d_0 + 6} M_u^4$. Then we have for every $t \in [0, 1]$, it holds:

$$\sum_{x \in \mathcal{S}} \|u_\theta(x, t) - u(x, t)\|_2^2 \cdot p_t(x) \lesssim \epsilon^{\frac{4}{M^d + 2}} M^{\frac{4M^d dd_0 + 13M^d d + 8M^d d_0 + 12M^d}{M^d + 2}} M_u^{\frac{8M^d}{M^d + 2}} \sum_{x \in \mathcal{S}} p_t(x)$$

$$= \epsilon^{\frac{4}{M^d + 2}} M^{\frac{4M^d dd_0 + 13M^d d + 8M^d d_0 + 12M^d}{M^d + 2}} M_u^{\frac{8M^d}{M^d + 2}},$$

This completes the proof. $\qquad\qquad\square$

# I   ESTIMATION THEORY FOR DISCRETE FLOW MATCHING: GENERAL CASE

This section derives estimation rates for discrete flow matching with transformers in the general case. The analysis adapts and modifies the risk-decomposition plus covering-number technique of (Fu et al., 2024) to our setting and parameter bounds from Theorem H.3.

**Organization.** This section consists of four steps to obtain the estimation rates of discrete flow matching. The proof structure follows Appendix F.

- **Preliminaries.** In Appendix I.1, we introduce several essential concepts, including empirical loss $\widehat{\mathcal{L}}_{\text{CDFM}}$, discrete flow matching risk $\mathcal{R}(\Theta)$ and empirical risk $\widehat{\mathcal{R}}(\Theta)$.

- **Covering Number Upper bound.** We obtain covering-number bounds for the transformer class using the parameter constraints from Theorem H.3 and for the induced loss class in Appendix I.2, Lemma I.2-Lemma I.4.

- **Generalization and Approximation Error Bound.** We apply the covering-number machinery and conclusions from Theorem H.3 to bound generalization and approximation error in Appendix I.2, Lemmas I.5 and I.6.

- **Estimation rates.** We apply conclusions from prior three steps to prove the velocity estimation rate in Theorem I.7 and then the distribution estimation rate in Theorem I.8.

## I.1   PRELIMINARIES

In practice, given $n$ i.i.d training samples $\{x_i\}_{i=1}^n$, the transformer network is trained through minimizing the empirical loss:

$$\widehat{\mathcal{L}}_{\text{CDFM}} := \frac{1}{n} \sum_{i=1}^n \int_0^1 \mathbb{E}_{X_0 \sim p_0, X_t \sim p_{t|x_0 = X_0, x_1 = x_i}} \|u(X_t, t) - u_\theta(X_t, t)\|_2^2 \mathrm{d}t.$$

Similar to notations in Appendix F.1, we define the loss of certain function $f$ with respect to some certain point $x$ as:

$$\ell(x; f) := \int_0^1 \mathbb{E}_{X_0 \sim p_0, X_t \sim p_{t|x_0 = X_0, x_1 = x}} \|u(X_t, t) - f(X_t, t)\|_2^2 \mathrm{d}t.$$

Then we have:

$$\widehat{\mathcal{L}}_{\text{CDFM}} = \frac{1}{n} \sum_{i=1}^n \ell(x_i; u_\theta).$$

We use $\widehat{\Theta}$ to denote the parameter of network trained by minimizing the empirical loss with $n$ i.i.d training samples $\{x_i\}_{i=1}^n$. That's to say, discrete flow matching network $\widehat{u}_\theta$ with parameter $\widehat{\Theta}$ is the empirical risk minimizer, satisfying $\widehat{\Theta} \in \underset{\Theta}{\operatorname{argmin}} \, \widehat{\mathcal{L}}_{\mathrm{CDFM}}(u_\theta)$.

Similar to the factorized case, for a discrete flow matching network $u_\theta$ with parameter $\Theta$, its performance in velocity estimation is measured by the discrete flow matching risk, which is defined as:

$$\mathcal{R}(\Theta) := \int_0^1 \underset{X_t \sim p_t}{\mathbb{E}} \|u(X_t, t) - u_\theta(X_t, t)\|_2^2 \mathrm{d}t. \tag{I.1}$$

In practice, we evaluate the performance of the network $u_\theta$ using empirical discrete flow matching risk, which is defined as:

$$\widehat{\mathcal{R}}(\Theta) := \frac{1}{n} \sum_{i=1}^n \ell(x_i; u_\theta) - \frac{1}{n} \sum_{i=1}^n \ell(x_i; u), \tag{I.2}$$

where $\{x_i\}_{i=1}^n$ are n i.i.d samples and $u$ is the true velocity. We have the following lemma showing that discrete flow matching risk is equal to the expectation of empirical discrete flow matching risk.

**Lemma I.1** (Modified from Lemma F.1). *For a discrete flow matching network $u_\theta(x, t)$ with parameters noted as $\Theta$ and i.i.d samples $\{x_i\}_{i=1}^n$, it holds:*

$$\underset{\{x_i\}_{i=1}^n}{\mathbb{E}} [\widehat{\mathcal{R}}(\Theta)] = \mathcal{R}(\Theta).$$

*Proof.* See the proof of Lemma F.1. The only difference is the integration domain in definition of $\ell$ and $\ell^{i^0}$, which is modified in the same way on both sides of the equality, from $[t_0, T]$ to $[0, 1]$. $\qquad\square$

## I.2 Auxiliary Lemmas

To bound $\underset{\{x_i\}_{i=1}^n}{\mathbb{E}} [\mathcal{R}^{i_0}(\widehat{\Theta}^{i_0})]$, we take the same decomposition approach introduced in Appendix F.2 Specifically, we have:

$$\underset{\{x_i\}_{i=1}^n}{\mathbb{E}} [\mathcal{R}(\widehat{\Theta})] = \underbrace{\underset{\{x_i\}_{i=1}^n}{\mathbb{E}} [\mathcal{R}(\widehat{\Theta}) - \widehat{\mathcal{R}}(\widehat{\Theta})]}_{(\mathrm{I})} + \underbrace{\underset{\{x_i\}_{i=1}^n}{\mathbb{E}} [\widehat{\mathcal{R}}(\widehat{\Theta})]}_{(\mathrm{II})}, \tag{I.3}$$

In this section, we introduce auxiliary lemmas helping us prove Theorem I.7. Specifically, we derive the covering number bound of transformers in Lemma I.2 and Lemma I.3. Then we get the covering number of loss function class in Lemma I.4. Lemma I.5 gives an upper bound on (I), the generalization bound. Finally, Lemma I.6 bound the empirical risk of a trained network.

To start with, we restate Lemma F.3 as Lemma I.2, giving an upper bound on the covering number of multiple-layer transformer network.

**Lemma I.2** (Lemma F.3 Restate, Lemma J.2 of (Hu et al., 2024b), Modified from Theorem A.17 of (Edelman et al., 2022)). *Let $\mathcal{T}_R^{h,s,r}(C_\mathcal{T}, C_{KQ}^{2,\infty}, C_{KQ}, C_{OV}^{2,\infty}, C_{OV}, C_E, C_F^{2,\infty}, C_F, L_\mathcal{T})$ represent the class of transformer network with parameter bound. Then for data points $x$ such that $\|x\|_2 \leq B_X$, we have:*

$$\log \mathcal{N}(\mathcal{T}_R^{h,s,r}, \epsilon, n, \|\cdot\|_2)$$
$$\leq \frac{\log(nL_\mathcal{T})}{\epsilon^2} \alpha^2 (d_0^{\frac{2}{3}} (C_F^{2,\infty})^{\frac{2}{3}} + d_0^{\frac{2}{3}} (2(C_F)^2 C_{OV} C_{KQ}^{2,\infty})^{\frac{2}{3}} + 2((C_F)^2 C_{OV}^{2,\infty})^{\frac{2}{3}})^3,$$

*where $\alpha = (C_F)^2 C_{OV}(1 + 4C_{KQ})(B_X + C_E)$.*

*Proof.* See the proof of Lemma F.3. $\qquad\square$

Equipped with Lemma I.2, we now compute the covering number of transformer network class with parameter bound given in Theorem H.3.

**Lemma I.3** (Covering Number Bound for Transformer Class, Modified from Lemma F.4). *Let $\epsilon_c > 0$. Consider the transformer class $\mathcal{T}_R^{h,s,r}(C_\mathcal{T}, C_{KQ}^{2,\infty}, C_{KQ}, C_{OV}^{2,\infty}, C_{OV}, C_E, C_F^{2,\infty}, C_F)$ with parameter bound given in Theorem H.3 and $x_i$ satisfying $x_i \in \mathcal{S}$. Then the $\epsilon_c$-covering number of $\mathcal{T}_R^{h,s,r}$ satisfies:*

$$\log \mathcal{N}(\mathcal{T}_R^{h,s,r}, \epsilon_c, n, \|\cdot\|_2) \lesssim \frac{\log(nMM_u\epsilon)}{\epsilon_c^2} M^{8dd_0+12d+16d_0+16} M_u^8 \epsilon^{-16d_0-12}.$$

*Proof.* The proof is modified from the proof of Lemma F.4.

From Theorem H.3, we have:

$$C_{KQ}, C_{KQ}^{2,\infty} = \widetilde{O}(M^{2dd_0+d+4d_0+2}\epsilon^{-4d_0-2}); C_{OV}, C_{OV}^{2,\infty} = O(M^{-\frac{1}{2}d}\epsilon)$$
$$C_F, C_F^{2,\infty} = O(M^{d+1}M_u\epsilon^{-1}); C_E = O(M^d); L_\mathcal{T} = O(M^{2dd_0+\frac{13}{2}d+4d_0+6}M_u^4). \quad \text{(I.4)}$$

Then we substitute (I.4) into Lemma I.2 and get:

$$\alpha \lesssim M^{2d+2}M_u^2\epsilon^{-2} \cdot M^{-\frac{1}{2}d}\epsilon \cdot M^{2dd_0+d+4d_0+2}\epsilon^{-4d_0-2} \cdot M^d = M^{2dd_0+\frac{7}{2}d+4d_0+4}M_u^2\epsilon^{-4d_0-3}.$$

Through further computation, we have

$$\log \mathcal{N}(\mathcal{T}_R^{h,s,r}, \epsilon_c, n, \|\cdot\|_2)$$
$$\lesssim \frac{\log(nL_\mathcal{T})}{\epsilon_c^2} \alpha^2 (d_0^{\frac{2}{3}}(2(C_F)^2 C_{OV} C_{KQ}^{2,\infty})^{\frac{2}{3}})^3$$
$$\lesssim \frac{\log(nMM_u)}{\epsilon_c^2} M^{8dd_0+12d+16d_0+16} M_u^8 \epsilon^{-16d_0-12}.$$

This completes the proof. $\qquad\square$

Then we compute the covering number of loss function class with Lemma I.3, using the same approach as in Lemma F.5.

**Lemma I.4** (Covering Number Bound for Loss Function Class, Modified from Lemma F.5). *Let $\epsilon_c > 0$. Suppose that for every given $x \in \mathcal{S}$, $u(x,t)$ is bounded and Lipschitz continuous with respect to $t$, such that $\|u(x,t)\|_2 \le M_u$ and Lipschitz constant is $L_u$. We define the loss function class by*

$$F_{\text{loss}} := \{\ell(x; u_\theta)|u_\theta \in \mathcal{T}_R^{h,s,r}\},$$

*where $\mathcal{T}_R^{h,s,r}$ is the transformer class with parameter bound given in Theorem H.3. Then we have:*

$$\log \mathcal{N}(F_{\text{loss}}, \epsilon_c, \{x_i\}_{x_i \in \mathcal{S}}, |\cdot|) \lesssim \frac{\log(MM_u) - \log(\epsilon_c)}{\epsilon_c^2} M^{8dd_0+12d+16d_0+16} M_u^{10} \epsilon^{-16d_0-12}.$$

*Proof.* The proof is modified from the proof of Lemma F.5.

Consider $\{x_i\}_{i=1}^n \in \mathcal{S}$ and function $u_1(x,t), u_2(x,t)$ satisfying $\|u_1(x,t) - u_2(x,t)\| \le \delta$ for all $x \in \mathcal{S}$ and $t = 0, \frac{1}{\lceil \frac{L_\mathcal{T}}{\delta} \rceil}, \frac{2}{\lceil \frac{L_\mathcal{T}}{\delta} \rceil}, \ldots, 1$. Then since $u_1$ and $u_2$ is Lipschitz continuous with Lipschitz constant $L_\mathcal{T}$ under $\ell_2$-norm, for $x \in \mathcal{S}$ and $t \in [0,1]$ we have $\|u_1(x,t) - u_2(x,t)\| \le 2\delta$.

Further, for $x = x_i, 1 \le i \le n$ we have:

$$|\ell(x; u_1) - \ell(x; u_2)|$$
$$= |\int_0^1 \mathop{\mathbb{E}}_{X_0 \sim p_0, X_t \sim p_{t|x_0=X_0, x_1=x}} (\|u_1(X_t,t) - u(X_t,t)\|_2^2 - \|u_2(X_t,t) - u(X_t,t)\|_2^2)\mathrm{d}t|$$

(By definition of $\ell$)

$$= |\int_0^1 \mathop{\mathbb{E}}_{X_0 \sim p_0, X_t \sim p_{t|x_0=X_0, x_1=x}} (u_1(X_t,t) - u_2(X_t,t))^\top (u_1(X_t,t) + u_2(X_t,t) - 2u(X_t,t)))\mathrm{d}t|$$

$$\le 2\delta \int_0^1 \mathop{\mathbb{E}}_{X_0 \sim p_0, X_t \sim p_{t|x_0=X_0, x_1=x}} \|u_1(X_t,t) + u_2(X_t,t) - 2u(X_t,t)\|_2 \mathrm{d}t$$

$$\leq 8\delta C_{\mathcal{T}},$$

where the third line is by $\|u_1 - u_2\|_2 \leq 2\delta$, and the last line is by assuming $C_{\mathcal{T}} \geq M_u$ without losing generality. Therefore, suppose $\mathcal{U}$ is a $\epsilon_c$-covering of $\mathcal{T}_R^{h,s,r}$ with respect to point set $S$, function class $\mathcal{L} = \{\ell(x; u) | u \in \mathcal{U}\}$ is a $4\epsilon_c C_{\mathcal{T}}$-covering of $F_{\text{loss}}$. Then we get:

$$
\begin{aligned}
\log \mathcal{N}(F_{\text{loss}}, \epsilon_c, \{x_i\}_{x_i \in \mathcal{S}}, |\cdot|) &\leq \log \mathcal{N}(\mathcal{T}_R^{h,s,r}, \frac{\epsilon_c}{8C_{\mathcal{T}}}, M^d(\lceil \frac{L_{\mathcal{T}}}{\epsilon_c} \rceil + 1), |\cdot|) \\
&\lesssim \frac{\log(MM_u) - \log(\epsilon_c)}{\epsilon_c^2} C_{\mathcal{T}}^2 M^{8dd_0 + 12d + 16d_0 + 16} M_u^8 \epsilon^{-16d_0 - 12} \\
&\qquad\qquad\qquad\qquad\qquad\qquad\qquad\qquad\qquad\qquad \text{(By Lemma I.3)} \\
&\lesssim \frac{\log(MM_u) - \log(\epsilon_c)}{\epsilon_c^2} M^{8dd_0 + 12d + 16d_0 + 16} M_u^{10} \epsilon^{-16d_0 - 12}. \\
&\qquad\qquad\qquad\qquad\qquad\qquad\qquad\qquad\qquad\qquad\quad (C_{\mathcal{T}} = O(M_u))
\end{aligned}
$$

This completes the proof. $\qquad\square$

We now bound (I) with the concept of covering number.

**Lemma I.5** (Generalization Bound, Modified from Lemma F.6). *Let $\widehat{u}_\theta$ with parameter $\widehat{\Theta}$ be the velocity estimator trained by minimizing $\widehat{\mathcal{L}}_{\text{CDFM}}$ with i.i.d training samples $\{x_i\}_{i=1}^n$, where $x_i \in \mathcal{S}$. For simplicity, we use $\mathcal{N}$ to denote $\mathcal{N}(F_{\text{loss}}, \epsilon_c, \{x_i\}_{x_i \in \mathcal{S}}, |\cdot|)$. Then we bound (I), the generalization bound as:*

$$
\mathop{\mathbb{E}}_{\{x_i\}_{i=1}^n} [\mathcal{R}(\widehat{\Theta}) - \widehat{\mathcal{R}}(\widehat{\Theta})] \lesssim \mathop{\mathbb{E}}_{\{x_i\}_{i=1}^n} [\widehat{\mathcal{R}}(\widehat{\Theta})] + O(\frac{\kappa}{n} \log \mathcal{N} + \epsilon_c),
$$

*where $\kappa$ denote the upper bound of $\ell(x; u_\theta)$.*

*Proof.* See the proof of Lemma F.6. Differences in notations do not influence the conclusion. $\qquad\square$

The next lemma bounds (II) with the approximation theory Theorem H.3.

**Lemma I.6** (Empirical Risk Bound, Modified from Lemma F.7). *Consider the transformer class $\mathcal{T}_R^{h,s,r}$ with parameter bound given in Theorem H.3. Let $\widehat{u}_\theta \in \mathcal{T}_R^{h,s,r}$ with parameter $\widehat{\Theta}$ be the velocity estimator trained by minimizing $\widehat{\mathcal{L}}_{\text{CDFM}}$ with i.i.d training samples $\{x_i\}_{i=1}^n$, where $x_i \in \mathcal{S}$. Let empirical risk $\widehat{\mathcal{R}}(\widehat{\Theta})$ be as defined in (I.2). Then we have:*

$$
\mathop{\mathbb{E}}_{\{x_i\}_{i=1}^n} [\widehat{\mathcal{R}}(\widehat{\Theta})] \lesssim \epsilon^{\frac{4}{M^d + 2}} M^{\frac{4M^d dd_0 + 13M^d d + 8M^d d_0 + 12M^d}{M^d + 2}} M_u^{\frac{8M^d}{M^d + 2}}.
$$

*Proof.* See the proof of Lemma F.7. The only difference is using the approximation error given in Theorem H.3 instead of the approximation error given in Theorem 4.7 in proof. $\qquad\square$

## I.3 ESTIMATION RATES FOR DISCRETE FLOW MATCHING

In this section, we derive the estimation error bounds for discrete flow matching in general case.

**Theorem I.7** (Discrete Flow Matching Velocity Estimation with Transformer). *Let $\widehat{u}_\theta$ with parameter $\widehat{\Theta}$ be the velocity estimator trained by minimizing empirical conditional discrete flow matching loss $\widehat{\mathcal{L}}_{\text{CDFM}}$ with i.i.d training samples $\{x_i\}_{i=1}^n$, where $x_i \in \mathcal{S}$. Suppose that for every given $x_i \in \mathcal{S}$, $u(x_i, t)$ is bounded, such that $\|u(x_i, t)\|_2 \leq M_u$. Moreover, for every given $x_i \in \mathcal{S}$ $u(x_i, t)$ is Lipschitz continuous with respect to $t$. Then for large enough $n$ we have:*

$$
\mathop{\mathbb{E}}_{\{x_i\}_{i=1}^n} [\mathcal{R}(\widehat{\Theta})] \lesssim M_u^8 M^{4dd_0 + 12d + 8d_0 + 12} n^{-\frac{1}{4M^d d_0 + 3M^d + 8d_0 + 9}} (\log n)^{\frac{1}{4M^d d_0 + 3M^d + 8d_0 + 9}}.
$$

*Proof.* Recall the decomposition given in (I.3):

$$
\mathop{\mathbb{E}}_{\{x_i\}_{i=1}^n} [\mathcal{R}(\widehat{\Theta})] = \mathop{\mathbb{E}}_{\{x_i\}_{i=1}^n} [\mathcal{R}(\widehat{\Theta}) - \widehat{\mathcal{R}}(\widehat{\Theta})] + \mathop{\mathbb{E}}_{\{x_i\}_{i=1}^n} [\widehat{\mathcal{R}}(\widehat{\Theta})].
$$

Substituting the result of Lemma I.5 and Lemma I.6 into the decomposition, we get:

$$\mathbb{E}_{\{x_i\}_{i=1}^n}[\mathcal{R}(\widehat{\Theta})] \lesssim O(\frac{\kappa}{n}\log\mathcal{N} + \epsilon_c) + 2\mathbb{E}_{\{x_i\}_{i=1}^n}[\widehat{\mathcal{R}}(\widehat{\Theta})] \qquad \text{(By Lemma I.5)}$$

$$\lesssim O(\frac{\kappa}{n}\log\mathcal{N} + \epsilon_c) + \epsilon^{\frac{4}{M^d+2}}M^{\frac{4M^d dd_0+13M^d d+8M^d d_0+12M^d}{M^d+2}}M_u^{\frac{8M^d}{M^d+2}} \quad \text{(By Lemma I.6)}$$

$$\lesssim \frac{\log(MM_u\epsilon) - \log(\epsilon_c)}{n\epsilon_c^2}M^{8dd_0+12d+16d_0+16}M_u^{12}\epsilon^{-16d_0-12}$$

$$+ \epsilon_c + \epsilon^{\frac{4}{M^d+2}}M^{\frac{4M^d dd_0+13M^d d+8M^d d_0+12M^d}{M^d+2}}M_u^{\frac{8M^d}{M^d+2}}. \quad \text{(By Lemma I.4 and } \kappa \lesssim M_u^2\text{)}$$

We then choose $\epsilon$ and $\epsilon_c$ to get an optimal estimation.

First, let $\epsilon_c = (\frac{\log(n\epsilon)M^{8dd_0+12d+16d_0+16}M_u^{12}\epsilon^{-16d_0-12}}{n})^{1/3}$ and we obtain:

$$\mathbb{E}_{\{x_i\}_{i=1}^n}[\mathcal{R}(\widehat{\Theta})] \lesssim (\log(n\epsilon))^{1/3}M_u^4 M^{\frac{8dd_0+12d+16d_0+16}{3}}\epsilon^{-\frac{16d_0+12}{3}}n^{-\frac{1}{3}}$$

$$+ \epsilon^{\frac{4}{M^d+2}}M^{\frac{4M^d dd_0+13M^d d+8M^d d_0+12M^d}{M^d+2}}M_u^{\frac{8M^d}{M^d+2}}.$$

Next, let $\epsilon = M^{-\frac{1}{4}d+1}n^{-\frac{M^d+2}{16M^d d_0+12M^d+32d_0+36}}(\log n)^{\frac{M^d+2}{16M^d d_0+12M^d+32d_0+36}}$, then for large enough $n$ we get:

$$\mathbb{E}_{\{x_i\}_{i=1}^n}[\mathcal{R}(\widehat{\Theta})] \lesssim M_u^8 M^{4dd_0+12d+8d_0+12}n^{-\frac{1}{4M^d d_0+3M^d+8d_0+9}}(\log n)^{\frac{1}{4M^d d_0+3M^d+8d_0+9}}.$$

This completes the proof. □

## I.4 DISCRETE FLOW MATCHING DISTRIBUTION ESTIMATION

Finally, we present the distribution estimation rate for discrete flow matching in general case.

**Theorem I.8** (Discrete Flow Matching Distribution Estimation Rates)**.** *Let $\widehat{u}_\theta$ with parameter $\widehat{\Theta}$ be the velocity estimator trained by minimizing empirical conditional discrete flow matching loss $\widehat{\mathcal{L}}_{\text{CDFM}}$ in Theorem I.7. Let $P$ stand for the true distribution and $\widehat{P}$ stand for the generated distribution with discrete flow matching network $\widehat{u}_\theta$. Then we have:*

$$\mathbb{E}_{\{x_i\}_{i=1}^n}[TV(P, \widehat{P})] \lesssim M_u^4 \exp(M_u M^d)M^{2dd_0+\frac{13}{2}d+4d_0+6}n^{-\frac{1}{8M^d d_0+6M^d+16d_0+18}}(\log n)^{\frac{1}{8M^d d_0+6M^d+16d_0+18}}.$$

*Proof.* Following Theorem C.4, we have:

$$\text{TV}(P, \widehat{P}) \lesssim \exp(2M_u)M^{\frac{d}{2}}\sqrt{\mathcal{R}(\Theta)}.$$

We then take expectations at both sides and apply the velocity estimation rate (Theorem I.7),

$$\mathbb{E}_{\{x_i\}_{i=1}^n}[\text{TV}(P, \widehat{P})] \lesssim \exp(2M_u)M^{\frac{d}{2}}\mathbb{E}_{\{x_i\}_{i=1}^n}[\sqrt{\mathcal{R}(\Theta)}]$$

$$\lesssim M_u^4 \exp(2M_u)M^{2dd_0+\frac{13}{2}d+4d_0+6}n^{-\frac{1}{8M^d d_0+6M^d+16d_0+18}}(\log n)^{\frac{1}{8M^d d_0+6M^d+16d_0+18}}.$$

$$\text{(By Theorem I.7)}$$

This completes the proof. □

## J NUMERICAL STUDIES

In this section, we validate Theorems 3.1, 5.1 and 5.2 with proof-of-concept experiments.

### J.1 VALIDATING INTRINSIC ERROR BOUND (THEOREM 3.1)

In this section, we examine Theorem 3.1 through numerical experiments.

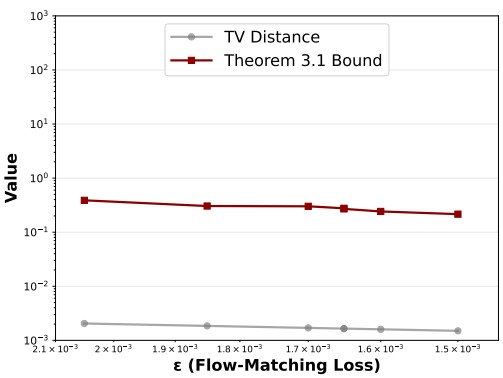

Figure 3: **Validating TV Distance Bound in Theorem 3.1 via Learned Velocity.** The x-axis shows the flow-matching loss is decreasing. The y-axis displays metric values on a logarithmic scale. The theoretical bound (red line) consistently upper-bounds the true TV distance (gray line), validating the guarantee from Theorem 3.1. Both metrics decrease as the model's flow-matching loss decreases.

Figure 4: **Validating TV Distance Bound in Theorem 3.1 via Training-Free Velocity.** The x-axis shows the flow-matching loss is decreasing. The y-axis displays metric values on a logarithmic scale. The theoretical bound (red line) consistently upper-bounds the true TV distance (gray line), validating the guarantee from Theorem 3.1. Both metrics decrease as the model's flow-matching loss decreases.

**Setup.** We validate the Total Variation (TV) distance between the estimated distribution $\widehat{P}$ and the ground-truth distribution $P$ is bounded by factorized velocity risk function $\mathcal{R}^{i_0}$ (F.1):

$$\text{TV}(P, \widehat{P}) \lesssim \sqrt{M} \exp(2M_u) \sum_{i_0 \in [d]} \sqrt{\mathcal{R}^{i_0}(\widehat{\Theta})}, \tag{J.1}$$

where $M$ is the vocabulary size, $M_u$ is the upper bound on velocity entries.

To show the generality of Theorem 3.1, we examine two settings:

- **Training-Free Flow.** We first validate Theorem 3.1 with velocity fields $u_\theta(x, t)$ given. We remark that, this particular setting subsumes the *near-optimally trained models* in practice (i.e., $u_\theta$ is almost ground truth, up to some small noise $\xi$), and our experiments (Figure 4) verify such setting directly.

- **Learned Flow.** We then validate Theorem 3.1 with velocity fields $u_\theta(x, t)$ learned from synthetic data.

**Data.** For each setting, we use the following synthetic data:

- For **training-free flow**, we work in $\mathbb{R}^1$ with discrete vocabulary size $M = 2$. The systems start at state 0. We set the true velocity $u(x, t)$ as
$$u(1, 0) = \alpha \quad u(1, 1) = 0 \quad u(0, 1) = 0 \quad u(0, 0) = -\alpha,$$
which satisfies the marginalization constraint $\sum_y u(y, x) = 0$ for each $x \in 0, 1$. Under this velocity, the distribution path $p_t(1) = 1 - e^{-\alpha t}, p_t(0) = e^{-\alpha t}$. For the estimated velocity $u_\theta$ we introduce a small constant estimation error $\xi > 0$, i.e.,
$$u_\theta(1, 0) = \alpha + \xi, \quad u_\theta(1, 1) = 0, \quad u_\theta(0, 1) = 0, \quad u_\theta(0, 0) = -\alpha - \xi,$$

Under this velocity, the estimated distribution path $q_t(1) = 1 - e^{-(\alpha+\xi)t}$, $q_t(0) = e^{-(\alpha+\xi)t}$.

- For **learned flow**, we work in $\mathbb{R}^2$ with discrete vocabulary size $M = 16$. We obtain discrete data $p_1$ from discretizing the continuous `make_moons` dataset with noise level 0.05 following the implementation in (Lipman et al., 2024):

$$p_1 \sim x = \text{Round}\left(\text{Clip}\left(35 \cdot x_{\text{continuous}} + 50, 0, M - 1\right)\right),$$

where $x \in \{0, 1, \dots, M - 1\}^2$. The initial distribution is $p_0 = U(\{0, 1, \dots, M - 1\}^2)$. We use training set sizes $n = 5000$.

We perform the interpolation using the discrete flow matching scheme:

$$X_t = \begin{cases} X_1 & \text{with probability } t, \\ X_0 & \text{with probability } 1 - t, \end{cases}$$

where $t \sim U[0, 1]$ during training.

**Model.** We parametrize the flow matching velocity $u_\theta(x, t) : \{0, \dots, M - 1\}^2 \times [0, 1] \to \mathbb{R}^{2 \times M}$ with a three layer MLP of hidden dimension 256. The model outputs logits for each dimension and vocabulary class. We optimize using Adam with learning rate $\lambda = 0.001$ and batch size $B = 256$.

**Baseline (Ground Truth).**

- **Training-Free Flow.** For the left-hand side of (J.1), we compute the Total Variation distance using its definition:

$$\text{TV}(P, \widehat{P}) = \sum_{x \in \mathcal{S}} |p_1(x) - q_1(x)|. \tag{J.2}$$

  It is straightforward to calculate ground-true probability $p_1(x)$. For $q_1(x)$, we use Monte Carlo simulation. Specifically, we generate 2000 samples from the flow model, and estimate probability $q_1(x)$ by empirical frequency.

  For the velocity risk, we use the true conditional velocity $u_t(X_0, X_1, t) = X_1 - X_0$. We then estimate:

$$\widehat{R}(\theta) = \frac{1}{N} \sum_{i=1}^{N} \mathcal{L}(u_\theta(X_t^{(i)}, t^{(i)}), X_1^{(i)}),$$

  where the loss $\mathcal{L}$ is the cross-entropy loss.

- **Learned Flow.** For the TV distance, we follow (J.2). In the learned flow setting, we use cross entropy loss to train the model, as no open-source implementation supports the $L_2$ velocity loss. This is justifiable. In the discrete setting, once the interpolation path (mixture path) is fixed, the target velocity field is determined by the path design. Hence, there exists a single correct velocity regardless of the chosen parametrization. Our cross entropy (CE) loss is an alternative parametrization of the same target velocity (Lipman et al., 2024). In particular, minimizing CE corresponds to minimizing the prediction error of the marginal posterior. The true velocity is a deterministic transform of this posterior (Equation 7.25 of (Lipman et al., 2024)). Therefore, driving CE loss to zero forces the learned model to converge to the unique correct velocity. In this sense CE loss is equivalent to $L_2$ loss.

**Results.** We present our results in Figure 4 and Figure 3:

- **Training-Free Flow (Figure 4).** We compare the empirical TV distance with the theoretical upper bound in Theorem 3.1. The red curve shows the theoretical bound, and the gray curve shows the measured TV distance. As the flow-matching loss $\epsilon$ decreases, the theoretical bound consistently upper-bounds the empirical TV distance throughout training. This shows tight agreement in the training-free setting.

- **Learned Flow (Figure 3).** A similar comparison reveals a larger gap between the theoretical bound and the measured TV distance. This behavior is expected: neural network training introduces approximation and generalization errors, and hence prevents the learned velocity from matching the true velocity as closely as in the training-free case.

## J.2 VALIDATING CONVERGENCE RATES OF VELOCITY AND DISTRIBUTION ESTIMATION ERRORS (THEOREMS 5.1 AND 5.2)

In this section, we examine Theorems 5.1 and 5.2 through numerical experiments.

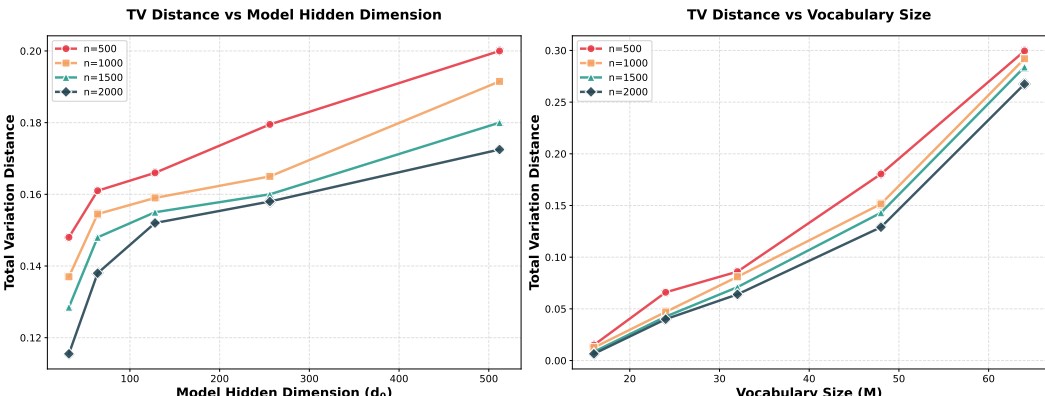

Figure 5: **Scaling Law of Total Variation (TV) Distance (Theorems 5.1 and 5.2).** Left: TV distance increases with model hidden dimension ($d_0$). Right: TV distance grows from near-zero at $M = 16$ to around 0.21 at $M = 64$. With a fixed training sample size $n$, increasing the model hidden dimension $d_0$ or the vocabulary size $M$ increases TV distance. These align well with our finite-sample upper bounds that grow polynomially in these complexity parameters.

**Setup.** Let $n$ be the training sample size. We validate the convergence rates of velocity estimation error and the distribution estimation error

$$\mathop{\mathbb{E}}_{\{x_i\}_{i=1}^n} [\mathcal{R}^{i_0}(\widehat{\Theta}^{i_0})] \lesssim M^{13d_0} n^{-\frac{1}{5Md_0}} (\log n)^{\frac{1}{5Md_0}}, \qquad \text{(Theorem 5.1)}$$

$$\mathop{\mathbb{E}}_{\{x_i\}_{i=1}^n} [\mathrm{TV}(P, \widehat{P})] \lesssim M^{7d_0} n^{-\frac{1}{9Md_0}} (\log n)^{\frac{1}{9Md_0}}, \qquad \text{(Theorem 5.2)}$$

where $M$ is the vocabulary size and $d_0$ is the transformer feature dimension.

**Data.** We use the same synthetic data as the *learned flow setting* in Appendix J.1. We use different training samples $n \in 500, 1000, 1500, 2000$ for our experiments to validate Theorems 5.1 and 5.2.

**Model.** The model architecture, optimzer, learning rate and batch size are follow the *learned flow setting* in Appendix J.1. To validate the scaling law of $d_0$ and $M$ in Theorems 5.1 and 5.2, we report two sets of experiments. First, we first fix the vocabulary size to $M = 256$ and use different model hidden dimension $d_0 \in 32, 64, 128, 256, 512$. Second, we fix the model hidden dimension to $d_0 = 256$ and use different vocabulary size $M \in 16, 24, 32, 48, 64$.

**Baseline (Ground Truth).** We use the same ground truth as the *learned flow setting* in Appendix J.1.

**Results.** We present our results about scaling law of total variation (TV) distance in Figure 5. It presents our findings using different experimental configurations. The left figure in Figure 5 shows that TV distance keeps increasing when the model hidden dimension $d_0$ increases. The right figure in Figure 5 shows that TV distance keeps increasing when the vocabulary size $M$ increases. These results align well with our theory (Theorems 5.1 and 5.2).

