# OpenReview forum: "A Theoretical Analysis of Discrete Flow Matching Generative Models"
_ICLR.cc/2026/Conference — Submitted to ICLR 2026_

### Official Review · Reviewer_8isv · 2025-10-16

**Soundness:** 3
**Presentation:** 3
**Contribution:** 4
**Rating:** 8
**Confidence:** 3

**Summary:**

The authors provide a full error analysis for a discrete flow-matching model parametrized by a transformer, trained on factorized mixture path interpolants. On the technical level, a central result is their construction of a smooth velocity field valued on a continuous domain, extending the true velocity field which takes discrete arguments. This in turn allows them to apply approximation results for transformers, and reach error guarantees, alongside norm bounds for the model parameters.

**Strengths:**

The work extends the growing body of works on theoretical analyses of flow-matching models to the discrete case. This extension requires technical extensions such as Lemma 4.4, which allows to bridge from the discrete case to the continuous case. The contribution is, to the best of my awareness, novel and interesting, and will thus prove of interest to the ICLR community. The paper is overall clearly written, and the main technical ideas sufficiently explained and motivated. In this light, I am overall in favor of acceptance; however, I have limited familiarity with the field, and did not read the proofs in detail.

**Weaknesses:**

My main criticism is the lack of sufficient discussion on certain aspects, which I highlight in the questions section.

**Questions:**

- The main weakness of the bound is an error guarantee exhibiting a curse with the effective dimensionality $Md_0$. Could the authors provide more intuition or discussion on why this is the case? Section $6$ discusses the prefactor but not the rate. Notably, it seems that this depends on the choice of the model through $d_0$ and is not a fundamental limitation. If this is the case, could the authors brieflw discuss this point and how it can be refined in future works?

- I understand that the discrete-to-continuous mapping leverages the natural inclusion embedding of $\mathcal{V}$ in $\mathbb{R}$, with vocabulary items of higher index receiving larger values. Naively, it seems that this embedding, unlike e.g. one-hot encodings, does not treat all vocabulary items in the same fashion, and is not the most natural embedding. Could the authors comment on this, and how this choice impacts (or not) the bounds?
- Further motivation on the choice of transformer architectures would be helpful. In particular, many previous works on continuous generative models studied ResNet or auto-encoder architectures (e.g. Boffi et al; Shallow diffusion networks provably learn hidden low-dimensional structure; 2024). Is the transformer particularly adapted to the discrete case, or simply chosen to reflect most practical settings?
- (Minor) l 266 : "total variation distance bound scales exponentially with vocabulary size M" I am confused as I do not see any exponential dependence in Theorem 3.1.
- I invite the authors to include further comparison with analyses in the continuous case to better highlight the main differences, which would be interesting. In particular, I am wondering whether the authors have an idea how an analysis of the time discretization error would proceed and differ in the discrete setting ?

---

> ### Author Response · Authors · 2025-11-21
> **Rebuttal 1**
>
> ### **Q1.** *The main weakness of the bound is an error guarantee exhibiting a curse with the effective dimensionality. Could the authors provide more intuition or discussion on why this is the case? Section 6 discusses the prefactor but not the rate. Notably, it seems that this depends on the choice of the model through d_0 and is not a fundamental limitation. If this is the case, could the authors briefly discuss this point and how it can be refined in future works?*
>
> ---
>
> **Response:** Thank you for your questions. The dimensional dependence arises from our proof technique, not from an inherent limitation of DFM. Because Transformer universal-approximation results are not pointwise, we must extend the discrete velocity field to a continuous domain. This discrete-continuous gap introduces the dimensionality curse in the approximation/estimation bounds.
>
> Crucially, our intrinsic error bound (Theorem 3.1) does not suffer from this curse. It only requires controlling the velocity risk and is independent of the approximation machinery. The dimensional dependence can likely be improved with pointwise or discrete domain approximation results.
>
> ---
>
> ### **Q2.** *I understand that the discrete-to-continuous mapping leverages the natural inclusion embedding of V in R , with vocabulary items of higher index receiving larger values. Naively, it seems that this embedding, unlike e.g. one-hot encodings, does not treat all vocabulary items in the same fashion, and is not the most natural embedding. Could the authors comment on this, and how this choice impacts (or not) the bounds?*
>
> ---
>
> **Response:** Thanks for the comment. We agree that the embedding we use in the main text is not the standard vocabulary embedding. However, functional extension in Lemma 4.4 holds for any frozen injective embedding of vocabulary. For any admissible injective embedding, analogues of Theorems 4.6, 5.1, and 5.2 remain valid, though the notation and statements become more cumbersome. Our intention is not to impose a restrictive modeling assumption but to keep the analysis simple and transparent.
> To clarify, we have added a Remark 4.1 in our latest revision (`line 324-327`.)
>
> ---

---

> ### Author Response · Authors · 2025-11-21
> **Rebuttal 2**
>
> ### **Q3.** *Further motivation on the choice of transformer architectures would be helpful. In particular, many previous works on continuous generative models studied ResNet or auto-encoder architectures (e.g. Boffi et al; Shallow diffusion networks provably learn hidden low-dimensional structure; 2024). Is the transformer particularly adapted to the discrete case, or simply chosen to reflect most practical settings?*
>
> ---
>
> **Response:** Thank you for your question.
>
> You are correct. It's simply chosen to reflect most practical settings.
>
> Why transformers? Our choice of the Transformer architecture is primarily to match current practical usage in generative modeling. In modern generative modeling, Transformers are the dominant backbone (e.g., Diffusion Transformer, Maskflow [1], DeFoG [2]) for parameterizing velocity (or score) networks. We have pointed this out at our introduction section (line 85 ~ 88).
>
> In addition, **this choice is orthogonal to the generative framework itself.** To clarify, ResNets, auto-encoders, diffusion networks, etc., correspond to different generative model classes. And the Transformer in our work refers only to the function class used to parameterize the velocity field. In principle, our intrinsic error bound (Theorem 3.1) applies to any architecture (MLP, RNN...etc.).
>
> [1] Michael Fuest, Vincent Tao Hu, and Björn Ommer. Maskflow: Discrete flows for flexible and efficient long video generation. arXiv preprint arXiv:2502.11234, 2025.
>
> [2] Yiming Qin, Manuel Madeira, Dorina Thanou, and Pascal Frossard. Defog: Discrete flow matching for graph generation. arXiv preprint arXiv:2410.04263, 2024.
>
> ---
>
> ### **Q4.** *(Minor) l 266 : "total variation distance bound scales exponentially with vocabulary size M" I am confused as I do not see any exponential dependence in Theorem 3.1.*
>
> ---
>
> **Response:** Thank you again for the careful reading. This is a typo: our distance bound scales exponentially with the upper bound on the estimated velocity $M_u$​, not with the vocabulary size $M$. We have corrected this in the revised version.
>
> ---
>
> ### **Q5.** *I invite the authors to include further comparison with analyses in the continuous case to better highlight the main differences, which would be interesting. In particular, I am wondering whether the authors have an idea how an analysis of the time discretization error would proceed and differ in the discrete setting?*
>
> ---
>
> **Response:** Thank you for your questions. Both continuous and discrete flow matching works on the velocity $u_t$ that generates the ODE flow and thereby induces the target distribution. The main difference is that the choice of state space leads to different families of admissible flow paths. For example, the affine path is a typical choice in continuous flow matching, whereas in discrete flow matching we mainly work with mixture paths. These distinct path choices result in different performance across tasks and require different techniques for theoretical analysis. The theoretical analysis for continuous flow matching is more straightforward, since most existing universal approximation theorems are formulated in the continuous setting, as shown in [1]. By contrast, for the discrete flow matching framework considered in our work, we need to develop a technique of functional extension (Lemma 4.5) and a discrete variation-of-constants formula for the error analysis (Lemma C.3).
>
> Regarding the time discretization error, we agree that this is an important direction, but a detailed analysis is beyond the scope of the present work. In particular, the quantitative control of the time discretization error depends on the specific ODE solver and its step–size discretization, and would require a solver–specific numerical analysis that we do not pursue here. Conceptually, however, we do have an outline for such error bound based on our current setting. As demonstrated in Lemma C.3, stability arguments hold for linear ODEs, meaning that distribution estimation error is bounded by velocity estimation error. Under a Lipschitz continuous velocity field and combined with classical error bounds for numerical ODE solvers, we hope to derive a bound on the distribution time discretization error. The above discussion outlines our current approach for analyzing the time discretization error, and indeed is an interesting and meaningful direction for future work.
>
> [1] Maojiang Su, Jerry Yao-Chieh Hu,Yi-Chen Lee,Ning Zhu,Jui-Hui Chung,Shang Wu,Zhao Song, Minshuo Chen, and Han Liu. High-order flow matching:Unified framework and sharp statistical rates. In Proceedings of the 39th Conference on Neural Information Processing Systems (NeurIPS), 2025.

---

> > ### Comment · Reviewer_8isv · 2025-11-24
> > **Acknowledgement of rebuttal**
> >
> > I thank the authors for the clarifications, which fully address my questions. I wish to maintain my score and evaluation, although still with low confidence, since I have not checked in detail the proofs and technical derivations.

---

> > > ### Author Response · Authors · 2025-11-24
> > > **Thank you note**
> > >
> > > Thank you for the positive endorsement. We are happy to hear that our clarifications addressed all your concerns.
> > >
> > > Regarding the validity of our proofs, we would like to draw your attention to our rebuttal exchange with reviewer ziA1.
> > >
> > > In that thread, our appendix was examined through meticulous line-by-line scrutiny.
> > >
> > > All results in our original submission remain correct.
> > >
> > > We hope this removes any remaining uncertainty. Thank you again for your review!

---

### Official Review · Reviewer_4jY8 · 2025-10-29

**Soundness:** 3
**Presentation:** 2
**Contribution:** 2
**Rating:** 4
**Confidence:** 3

**Summary:**

The paper provides the first end-to-end theoretical analysis of Discrete Flow Matching (DFM). It establishes: (1) Intrinsic error bound linking velocity field risk to distributional error; (2)Approximation theory showing Transformers can approximate DFM velocity fields; (3)Estimation theory giving statistical convergence rates for training DFM-Transformers. In summary, the paper produces a complete approximation and estimation theory analogous to classical diffusion and flow matching results but for discrete flow matching.

**Strengths:**

(1) The paper proposes a rigorous end-to-end theory for descrete flow matching.
(2) The chain of results (intrinsic error, approximation error, estimation error) is elegant and well-motivated.
(3) The derivation of total variation bounds via Kolmogorov equations and Grönwall’s inequality seems technically non-trivial and new.

**Weaknesses:**

(1) The paper currently lacks empirical experiments. In pratice, the is hard to make the training objective to be very small while the error bound relies heavily on terms like \sqrt(M)exp(M). The resulting guarantees could be loose for typical M, limiting practical interpretability.
(2) Some modeling assumptions appear stronger than usual, for example, time-Hölder smoothness of CTMC paths, exponentially large UA constants for transformers, and a global velocity-field bound. Clarifying why these are needed and how sensitive the results are to them would improve transparency.

**Questions:**

The theoretical analysis is interesting; I would consider raising my score if the authors address the following question:
(1) Could the authors analyze error bounds for the out-of-vocabulary (OOV) case, now or in future work? The current theory appears to assume a finite discrete state space; with OOV tokens, how should the problem be treated?
(2) Recent work (e.g., Dirichlet Flow, H. Stark 2024; Diffusion LM, Shen et al., 2025) models discrete diffusion/flow using cross-entropy objectives that naturally yield KL bounds. Why is MSE chosen here instead, and why analyze total-variation (TV) bounds rather than KL?

---

> ### Author Response · Authors · 2025-11-21
> **Rebuttal 1**
>
> ### **W1.** *The paper currently lacks empirical experiments. In practice, it is hard to make the training objective to be very small while the error bound relies heavily on terms like $\sqrt{M}\exp{M}$. The resulting guarantees could be loose for typical M, limiting practical interpretability. (need empirical experiment)*
>
> ---
>
> **Response:** Thank you for your comments.
>
> We clarify that the exponential dependence on $M_u$​ is unavoidable due to the application of Grönwall’s inequality in bounding the evolution operator. Similar exponential terms also appear in classical intrinsic error bounds for denoising generative models (e.g., Benton et al., 2023, [1]), which involves $\exp⁡(L)$ with L being the Lipschitz constant of the velocity field.
>
> Moreover, we have conducted controlled experiments to validate our bounds and rate. Our numerical results align well with theory. Please see Section J (in our latest revision pdf) for details.
>
> [1] Benton, Joe, George Deligiannidis, and Arnaud Doucet. "Error bounds for flow matching methods." arXiv preprint arXiv:2305.16860 (2023).
>
> ---
>
> ### **W2.** *Some modeling assumptions appear stronger than usual, for example, time-Hölder smoothness of CTMC paths, exponentially large UA constants for transformers, and a global velocity-field bound. Clarifying why these are needed and how sensitive the results are to them would improve transparency.*
>
> ---
>
> **Response:** Thank you for the comment. We appreciate this opportunity to clarify:
>
> In Assumption 4.2, the time-Hölder smoothness is not as strong as it appears to be. Lemma 4.4 is stated as a more general result for the Hölder class with $\beta \geq 1$, while in later proof we only need the case of $\beta = 1$. For $\beta = 1$, Assumption 4.2 reduces to the statement that velocity is bounded and Lipschitz-continuous. This assumption together with the global boundness are natural for the mixture path case with clipping the time interval, holding for most valid selection of $\kappa(t)$.
> As for the constants in our transformer bounds, the exponentially large constants are unavoidable under the existing contextual-mapping technique [Yun et al., 2019 [1]; Kajitsuka&Sato,2023 [2]; Hu et al., 2024 [3]]. The obtained bounds follow from universal-approximation arguments and therefore represent worst-case upper bounds. This trade-off between generality and complexity is well known in expressiveness theory. For instance, for sufficiently smooth target functions, one can use a coarser grid while achieving the same approximation accuracy, thereby reducing the corresponding upper bound on the parameters.
>
> [1]  Chulhee Yun, Srinadh Bhojanapalli, Ankit Singh Rawat, Sashank Reddi, and Sanjiv Kumar. Are Transformers universal approximators of sequence-to-sequence functions? ICLR 2020 URL https://openreview.net/forum?id=ByxRM0Ntvr
>
> [2]  Tokio Kajitsuka and Issei Sato. Are transformers with one layer self-attention using low-rank weight matrices universal approximators? ICLR 2024 arXiv preprint arXiv:2307.14023,2023.
>
> [3]  Jerry Yao-Chieh Hu,Weimin Wu,Yi-Chen Lee,Yu-Chao Huang,Minshuo Chen,and Han Liu. On statistical rates of conditional diffusion transformers: Approximation, estimation and minimax optimality. ICLR 2025 arXiv preprint arXiv:2411.17522,2024.

---

> ### Author Response · Authors · 2025-11-21
> **Rebuttal 2**
>
> ### **Q1.** *The theoretical analysis is interesting; I would consider raising my score if the authors address the following question: (1) Could the authors analyze error bounds for the out-of-vocabulary (OOV) case, now or in future work? The current theory appears to assume a finite discrete state space; with OOV tokens, how should the problem be treated?*
>
> ---
>
> **Response:** Thank you for this insightful question. Our current theoretical analysis assumes a fixed finite vocabulary, which is consistent with standard tokenization schemes used in modern language models (e.g., BPE or subword tokenization).Analyzing the out-of-vocabulary (OOV) case falls under the broader topic of generalization beyond the training vocabulary, which is outside the scope of the current work.
>
> Nevertheless, we believe our theoretical framework can extend to OOV cases, and we plan to explore this direction in future work. A possible approach is as follows:
>
> Problem redefinition: Introduce OOV tokens as a separate component of the data distribution. When training, we don’t know OOV tokens. We want to characterize the distribution error between data distribution (contain OOV) and estimated distribution.
> Error decomposition: Bound the total distributional error by (i) the discrepancy between the true data distribution (including OOV tokens) and its projection onto the fixed vocabulary space, and (ii) the error between the projected data distribution and the flow-matching–estimated distribution.
>
> This decomposition would allow us to integrate OOV-related errors into the current flow matching framework and extend the existing convergence guarantees to OOV settings.
>
> ---
>
> ### **Q2.** *Recent work (e.g., Dirichlet Flow, H. Stark 2024; Diffusion LM, Shen et al., 2025) models discrete diffusion/flow using cross-entropy objectives that naturally yield KL bounds. Why is MSE chosen here instead, and why analyze total-variation (TV) bounds rather than KL?*
>
> ---
>
> **Response:** Thank you for your question. As discussed in Lipman et al. (2024, Eq. 7.4) [1], the discrete flow matching objective can be formulated under a general Bregman divergence form (containing KL, squared loss etc). The loss used in Gat et al. (2024) corresponds to a specific parameterization of this general form. Our formulation follows the “velocity-parameterized” variant described by Lipman et al. (page 48). It directly models the velocity field. Under this velocity-parameterization, we adopt the squared loss (a special case of the general Bregman divergence) for analytical simplicity.
>
> We focus on the Total Variation (TV) metric because:
> (i) it naturally arises from ODE perturbation analysis in Theorem 3.1. But deriving KL bounds would require stronger regularity assumptions, and
> (ii) TV admits known minimax-optimal rates under Hölder-smooth densities. So we can explore if the convergence rate matches the minimax optimal rate under specific conditions. We leave it as our future work.
>
> [1] Yaron Lipman, Marton Havasi, Peter Holderrieth, Neta Shaul, Matt Le, Brian Karrer, Ricky TQ Chen, David Lopez-Paz, Heli Ben-Hamu, and Itai Gat. Flow matching guide and code. arXiv preprint arXiv:2412.06264, 2024

---

> ### Comment · Reviewer_4jY8 · 2025-11-26
>
> Thank you for your rebuttal to address my concern.

---

> > ### Author Response · Authors · 2025-11-26
> > **Thank you note**
> >
> > We are very happy to hear that our rebuttal was effective.
> >
> > As you mentioned in your initial review, we would be grateful if you could consider score raise to reflect your satisfactions. Or, please let us know if there is anything else we can further clarify.
> >
> > Thank you again for your time, efforts and consideration!

---

### Official Review · Reviewer_ziA1 · 2025-11-01

**Soundness:** 1
**Presentation:** 3
**Contribution:** 2
**Rating:** 2
**Confidence:** 5

**Summary:**

The paper provides a useful bounds for Conditional Discrete Flow Matching (CDFM)  controlling the sampling error in total variation by a so-called riskk directly linked to the training loss. The authors also provide quantitative universality bounds for Transformers in the context of CDFM.

**Strengths:**

- The main body of the paper is well written.
- The main TV bound is useful
- The Roadmap on page 6 is a nice addition to the paper to help understand how the proof works.

**Weaknesses:**

Major: the math is particularly clumsy and not to the expected level of a purely theoretical paper.
- The hypothesis that the velocity approximator is "factorized" seems unnecessary
- The whole Appendix B needs rewritting. In fact, the whole section is a combination of statement and proofs of undergraduate level written in a rather inefficient way. For instance,
    -  Lemmas B.5 and B.8 are standard undergraduate analysis statements (and the proof of B.8 is incorrect under the assumptions used, it is a common mistake but still...). Similar remarks apply to  part of section D.
     -  Lemma B.9 and B.6 are so standard they may probably be stated without proof.  Lemma B.10 would be highly simplified by using the Lipchitz constant of (X,Y) -> X+Y and better used of B.5.
     -  Theorem B.17 is not original and I fail to understand why it is rewritten here or what the proofs written here brings to the discussion.
- Appendix C has too many problems to be acceptable:
     -  Lemma C.3 is FALSE!!!!  It is only true if the matrices $U_{t,\theta}$ commute.
     -  Holder inequality seems to be unknown to the authors as they repeatedly bound the L1 norm of a product by the product of the L1 norms.
     - That being said, I do not doubt the final result, pending replacement of Lemma C.3 by slightly more sophisticated inequality one can obtain using Gronwall Lemma.

Minor
- The chosen embedding of vocabulary is non-standard, quite restrictive and an unnecessary hypothesis.
- The same holds for the timestep embedding.
- Theorem 4.6 statement is unecessarily complicated, the notations of the space of transformers is not defined  - Lemma 4.4 may be most likely found in a functional analysis book, although in the context of an ML article, reproving it directly as in Appendix E is acceptable.
- some typos
- Theorem 4.6 is expected to hold in some form for any architectures for which one is able to prove a quantitative universality theorem. It is not specific to transformers, stating the result only for transformers is rather odd.

**Questions:**

See Weaknesses.

---

> ### Author Response · Authors · 2025-11-16
> **Rebuttal 1**
>
> ### **W1.** *The hypothesis that the velocity approximator is "factorized" seems unnecessary*
>
> ---
>
> **Response:** Thanks for the comment. We respectfully remind the reviewer, the “factorized velocity” was introduced in the original discrete flow matching work (Gat et al., 2024). This design helps mitigate the curse of dimensionality and remains the most practical and widely adopted approach in discrete flow matching. Therefore, our setup aligns with how the method is used in practice.
>
> In addition, our framework can also handle general, non-factorized velocity fields (please see Appendices H and I of our original submission). We mentioned this in the intro (`line 45-47` and `line 93-94`) and the discussion section (`line 431-433` of Sec 6) of the original submission. Our general, non-factorized results are strictly weaker, whereas the main-text theorems using factorized velocity yield better conclusions. These results support why we need factorized velocity in practice.
>
> ---
>
> ### **W2.** *Lemmas B.5 and B.8 are standard undergraduate analysis statements (and the proof of B.8 is incorrect under the assumptions used, it is a common mistake but still...). Similar remarks apply to part of section D.*
>
> ---
>
> **Response:** Thank you for your attention to details. We agree that Lemmas B.5 and B.8 are standard analysis facts. We include them to keep the paper self-contained. Regarding Lemma B.8, we have carefully re-checked the proof under the stated assumptions but could not identify a specific error. When f is differentiable and $\|J\_f\|\_2$ is bounded, $g(t) = f(x1 + t(x_2-x_1)$ is absolute continuous to $t$. We would greatly appreciate it if the reviewer could clarify which step might be problematic.
>
> ---
>
> ### **W3.** *Lemma B.9 and B.6 are so standard they may probably be stated without proof. Lemma B.10 would be highly simplified by using the Lipchitz constant of (X,Y) -> X+Y and better used of B.5.*
>
> ---
>
> **Response:** Thank you for the suggestions. We state the proofs of Lemma B.6 and Lemma B.9 for completeness. As for proof of Lemma B.10, we have simplified the proof according to your suggestion. Please see `line 928-961` of our latest revision pdf.
>
> ---
>
> ### **W4.** *Theorem B.17 is not original and I fail to understand why it is rewritten here or what the proofs written here brings to the discussion.*
>
> ---
>
> **Response:** Thank you for the comment. We agree that Theorem B.17 is a known result and we do not claim originality for it. We restate the theorem because the parameter bound in Theorem B.18 depends on the quantities $\delta$ and $D$ from the proof of B.17. So we restate the proof of Theorem B.17 for completeness.
>
> In response, we have changed Theorem B.17 into a proposition to avoid confusion. Thank you again for pointing this out.
>
> ---
>
> ### **W5.** *Lemma C.3 is FALSE!!!! It is only true if the matrices U commute.*
>
> ---
>
> **Response:** Thanks for your attention to details. We've carefully rechecked proof of Lemma C.3 and do not find any step that assumes $U$ commutes. In particular, we do not use any form of identities $e^A e^B = e^{(A+B)}$. If we have misunderstood your concern, could you please indicate the specific line/equation where a commutation step is implicitly used? Thank you very much!
>
> ---
>
> ### **W6.** *Holder inequality seems to be unknown to the authors as they repeatedly bound the L1 norm of a product by the product of the L1 norms.*
>
> ---
>
> **Response:** Thank you for the pointer. To clarify, our intention throughout is to use induced/operator norms. Under these norms, the submultiplicative property holds directly, so no Hölder inequality step is needed.
>
> ---

---

> ### Author Response · Authors · 2025-11-16
> **Rebuttal 2**
>
> ### **W7.** *That being said, I do not doubt the final result, pending replacement of Lemma C.3 by slightly more sophisticated inequality one can obtain using Gronwall Lemma.*
>
> ---
>
> **Response:** Please see our response above regarding Lemma C.3. We’d appreciate further clarifications.
>
> ---
>
> ### **W8.** *The chosen embedding of vocabulary is non-standard, quite restrictive and an unnecessary hypothesis.*
>
> ---
>
> **Response:** Thanks for the comment. We agree that the embedding we use in the main text is not the standard vocabulary embedding. However, functional extension in Lemma 4.4 holds for any frozen injective embedding of vocabulary. For any admissible injective embedding, analogues of Theorems 4.6, 5.1, and 5.2 remain valid, though the notation and statements become more cumbersome. Our intention is not to impose a restrictive modeling assumption but to keep the analysis simple and transparent.
> To clarify, we have added a Remark 4.1 in our latest revision (`line 324-327`.)
>
> ---
>
> ### **W9.** *The same holds for the timestep embedding.*
>
> ---
>
> **Response:** Thank you for the question. We believe there might be a potential overlook. In our original submission, we explained the necessity of clipping the time interval in Remark E.1 (`line 1772-1781` of our latest revision pdf). Here we quote it for your convenience:
> > **Remark E.1.** We demonstrate that clipping the interval of $t$ is necessary in discrete flow matching, for there doesn't exist a construction of $\kappa(t)$ that keeps stability of $\frac{\dot{\kappa}(t)}{1-\kappa(t)}$ at both $t=0$ and $t=1$. To show this, we set $r(t)=\frac{\dot{\kappa}(t)}{1-\kappa(t)}$ and $g(t)=1-\kappa(t)$. Then we have:
> $$
>     (-\log(g(t))'=\frac{-g'(t)}{g(t)}=r(t).
> $$
> Since $\kappa(0)=0$ and $\kappa(1)=1$, we have $-\log g(0)=0$ and $-\log g(1)=\infty$.
> This means $r(t)=(-\log(g(t))'$ is not bounded on [0,1].
> Then for $\kappa(t)$ finite on (0,1), it must be instable at $t=0$ or $t=1$.
> Therefore, clipping the interval of $t$ is necessary.
>
> ---
>
> ### **W10.** *Theorem 4.6 statement is unnecessarily complicated, the notations of the space of transformers is not defined - Lemma 4.4 may be most likely found in a functional analysis book, although in the context of an ML article, reproving it directly as in Appendix E is acceptable.*
>
> ---
>
> **Response:** Thank you again for your attention to details. As remarked on page 5 and the statement of Thm 4.6, we defer the detailed definition of the transformer function space to Appendix B due to space constraints. As for its further simplicity, we believe the current form is already minimally sufficient to keep the Theorem precise and self-contained. Dropping any further information will unavoidably hamper its clarity. However, we are still open to revision suggestions to make it more concise. Thank you!
>
> ---
>
> ### **W11.** *Theorem 4.6 is expected to hold in some form for any architectures for which one is able to prove a quantitative universality theorem. It is not specific to transformers, stating the result only for transformers is rather odd.*
>
> ---
>
> **Response:** We agree with the reviewer. We agree that tt is feasible to obtain analogous results for other architectures provided the relevant approximation theory is available. However, as stated in our intro, we intentionally focus on Transformer architecture because of its prevalence in practice. Namely, throughout the paper, the velocity field is parameterized by a transformer network, and the subsequent statistical rate analysis relies on its specific norm bounds to compute covering numbers. If you deem necessary, we are open to adding a remark to emphasize the possibility to extend to other architectures.

---

> > ### Comment · Reviewer_ziA1 · 2025-11-16
> >
> > W7. I'll wait for a corrected version of Lemma C3
> >
> > W8. The added remark answers this problem satisfactorily.
> >
> > W9. I am sorry, I should have been clearer. I meant that time embedding in the model is naive as for the vocabulary embedding. I'm guessing the same answer applies as well.
> >
> > W10. I disagree. Deferring such notations to appendix only make the statement harder to read and understand.  However, you can state an unformal version in the main text while keeping details in appendix for readability. As I mention, the final statement is expected for ANY architecture with some quantitative universality. The space of transformer you introduce is merely there to control the hyperparameters of your transformer architecture. It does not play a crucial role in understanding the statement, it plays a role clarifying its scope.
> >
> > I understand you are proud of you big statement but in my honest opinion, that it not a good way to present it.
> >
> > W11. That is a rather secondary point compared to mathematical rigor. I won't push for a remark in main text, a remark in appendix is sufficient in my opinion.

---

> ### Comment · Reviewer_ziA1 · 2025-11-16
> **Let's go back to undergrad**
>
> W1. I understand that and agree. My question is related to the generality of the theory laid out. It's a Theoretical work, limiting the range of the theory because of the applications are unable to follow is not good practice.
>
> W2. Differentiability and integrability are more subtle than you think. You have no guarantee that the Differential is even Riemann integrable.
>
> The right way to prove your Lemma is via 1D Mean value Theorem which comes directly from Rolle Lemma.  That's elementary, there is no need for a sophisticated integration theory that would allow to integrate the Differential or sophisticated property. But then again, I don't even understand why you would reprove it in a research paper except to inflate the work.
>
> BTW: Yes, the proof in Edelman's paper is also wrong. The fact that you copied it unknowingly is a good illustration of why you should not provide an erroneous proof. Let's stop the propagation of imprecise/incorrect mathematics. The 1D use of Newton-Leibniz Theorem would work assuming that you use Kurzweil-Henstock integral (which is not the most commonly taught integral), see for instance  https://dmi.units.it/~fonda/p2017_book_KH.pdf. The use of the change of variable formula in dimension higher than 2 is not a good idea either as it also depends on a sophisticated body of mathematics.
>
> W5. The derivative of  the Backward Flow uses implicitly the exponential formula for the resolvant of the Linear ODE. Make it explicit, you have your faulty step.
>
> W6. Holder inequality is not needed. True, it however improves your bound. Considering that the only good reason I find for the work to reproduce so many elementary steps of reasoning is  to have a tighter bound, Loosing a factor in the scale of feature dimension is rather inefficient.

---

> ### Author Response · Authors · 2025-11-17
> **Rebuttal 3**
>
> ---
> ### **W1.** *I understand that and agree. My question is related to the generality of the theory laid out. It's a Theoretical work, limiting the range of the theory because of the applications are unable to follow is not good practice.*
> ---
>
>  **Response:** Thank you for the clarification. We analyze the factorized velocity in the main text for four reasons.
> 1. Factorized velocity is the standard implementation used in discrete flow matching (e.g., Gat et al., 2024) and is widely adopted in practice.
> 2. The factorized structure enables significantly tighter statistical rates, which is why we present these results in the main text.
> 3. Our framework already covers the general (non-factorized) case in the appendix, but the resulting bounds are weaker.
> 4. Since this work aims to provide sharp bounds for architectures used in practice, we focus the main theorems on the factorized setting while still including the general statements for completeness.
> ---
>
> ### **W2.** *Differentiability and integrability are more subtle than you think. You have no guarantee that the differential is even Riemann integrable.*
> ---
>
>  **Response:** Thanks for your clarification. We agree that the derivative of a differentiable function is not always Riemann integrable. But in our specific setting, one can indeed work with Lebesgue integration and absolute continuity to justify the formula. However, we do respect the reviewer’s suggestions since this choice of integration theory was not clearly specified and could cause confusion. To avoid these subtleties altogether, in the revised version we strengthen the assumption to $f \in C^1$ instead of $f$ being differentiable. Our proof of Lemma C.8 is then fully valid using the standard Riemann integral. This modification does not affect any subsequent arguments. We thank the reviewer once again for pointing this out and helping to prevent the propagation of errors in future work.
>
> ---
>
> ### **W5.** *The derivative of the Backward Flow uses implicitly the exponential formula for the resolvant of the Linear ODE. Make it explicit, you have your faulty step.*
> ---
>
> **Response:** Thank you for your question. To address  your concern, we explain more about the backward equation on line 1354. For the backwards flow, by definition we have $P_{s,t,\theta}P_{r,s,\theta} = P_{r,t,\theta}$, for $r \leq s \leq t$.
> Take the derivative with respect to $s$ and we get
> $$
> (\partial_sP_{s,t,\theta})P_{r,s,\theta}+P_{s,t,\theta}(\partial_sP_{r,s,\theta})= 0
> $$.
> By definition, $\frac{\partial}{\partial_s}P_{r,s,\theta}=U_{s,\theta}P_{r,s,\theta}$.
> Since $P_{s,t,\theta}$ is invertible and thus has full rank, we get
> $$
> \frac{\mathrm{d}}{\mathrm{d} s}P_{s,t,\theta} = -P_{s,t,\theta}U_{t,\theta}.
> $$
> This is the backward equation.
>
> ---
>
> ### **W6.** *Holder inequality is not needed. True, it however improves your bound. Considering that the only good reason I find for the work to reproduce so many elementary steps of reasoning is to have a tighter bound, Loosing a factor in the scale of feature dimension is rather inefficient.*
> ---
>
>  **Response:** Thank you for the suggestion. However, under our current proof technique the dependence on the feature dimension remains at the order of the square root of the dimension($M^{d/2}$ in the general case and $\sqrt{M}$ in the factorized case). Consequently, we find it challenging to further reduce this dependence and have not identified a suitable place where Hölder’s inequality could be applied to obtain a tighter bound. Could you please clarify how Hölder’s inequality would help improve the dependence on the feature dimension in this context?

---

> ### Author Response · Authors · 2025-11-17
> **Rebuttal 4**
>
> ---
>
> ### **W10.** *I disagree. Deferring such notations to appendix only make the statement harder to read and understand. However, you can state an informal version in the main text while keeping details in appendix for readability. As I mention, the final statement is expected for ANY architecture with some quantitative universality.*
> ---
>
> **Response:** Thank you for the thoughtful suggestion. We agree that presenting an informal version of the result in the main text can improve accessibility. We revised Theorem 4.7 in our latest version.
> Regarding the scope of the theorem: while the qualitative statement can indeed hold for any architecture with a quantitative universality theorem, our work aims to provide statistical rates. This is  dependent on architecture-specific norm bounds and covering-number estimates (Theorem F.4). Different architectures (MLPs, Transformers, autoencoders, etc.) lead to different estimation rates. The recent literature on statistical theories for diffusion and flow models all specifies the architecture in order to derive meaningful convergence guarantees (e.g., [1–4]).
>
> [1]  Kazusato Oko, Shunta Akiyama, and Taiji Suzuki. Diffusion models are minimax optimal distribution estimators. volume 202, pages 26517–26582. PMLR, 4 2023. URL https://proceedings.mlr.press/v202/oko23a.html.
>
>
> [2] Hengyu Fu, Zhuoran Yang, Mengdi Wang, and Minshuo Chen. Unveil conditional diffusion models with classifier-free guidance: A sharp statistical theory. arXiv preprint arXiv:2403.11968, 2024.
>
> [3] Kenji Fukumizu, Taiji Suzuki, Noboru Isobe, Kazusato Oko, and Masanori Koyama. Flow matching achieves almost minimax optimal convergence. arXiv preprint arXiv:2405.20879, 2024.
>
> [4] Yuling Jiao, Yanming Lai, Yang Wang, and Bokai Yan. Convergence analysis of flow matching in
> latent space with transformers. arXiv preprint arXiv:2404.02538, 2024.
>
> ---
>
> ### **W11.** *That is a rather secondary point compared to mathematical rigor. I won't push for a remark in main text, a remark in appendix is sufficient in my opinion.*
> ---
>
>  **Response:** Thank you for the suggestion. We have added Remark E.8 (lines 1998–2006) to clarify why we focus on Transformer architectures and to explain how our results can be extended to other architectures that satisfy quantitative universal approximation properties.

---

> ### Comment · Reviewer_ziA1 · 2025-11-17
>
> W2: "But in our specific setting, one can indeed work with Lebesgue integration and absolute continuity to justify the formula."  Absolute continuity is a consequence of the Lemma you are trying to prove...
>
> " To avoid these subtleties altogether, in the revised version we strengthen the assumption to $f \in C^1$ instead of $f$ being differentiable"  This is an option. I suggested "piecewise C^1" to avoid excluding ReLU.
> I urge the authors to take some time browsing an rigorous calcul and integration textbook.  The Rudin or Dieudonne are good options.
>
>  W5. Please accept my apologies. While redoing the proof (the one you wrote) I made a mistake in the order of the resolvant leading to a formula that  requires commutation. Thinking about it again, intuitively, commutativity should not be required.
>
> W6.  I am looking more closely at Theorem C.4 to see how Holder inequality changes your bound. Interestingly, it's a mess. I understand that the L1-norm on matrix is in fact the operator norm induced by the L1-norm of vectors (so really the $L_1/L_\infty$ - norm).
> - Until C.4, ok.
> - Then, I fail to find polite words to describe the computation leading to C.5.
>     - The first comment is about the L1 norm...the true L1 norm seemingly not the operator norm induced by L1. In any case, the Lipchitz constant is $M^{d/2}$ which is missing. If you stick to the true L1-norm then the constant is $M^d$.
>     - The fourth line is surprising: why 2? The comment does not clarify as it mentions the maximum of the sum of absolute values  (I'm thinking it is the sum over the lines of the maximum of the absolute values on each line since it is the operator norm by applying matrices on the left of vectors).
> - Therefore, the 2 should be a $M^{(3d/2)}M_u$   ($M^{3d}$Mu if you stick to L1-norm)
> - The L1/L_infty norm of identity is $M^d$ and the Gronwall Lemma from C.5 gives
> $$|| P_{s,1,\theta} ||_1 \leq || P_{s,s,\theta} ||_1 \exp(\int_{s}^1) M^{3d/2} M_u dr) \leq M^{d} \exp(M^{3d/2}M_u)$$
> - If I take your C.5 and apply Gronwall I get $\exp(2Mu)$ not $2\exp(Mu)$
>
> Now we can continue  from C.4, I have different constant than yours and then Cauchy-Schwarz (or L\infty,L1) -> L2 gives the $M^{d/2}$.

---

> ### Author Response · Authors · 2025-11-18
> **Rebuttal 5**
>
> We thank the reviewer for the careful inspection of Appendix C and appreciate the clarification in W2 and W5.
>
> Regarding W6, we clarify that all matrices “L1 norms’’ in our work are the operator L1 norms induced by the vector L1 norm. That is, the maximum column sum of absolute values. This is the standard convention in matrix analysis for machine learning theory.
>
> The bound on the L1 operator norm of U follows from the rate condition stated in the preliminaries (equation 2.2): each column has non-negative off-diagonal entries, a negative diagonal entry, and zero total sum. Under this structure, the column-wise absolute sum is bounded by $2 M_u$.
>
> To be concrete, let $U_{ij}$ be the $(i,j)$-element of $U_{t,\theta}$. For each column $j$, $\sum_{i=1}^{M^d} |U_{ij}| = -U_{jj} + \sum_{i\neq j} |U_{ij}| \leq 2 M_u$.
>
> Finally, thanks for pointing out the typo in the Grönwall step (a factor of 2) in the original submission and the comment below Eq.C.5. We’ve corrected them accordingly.
>
> Thanks again for your time and efforts invested!

---

> > ### Comment · Reviewer_ziA1 · 2025-11-18
> >
> > - My comment on Holder norm is now moot since the notation is clarified. We agree on the induced norm on matrices. Beware that such notations are not standardised across fields and it may be confusing to some (me for instance).  BTW, I would replace the comment on line 3 leading to equation C.5 by  "By sub-multiplicativity of operator norm" which is general instead of writing "By triangle inequality" which is unclear.
> >
> > - You did not answer my comment regarding the Lipchitz constant of the induced L1 norm on matrices. It has important consequences.
> >
> > - About the fact that each column sums to zero.  Reference to equation 2.2 is made neither in the statement nor in the proof. A direct reference has to be added in both. Also, the bound also depends on the sign assumption (also in 2.2). The same applies to Theorem C.5.
> >
> > - Considering you are trying to be pedagogical by reproving almost every thing. I think the following should be done to significantly improving quality:
> >     - Recall induced norm formula together with induced norm notations. For isntance at the beginning of section C.
> >     - Regarding Lemma B.8. I would suggest not reproving it at all and pointing toward a clean calculus textbook. This is typically the kind of statement that if proved this way conveys either
> >
> >          -  restrictive assumptions (which leads to other people wanting to reprove it later on in some different context, or adds convoluted arguments to circumvent  corner cases) ; or
> >          -  incorrect proofs.
> >
> > Both defeat the purpose of rigorous mathematics and pedagogy.

---

> > ### Comment · Reviewer_ziA1 · 2025-11-18
> >
> > I am ready do go through the whole paper if need be. As long as whenever I look at a proof there is something wrong at first sight I will have doubt on soundness. Things are getting better progressively, I am now open to supporting acceptance once I'm satisfied with the mathematical rigor. I am sorry about the tone I have sometimes but frankly some proofs seem to have been written by an undergrad.

---

> ### Comment · Reviewer_ziA1 · 2025-11-19
> **About Lemma D.1**
>
> First of all, there is an error in the step 3 as you use $n! \leq ( n/e )^n$ which is false as Stirling formula states that $n! \sim \sqrt{2\pi n} ( n/e )^n $. The final bound changes as well...Lemma D.2...all the way to Theorem E.7....hence Also Theorem G.1.
>
>
> Second, allow me to suggest a slight variation over Lemma D.1
> $$\forall x\in [0,1], |\frac{d^n\eta}{dx^n}| \leq 2^{3n+1} n^{2n+1}e^{-2n} n! $$
> The proof is  elementary and does not uses Residue theorem.
>
> Define for a polynomial $P$, define  $[P]$ the max of the absolute value of its coefficients.
>
> Start from the recurrence relation on $p_n$, from it you deduce:
> - $\mathrm{deg} (p_n) = \mathrm{deg} (p_n) +2$
> - $[p_{n+1}] \leq [p_n]+ [p_n'] \leq (\mathrm{deg} (p_n)+1) \times [p_n]$
> - $p_0=1$ so $[p_0]=1$ and $\mathrm{deg} (p_0)=0$
>
> Therefore $ \mathrm{deg} (p_n) = 2 n $
> and,  $$[p_n]\leq \prod_{i=0}^{n-1} (2i+1) \leq 2^n n!  $$
>
> From that you deduce that, using the change of variable $y = \frac{1}{1-x}$
> $$ |\frac{d \eta^n}{dx^n}(x) | \leq 2n \times 2^n n!  y^{2n} e^{-y}$$
>
> Study the function $y^{2n}e^{-y}$ for $y \geq 1$, it achieves it maximum at $y=2n$  with value $(2n)^{2n}e^{-2n}$
>
> Therefore:
>
> $$ |\frac{d\eta^n}{dx^n}| \leq 2^{3n+1} n^{2n+1}e^{-2n} n! $$
>
> This changes Lemma D.2 a bit with a constant that is weaker for what matters is the domination in  $M^d$.

---

> > ### Author Response · Authors · 2025-11-22
> > **Rebuttal 7: reponse to alternative proof of Lemma D.1**
> >
> > Thanks for the suggestions and your attention to details.
> >
> > You are absolutely correct. There’s indeed a typo in the Stirling inequality in step 3 of Lemma D.1.
> >
> > In response, we have now added a constant 3 to the result of Lemma D.1 to fix this issue.
> >
> > We would like to clarify that the conclusion of Lemma D.1 is only used in computing the Holder constant in Lemma D.2. The upper bound of the derivatives in Lemma D.1 is given to make it explicit that the functional extension in Lemma D.2 lies in the Hölder class. However, the precise value of Holder constant does not influence subsequent argument (by the definition of Holder class). Therefore, the precise value of this constant does not affect any of our main results, from Thm 4.7 to Thm 5.2. Thanks for the opportunity to clarify.
> >
> > Lastly, we thank the reviewer for proposing an alternative proof for Lemma D.1, and for taking the time to work out these details.

---

> ### Author Response · Authors · 2025-11-21
> **Rebuttal 6**
>
> Thank you for the detailed follow-up and suggestions. As recommended, we replaced the phrase “By triangle inequality’’ with the clearer and more appropriate “By sub-multiplicativity of the operator norm.’’
>
> For the lipchitz constant of the induced L1 norm on matrices, can you kindly clarify where the problem is? We are not sure where we use Lipchitz constant of induced L1 norm and introduce problems.
>
> In response to referring Eqn 2.2, we now explicitly refer it in both the statement and the proof.  Yet, we emphasize that, this is not the assumption we made. It is the standard property of transition velocity U in the continuous time markov chain. Please see Eqn 6.4 in [1] for more details.
>
> In response to the induced-norm notation, you are absolutely correct. Recalling it in Appendix C would certainly boost the clarity. We have revise accordingly.
>
> Finally, in response to Lemma B.8, we have done our due diligence but failed to find a suitable text for these specific results. Could you kindly direct us to the specific lemma and textbook you have in mind? If you can share the exact source, we will gladly cite it for completeness.
>
> [1] Yaron Lipman, Marton Havasi, Peter Holderrieth, Neta Shaul, Matt Le, Brian Karrer, Ricky TQ Chen, David Lopez-Paz, Heli Ben-Hamu, and Itai Gat. Flow matching guide and code. arXiv preprint arXiv:2412.06264, 2024

---

> > ### Comment · Reviewer_ziA1 · 2025-11-22
> > **"For the lipchitz constant of the induced L1 norm on matrices, can you kindly clarify where the problem is? We are not sure where we use Lipchitz constant of induced L1 norm and introduce problems."**
> >
> > page 27, first line of the equations leading to now C.5.
> > You write as comment: "(By Lipchitz continious of $\ell_1$ norm)".
> >
> > This suggests that you are using the composition Lemma B.5 hence the Lipchitz constant of $x \mapsto ||x ||_1$ on matrix with respect to say to some norm of matrices say the Frobenius norm. This argument is convoluted.
> >
> > I would say that a more accurate comment would be "by triangle inequality" since the straightforward manner to get you inequality without any other factor is
> > $$\left|\frac{||A(t+h)|| - ||A(t)||}{h} \right|\leq \frac{||A(t+h) - A(t)||}{|h|}   $$
> > then go to limit as $h\rightarrow 0$. Which is true for any norm.

---

> ### Comment · Reviewer_ziA1 · 2025-11-22
>
> About Lemma D.1
>
> OK.
>
> Please add a proper reference for "Sitrling inequality". For instance:
> Robbins, H. (1955). A remark on Stirling's formula. The American mathematical monthly, 62(1), 26-29.
> It's an undergrad exercise but considering your  "pedagogical" take, it's necessary. Stirling formula is more complicated to prove than the fundamental theorem of calculus (Newton-Leibniz).

---

> ### Comment · Reviewer_ziA1 · 2025-11-22
> **I think we are done**
>
> I updated my review

---

> ### Author Response · Authors · 2025-11-22
> **Rebuttal 9:  thanks for your new review**
>
> **We notice that you have replaced the entire “Weaknesses” section of your original review.**
>
> **We are happy that our responses have addressed your concerns. Thank you for your careful attention to detail and scrutiny of mathematical rigor.**
>
> **However, we respectfully request the reviewer to restore the original review to ensure the fairness and transparency of our lengthy rebuttal exchange. Especially, the majority of your original accusations have been pushed back perfectly, expect for a few inconsequential typos.**
>
> We also thank the reviewer for your new reflections on rigor, originality, and style. Here are some clarifications:
>
> In response to generality and stylistic choices,
> > More generality other f-divergences or Wasserstein distances instead of only TV.
>
> We agree that extending the analysis to broader classes of divergences (e.g., $f$-divergences or Wasserstein metrics) is meaningful. However, these extensions lie beyond the scope of the current paper and would require substantial new technical development. Existing work in this area likewise focuses on specific bounds rather than attempting such generality. We therefore view these directions as valuable future work and plan to pursue them in follow-up research.
>
>
> In response to reproving standard results,
> > Reproving standard results (the whole section B, section C.1,...) makes the whole appendix heavy to read, obscure the line of argumentation and frankly is borderline plagiarism as all this can be found elsewhere, possibly better written. Properly referencing standard methods beyond citing the last paper the author read is a necessity to maintain a good overall mathematical level. Especially considering the exponential growth of the Machine Learning litterature
>
> We respectfully remind the reviewer that, each appendix section explicitly separates the main proof from the preliminary lemmas. Every result drawn from prior work, including those adapted or modified, is clearly cited. These preliminary components are included to keep the argument self-contained and accessible to general machine-learning readers. In fact, in past submissions we were often asked to provide more details of this kind.
>
> We also emphasize that, as stated throughout the paper, several prior results require adaptation under our assumptions, and minimal restatements are necessary.  It is not feasible to cite a prior work and assert that, after altering assumptions, “its 20-page proof carries over.” This expectation is not realistic. However, we are open to reorganize and move Appendix B to the end of appendix if you deem it necessary.
>
> Readers already familiar with these classical results can safely skip directly to the main proofs.
>
> **If the reviewer genuinely believes that any part of this constitutes plagiarism, we respectfully invite you to flag the ethic concern or submit an academic-integrity report under the ICLR guidelines.**
>
> We appreciate the reviewer’s feedback. The comments have helped us improve the presentation, and we will continue refining the exposition in the revised version.

---

> ### Comment · Reviewer_ziA1 · 2025-11-23
>
> First of all, as requested I put back the former weakness section.
>
> Let me clarify that I am not accusing the authors of plagiarism. To avoid any confusion, I am removing the word and tone down the final review. My goal is not to sink the submission but to state my appreciation of the work.
>
> Regarding the scope and the required extra work to reach applications to other losses and more general bounds. My opinion may be summarized by stating that the work lacks the element of surprise I expect for a first tier conference paper.
> - The statements are expected.
> - Most of the proof provided is irrelevant for research purposes since they are reformulations or special cases of classical statements. Some statements are exercises I solved during my first year of undergraduate study and half of the proof is a composition of statement I would not bother to prove as writing down a proof is a matter of minutes (for me anyway).
> - The proofs provided are not particularly elegant and may be generalised for cheap by simply focusing on the essential parts and using more general statements that are known for 70 years and are taught in mathematics master degree programs.
> -  The work is purely theoretical and consists only of mathematical Theorems, it thus seems reasonable to raise the bar to this level, especially for a first tier conference such as ICLR.
>
>
> I already raised my score from reject to weak reject on the basis that requirements may change from one reviewer to the other. I respect the endeavor, I see it is serious work. The area chair may rule that the good presentation of the main body of the paper and a pedestrian proof  in appendix is acceptable, I won't fight it but I don't have to be enthusiastic about it.

---

### Official Review · Reviewer_nwxq · 2025-11-01

**Soundness:** 3
**Presentation:** 3
**Contribution:** 3
**Rating:** 4
**Confidence:** 3

**Summary:**

This paper provides a theoretical analysis of end‐to‐end training of DFM models. They bound the distance between the generated distribution and the true data distribution by the risk of the learned probability velocity. Then this risk is analyzed through two sources: approximation and estimation errors. The approximation error is determined by the capacity of the transformer, and the estimate error is due to the training on a finite dataset.

The paper provides theoretical justification (for the first time in discrete flow‐matching) that model error can be controlled and that convergence is guaranteed under ideal conditions.

**Strengths:**

The paper is well written. To my knowledge this is the first formal proof of convergence for DFM. It provides a valuable theoretical contribution to the growing area of discrete generative modeling and flow‐matching methods. They decompose the errors for learning the probability velocity into the approximation error (due to model architecture) and the estimation error (due to the sample size), assuming that training is done ideally.

**Weaknesses:**

The analysis is built upon a slightly different version of the loss function from the original discrete flow matching (Gat et al, 2024). Maybe it is better to elaborate a bit more on the connection between those loss functions and justify the use of squared $\ell_2$ as the Bregman divergence.

There is a connection between discrete diffusion models and discrete flow matching. The original DFM training loss is identical to the masked diffusion training loss. There exists earlier literature on the convergence analysis of the discrete diffusion models. It’s better to include some discussions on the similarity and differences, and novelty compared to earlier works on discrete diffusion.

In addition, since the paper only provides upper bounds on the TV distance and probability velocity risk, I feel that it needs some empirical evidence to show how tight those bounds are.

**Questions:**

1. In the original discrete flow matching, the training loss uses cross-entropy terms (see Eq. (28) in Gat et al, 2024). Is it equivalent to the loss term in Eq. (2.10) by using the $\ell_2$-norm in Bregmann divergence?

2. Does using different form of Bregmann divergence affect the results?

3. In Theorem 3.1, is $T = 1$? How big is $t_0$?

4. Is it possible to show some empirical evidence of the scaling behavior (e.g., $M$, $n$, or $d_0$) on some toy data?

---

> ### Author Response · Authors · 2025-11-21
> **Rebutal 1**
>
> ### **W1.** *The analysis is built upon a slightly different version of the loss function from the original discrete flow matching (Gat et al, 2024). Maybe it is better to elaborate a bit more on the connection between those loss functions and justify the use of squared as the Bregman divergence.*
>
> ---
>
> **Response:** Thanks for your comment. As discussed in Lipman et al. (2024, Eq. 7.4), the discrete flow matching objective can be formulated under a general Bregman divergence form (containing KL, squared loss etc). The loss used in Gat et al. (2024) corresponds to a specific parameterization of this general form. Our formulation follows the “velocity-parameterized” variant described by Lipman et al. (page 48). It directly models the velocity field. Under this velocity-parameterization, we adopt the squared loss (a special case of the general Bregman divergence) for analytical simplicity.
>
> ---
>
> ### **W2.** *There is a connection between discrete diffusion models and discrete flow matching. The original DFM training loss is identical to the masked diffusion training loss. There exists earlier literature on the convergence analysis of the discrete diffusion models. It’s better to include some discussions on the similarity and differences, and novelty compared to earlier works on discrete diffusion.*
>
> ---
>
> **Response:** Thanks for pointing this out!
>
> To clarify, as highlighted in Lipman et al. (2024, Section 10.1-10.3), training algorithm for diffusion models is equivalent to training a specific FM model with x0-prediction under time reparameterization.
>
> * In particular, masked discrete diffusion model introduced in Austin et al. (2023, Appendix A.2.2) can be viewed as a special case of mixture path model (after an appropriate time reparameterization), with source distribution $P_0 \sim \delta_(M, M,..., M)$ and cross-entropy loss function.
> * The main difference lies in that discrete flow matching naturally accommodates general Bregman divergence loss functions, which provides additional flexibility.
> * Our main contribution is giving a sample-complexity analysis of discrete flow matching and deriving an explicit convergence rate.
> * Our techniques can be extended to other choices of loss function once equipped with corresponding universal approximation theorems. They are not restricted to MSE used in paper or cross-entropy loss function as in discrete diffusion models.
>
> We have added additional discussions on this in the revised version (see line 730-745 of our latest revision).
>
> [1] Yaron Lipman, Marton Havasi, Peter Holderrieth, Neta Shaul, Matt Le, Brian Karrer, Ricky TQ Chen, David Lopez-Paz, Heli Ben-Hamu, and Itai Gat. Flow matching guide and code. arXiv preprint arXiv:2412.06264, 2024
> [2] Jacob Austin, Daniel D. Johnson, Jonathan Ho, Daniel Tarlow, and Rianne van den Berg. Structured denoising diffusion models in discrete state-spaces. arXiv preprint arXiv:2107.03006, 2023.
>
> ---
>
> ### **W3.** *In addition, since the paper only provides upper bounds on the TV distance and probability velocity risk, I feel that it needs some empirical evidence to show how tight those bounds are.*
>
> ---
>
> **Response:** Thanks for the suggestions. We have conducted controlled experiments to validate our bounds and rate. Our numerical results align well with theory. Please see Section J (in our latest revision pdf) for details.

---

> ### Author Response · Authors · 2025-11-21
> **Rebuttal**
>
> ### **Q1.** *In the original discrete flow matching, the training loss uses cross-entropy terms (see Eq. (28) in Gat et al, 2024). Is it equivalent to the loss term in Eq. (2.10) by using the L_2-norm in Bregmann divergence?*
>
> ---
>
> **Response:** Thanks for the question, please see W1 for a detailed response.
>
> ---
>
> ### **Q2.** *Does using a different form of Bregmann divergence affect the results?*
>
> ---
>
> **Response:** Thanks for the question. Using a different Bregman divergence indeed influences the analysis. Our results rely on applying the transformer’s universal approximation property, which is straightforward under the squared-loss. In contrast, if we were to use the KL divergence, the objective would reduce to a maximum-likelihood form. This makes it difficult to apply standard universal approximation results. Therefore, the squared Bregman divergence is chosen to maintain analytical tractability. In principle, if a similar approximation theorem were available for other objective functions, our framework and rates would extend accordingly.
>
> ---
>
> ### **Q3.** *In Theorem 3.1, is T=1? How big is t_0?*
>
> ---
>
> **Response:** Thanks for the question. $T$ and $t_0$ relies on specific selection of $\kappa$ (i.e., mixture path scheduler, see `Preliminary, line 197-210` for precise definition). In Remark E.1 we explain the necessity of clipping the interval, meaning that we cannot let both $t_0 = 0$ and $T = 1$ hold. However, it’s possible to let either $t_0 = 0$. For example, if we let $\kappa(t) = t$, then $r(t) = \frac{\dot{\kappa}(t)}{1 - \kappa(t)} = \frac{1}{1 - t}$ exists at $t = 0$ so we can take $t_0 = 0$ in this case. Additionally, in this case we cannot let $T = 1$ since $r(1)$ does not exist.
>
> ---
>
> ### **Q4.** *Is it possible to show some empirical evidence of the scaling behavior (e.g., M, n, d_0) on some toy data?*
> ---
> **Response:** Thanks for the suggestions. We have conducted controlled experiments to validate our bounds and rate. Our numerical results align well with theory. Please see Section J (in our latest revision pdf) for details.

---

### Author Response · Authors · 2025-11-29
**A Clarification on the Technical Record in Response to Reviewer ziA1**

Dear Reviewers and Program Chairs (and Future Readers),

Since reviewer ziA1 posted a new review, we believe it’s more appropriate to respond in a separate thread. So here is our public response to reviewer ziA1’s new review.

Also, we believe several aspects of **[reviewer ziA1’s final remark](https://openreview.net/forum?id=opc6Qy9FZH&noteId=XY1HGu9UbH)** require clarification for the sake of fairness and accuracy. We include it here as a front-facing summary of our lengthy, technical rebuttal exchange.

> Regarding the scope and the required extra work to reach application to other losses and more general bounds. My point is that most of the proof provided is irrelevant for research purposes since they are reformulation or special cases of classical statements, some are even exercises I solved during my first year of undergraduate study. Therefore, the whole proof could be significantly shortened (at least half shorter) and generalised for cheap by simply focusing on the essential parts and using more general statements that are known for 70 years and are taught in mathematics master degree programs. Again, considering the work is purely Theoretical and consist on mathematical Theorems, it seems reasonable to raise the bar to this level, especially for a first tier conference.

> I already raised my score from reject to weak reject on the basis that requirements may change from one reviewer to the other. I respect the endeavor, I see it is serious work. Now you have to understand that I am not enthusiastic about it.

Thank you for the time invested into this paper. We respect your right to hold your own opinions and standards as a reviewer. We also agree that, given the enormous ML papers, we need more gatekeeper-style scrutiny.

However, we do not find your review or rebuttal exchange to show utmost professionalism.
Also, several parts of your final remark are inconsistent with the public record and require clarification.

0. **On basic familiarity with the topic.**  We may be wrong, but **your [early remarks](https://openreview.net/forum?id=opc6Qy9FZH&noteId=8VgRaOuC1D) suggest you were not familiar with the core setup of discrete flow matching before the rebuttal**.

   >  The hypothesis that the velocity approximator is "factorized" seems unnecessary

   Even though it is a foundational component of original Discrete Flow Matching (Gat et al., 2024) from Lipman’s group and the standard implementation used in practice.

   [Gat et al., 2024] Gat, Itai, Tal Remez, Neta Shaul, Felix Kreuk, Ricky TQ Chen, Gabriel Synnaeve, Yossi Adi, and Yaron Lipman. "Discrete flow matching." NeurIPS 2024 https://arxiv.org/abs/2407.15595

1. **On technical points.** Every major mathematical claim you raised was examined. And every one was resolved on our side (except for a few inconsequential typos.)  The thread shows multiple misreads and misunderstandings on your end:
    - Lemma C.3 (we clarified in **[Rebuttal 1](https://openreview.net/forum?id=opc6Qy9FZH&noteId=Rod4IhDWfb)**, **[Rebuttal 3](https://openreview.net/forum?id=opc6Qy9FZH&noteId=5qNNgyWmI4)** and you acknowledged **[here](https://openreview.net/forum?id=opc6Qy9FZH&noteId=Lcro5sXEyc)**,
    - the necessary of Holder inequality (we clarified in **[Rebuttal 1](https://openreview.net/forum?id=opc6Qy9FZH&noteId=Rod4IhDWfb)**, **[Rebuttal 3](https://openreview.net/forum?id=opc6Qy9FZH&noteId=5qNNgyWmI4)** and you acknowledged **[here](https://openreview.net/forum?id=opc6Qy9FZH&noteId=YsiJthkKl0)**)
    - the interpretation of our operator L1 norm (we clarified in **[Rebuttal 5](https://openreview.net/forum?id=opc6Qy9FZH&noteId=wIKn5zVtEg)**), and
    - several derivation steps you later revised or retracted  (we clarified in **[Rebuttal 5](https://openreview.net/forum?id=opc6Qy9FZH&noteId=wIKn5zVtEg)** and you acknowledged **[here](https://openreview.net/forum?id=opc6Qy9FZH&noteId=YsiJthkKl0)**).

   Given this, **describing these proofs as “undergraduate exercises” or “a mess” is not consistent with what actually happened in the exchange.**

2. **On the mismatch between tone and correctness.** Throughout the rebuttal exchange, the reviewer uses a strongly condescending tone (“**[Let’s go back to undergrad](https://openreview.net/forum?id=opc6Qy9FZH&noteId=LmLQljF1jw)**”, “**[interestingly, it’s a mess](https://openreview.net/forum?id=opc6Qy9FZH&noteId=Lcro5sXEyc)**”, “**[I fail to find polite words](https://openreview.net/forum?id=opc6Qy9FZH&noteId=Lcro5sXEyc)**”, etc.). This is despite the fact that, **after the detailed rebuttal, all alleged major mathematical errors were resolved on our side**, apart from a few inconsequential typos. The long exchange shows multiple your misreads and miscomputations. Yet, the tone remained the same, including suggestions that we “**[read a calculus textbook](https://openreview.net/forum?id=opc6Qy9FZH&noteId=Lcro5sXEyc)**” while several of your objections misapplied standard analysis results.


## **To be continued**

---

> ### Author Response · Authors · 2025-11-29
> **A Clarification on the Technical Record in Response to Reviewer ziA1- continued1**
>
> 3. **On inflating the paper.** You are still accusing us of “**[inflating the paper with explicit details” and “irrelevant for research purposes](https://openreview.net/forum?id=opc6Qy9FZH&noteId=CoIXAimkHK)**”. We now remind you: even with explicit derivations, you repeatedly misread or mis-computed basic steps in reproducing the proofs (for example, please see **[this (W5)](https://openreview.net/forum?id=opc6Qy9FZH&noteId=LmLQljF1jw)**, **[this](https://openreview.net/forum?id=opc6Qy9FZH&noteId=Lcro5sXEyc)**, **[this (the entire accusation on Lemma C.4 arises from misreading)](https://openreview.net/forum?id=opc6Qy9FZH&noteId=Lcro5sXEyc)** and **[this (on Lipschitz constant)](https://openreview.net/forum?id=opc6Qy9FZH&noteId=KXEAVUNsvJ), etc.)**. This suggests that **omitting those steps would make the arguments less, not more, verifiable for readers who wish to check the mathematics carefully.**
>
>    > Some statements are exercises I solved during my first year of undergraduate study and half of the proof is a composition of statement ***[I would not bother to prove as writing down a proof is a matter of minutes (for me anyway)](https://openreview.net/forum?id=opc6Qy9FZH&noteId=XY1HGu9UbH).***
>
>    Given the nature and number of mistakes documented in the discussion, we find this claim hard to reconcile with the actual record. ***If these steps truly take “minutes,” it is reasonable to expect they would also be done correctly.***
>
> 4. **On the consistency of the review record.** When your points proved invalid, you often did not acknowledge it. Instead, you revised or deleted your original claims, as shown in **[here](https://openreview.net/forum?id=opc6Qy9FZH&noteId=ENTqxkpFiO)** and **[here](https://openreview.net/forum?id=opc6Qy9FZH&noteId=CoIXAimkHK)**. Your stated rationale ([“removing the word and tone to avoid confusion’’](https://openreview.net/forum?id=opc6Qy9FZH&noteId=XY1HGu9UbH)) does not explain why entire criticisms disappeared. **Replacing your original review with a new one is, in our view, unacceptable. It misleads others and harms transparency.** For a theoretical submission, where the line-by-line record matters, this is a significant concern. (We appreciate that you have restored parts of the original text, but this is not sufficient: the full sequence of claims and corrections should remain visible.)
>
> 5. **On your “significantly shorter” proof claim.** You state that the paper could be “significantly shortened (at least half shorter)” by appealing to classical results that can be “generalised for cheap.”(Please see above quotes.) We have to be very clear that **all of your proposed “simplifications’’ have been shown to be either incorrect (i.e., [Holder norm](https://openreview.net/forum?id=opc6Qy9FZH&noteId=5qNNgyWmI4)), not simplifying anything ([Lemma B.10, if you check before and after versions](https://openreview.net/forum?id=opc6Qy9FZH&noteId=Rod4IhDWfb)), or only [yielding weaker special-case statements](https://openreview.net/forum?id=opc6Qy9FZH&noteId=nloiJTfeLx).** The rebuttal record is explicit about each of these points. In light of this, we do not understand why you still insist that the paper could be “significantly shortened (at least half shorter)” by appealing to “classical statements” that are “a matter of minutes” to prove.
>
> **At this point, it feels like you had made up your mind before the rebuttal. Even so, we engaged with utmost respect, preciseness and fairness. But asserting multiple incorrect mathematical claims in a dismissive tone, and then rewriting the public record instead of clearly acknowledging those errors, is not a very nice thing to do.**
>
> The unfortunate recent openreview drama has brought a state of uncertainty to all of us. Even small things like rebuttal communications have become challenging. Even so, we have firm confidence that we acted with the highest fairness and transparency we could. In light of this, we ask that you restore your original review (Edit: we thank ICLR committees accidentally  (and very unfortunately) resolved this). You can add your new review beside it. This is the only way to keep our exchange fair, factual, and transparent for future readers, even if the paper is rejected.
>
> ## **We also ask that you do not take advantage of the fact that, we cannot reply further after the rebuttal period, to reinterpret or selectively edit responses. We expect that the professionalism we have maintained throughout this process earns at least that level of respect in return.**
>
> Best,
>
> Authors

---

### Author Response · Authors · 2025-12-03
**Global Rebuttal Respon**

Dear Reviewers,

We thank the reviewers for your questions and detailed reviews. We have answered all the questions and addressed all the concerns in detail in rebuttal and our latest revision.

The latest revision pdf is readily available. Any changes or modifications made from the revised version are highlighted in blue in this updated draft.

In response to the reviewers' suggestions, we have made revisions to improve the overall readability of the paper. We have conducted another round of proofreading and have fixed all typos. Several parts have been updated for clarity and completeness. These revisions include additional 2 remarks and one new section to help readers to understand our theory better. Most importantly, we added a new numerical study (Appendix J) that validates our main theoretical results. It examines our intrinsic error bound and the convergence rates for velocity and distribution estimation.

## Major Revisions
1. New Numerical Study Section (Appendix J).
We added a new appendix section of numerical studies. Specifically,
Appendix J.1 validates the intrinsic error bound (Theorem 3.1) by comparing empirical total variation distance with our theoretical upper bound.
Appendix J.2 verifies the convergence behaviors predicted by Theorems 5.1 and 5.2, validating both the velocity estimation rates and the distribution estimation rates.
All numerical results align well with our theory.
We further visualize the experimental results in:
Figure 3: Comparison of the theoretical TV bound vs. empirical TV distance under learned velocities, showing that the bound consistently upper-bounds the true TV distance as the flow-matching loss decreases.
Figure 4: The same comparison under training-free velocities, demonstrating a tighter match between the theoretical bound and the empirical TV distance.
Figure 5: Scaling laws of TV distance as a function of model hidden dimension d_0​ and vocabulary size M, illustrating how the empirical estimated tasks aligns with our finite-sample upper bounds.

2. Improved Proof Writing.
We improved clarity in both the main text and appendix. Revisions include rewriting several paragraphs and adding explanations for our proofs:
Lemma B.8. We strengthened the assumption to $f \in C^1$ and provided a concise, self-contained proof based on the Newton-Leibniz formula and Jensen’s inequality.
Lemma B.10. We rewrote the proof of the Lipschitzness of Multi-Head Self-Attention using a clearer proof while obtaining the same Lipschitz constant as before.
Lemma C.4 and Lemma C.5.  We added explicit explanations on how to obtain the upper bound on the L1-norm of the velocity matrix. This makes the transition to equations (C.5) and (C.8) fully transparent and easier to verify.

3. Additional Discussion for Related Works.
We expanded Appendix A with a clearer comparison to prior “discrete diffusion” models. It explains in detail the key similarities and differences. It also highlights the novelty of our intrinsic error bounds and statistical-rate guarantees for discrete flow matching.


## Minor Revisions
1. Remark 3.2. We corrected a typo and revised the remark to clearly state the relationship between our intrinsic error bound, the vocabulary size $M$, and the upper bound of the velocity magnitude $M_u$. The updated remark now accurately reflects how these parameters influence the theoretical bound.


2. Remark 4.1. We added a new remark providing an explanation of our natural embedding construction. This clarification helps readers understand how discrete loss function is extended into the continuous domain in a structurally consistent way.


3. Lemmas C.4 and C.5. We fixed typos in both lemmas and comments to make the proof steps transparent. In particular, the derivation of the intermediate bounds and how these lead to the final inequalities presented in equations (C.5) and (C.8).

4. We hope these revisions address the reviewers' concerns and improve the overall quality of our paper.

---

### Meta-Review · Area_Chair_wMFH · 2025-12-24

**Summary:**

This paper provides a theoretical analysis of the discrete flow matching framework by establishing error bounds between generated and target distributions as well as convergence proof.

Three out of four reviewers were initially negative about this paper and one reviewer positive. Some of the rebuttal discussions seemed to have converged to reviewers indicating willingness to raise the score but this AC does not see if the score was indeed raised or what the new score might be (due to the policy by the PCs), excluding one case. This probably and unfortunately has a negative effect on this final recommendation of this paper.
The reviews indicated the following pros and cons:
+ Clearly written paper, first analysis of this kind, useful theoretical result.
- Complaints about the math and rigor (acknowledging issues this review).
- Connection to previous diffusion convergence results missing.
- Missing numerical experiments to support the theoretical result, quantifying empirically the looseness of the bound and its coefficients.

**Reviewer Concerns:**

See above.

**Reviewer Scores:**

I believe two reviewers would have raised their scores but dont know for sure to what final score.

---

### Decision · Program_Chairs · 2026-01-26

Reject